# Post-infection sequelae of COVID-19 and other infectious diseases—a nationwide Danish study with 40-month follow-up

Clara S. Grønkjær [1], Rune Haubo B. Christensen [1], Daniel Kondziella [2,3,4] & Michael E. Benros [1,3,4]

Nationwide follow-up studies of long-term post-acute COVID-19 sequelae compared with sequelae of other infections have been lacking. Using nationwide registers, we analyzed all SARS-CoV-2 PCR test results, prescriptions for anti-infective agents, and hospitalizations with COVID-19 or other infections in Denmark from March 2020 to June 2023, including up to 40-month follow-up. We used Cox proportional hazards models with time-varying exposures to estimate the rates of first-time mental disorders ($n = 5,306,132$) and general medical conditions ($n = 3,517,630$). Here we show that positive SARS-CoV-2 PCR tests alone were not associated with clinically relevant increased rates of mental disorders or general medical conditions when compared with negative SARS-CoV-2 PCR tests, nor when compared to individuals with anti-infective prescriptions. Rates of general medical conditions after a positive compared with a negative SARS-CoV-2 PCR test were only elevated for virus types preceding Omicron and for individuals with less than 3 vaccination doses. Compared with the general population, the rates of mental disorders or general medical conditions were elevated among hospitalized COVID-19 patients, and particularly when ICU treatment was required. However, when comparing patients hospitalized with COVID-19 to patients hospitalized with non-COVID-19 pulmonary infections or other infections, the rates of mental disorders or general medical conditions were increased to the same extent. In conclusion, severe COVID-19 post-infection sequelae are comparable to sequelae observed after other infections of similar severity.

The COVID-19 pandemic resulted in over 700 million people infected with SARS-CoV-2 worldwide[1]. Ten to 20% of individuals with COVID-19 have been estimated to experience sequelae, even after mild illness, and thus, long-COVID is estimated to impact the lives of at least 65 million individuals[2,3]. However, many other infections, such as influenza and pneumonia, can also lead to post-infection sequelae, and the post-COVID complications have yet to be compared comprehensively with sequelae after other infectious diseases. Strong epidemiological evidence regarding COVID-19 sequelae and whether they differ from sequelae after other infections is of utmost importance for health care planning, including resource allocations, but also for the society through increased

[1]Copenhagen Research Centre for Biological and Precision Psychiatry, Mental Health Centre Copenhagen, Copenhagen University Hospital, Copenhagen, Denmark. [2]Departments of Neurology, Rigshospitalet, Copenhagen University Hospital, Copenhagen, Denmark. [3]Department of Clinical Medicine, Faculty of Health and Medical Sciences, University of Copenhagen, Copenhagen, Denmark. [4]These authors jointly supervised this work: Daniel Kondziella, Michael E. Benros. ✉e-mail: daniel.kondziella@regionh.dk; michael.eriksen.benros@regionh.dk

understanding of the specificity of COVID-19 sequelae compared to other infections.

Prior studies investigating different aspects of COVID-19 and the risk of sequelae have reached conflicting results depending on the time of follow-up, admission status of COVID-19 patients, and whether the control groups were individuals with negative tests, no positive test, or various non-COVID-19 infections[4–19]. Thus, a large-scale study including all of these aspects is needed to clarify the status on the directions and specificity of these associations. Prior large-scale studies and meta-analyses, not comparing to other infections with similar severity, found that individuals with COVID-19 had higher risks of numerous adverse health outcomes, including increased risks of mental, neurological, respiratory[4,5], hematological[6,7], neurodegenerative[8], cardiovascular[9–12], kidney and liver disorders[13], and diabetes[14]. Studies comparing positive with negative SARS-CoV-2 PCR tests among non-hospitalized individuals found higher 6-month post-COVID risks of certain hospital diagnoses, such as venous thromboembolism, although the risk was not higher for serious complications like ischemic stroke, encephalitis, or psychoses[15]. Also, risks of many sequelae declined 2 years post-COVID among hospitalized and non-hospitalized individuals compared with individuals with no evidence of a SARS-CoV-2 infection[6]. Additionally, the risk of new-onset mental and neurological disorders was not significantly higher after COVID-19 when comparing with the increased risks observed after other non-COVID infections[16–19]. This emphasizes that different reference groups are important for clarifying the long-term sequelae and to assess if the findings are specific to COVID-19 in comparison to other non-COVID infections with similar severity. As SARS-CoV-2 only very rarely reaches the brain, the immune response after infection might be one of the main drivers for long-term sequelae, and here the most used measure of general inflammation is C-reactive protein (CRP), which increases with the severity of infection. Thus, CRP has been suggested as a potential biomarker indicative of elevated risk for long COVID symptoms[20], however, few studies have investigated this[21]. Moreover, it is unclear how clinically relevant sequelae of COVID-19 affects individuals not admitted to the hospital with COVID-19 and whether the sequelae following COVID-19 are comparable to those following other infectious diseases.

In this nationwide cohort study, we utilized nationwide Danish registers to comprehensively analyze all SARS-CoV-2 PCR tests and all treated infections across the entire population of Denmark. Our primary objective was to estimate the association between COVID-19 and subsequent first-time mental disorders and general medical conditions from inpatient, outpatient, and emergency room visits, among non-hospitalized and hospitalized individuals. Our secondary objective was to evaluate how these associations compared with those observed after other infectious diseases. The impact of SARS-CoV-2 on subsequent sequelae were investigated by comparing (1) individuals with positive SARS-CoV-2 PCR test to individuals with negative and no SARS-CoV-2 PCR tests, (2) hospital admission with COVID-19 to no admission with COVID-19, and (3) by comparing COVID-19 with other infections of comparable severity to examine to what extent post-infection sequelae were specific for COVID-19. To further nuance the findings, we performed analyses stratified by calendar time to account for differences in virus variants, lockdown periods, and vaccination status. Lastly, to explore the potential role of CRP in the development of long COVID, we analyzed the association between peak CRP levels and COVID-19 sequelae.

## Results
The population of Denmark as of March 1, 2020, consisted of 5,806,583 individuals (see Supplementary Fig. 1) for a schematic visualization of the study population. Of these, 500,451 had a prior hospital contact for mental disorders (including mental and behavioral disorders due to psychoactive substance use, psychotic, mood, and anxiety disorders) and 2,288,953 had a prior hospital contact for general medical conditions (including selected neurological, respiratory, gastrointestinal, circulatory, kidney, endocrine, hematological, musculoskeletal, and dermatological disorders, as well as neuropsychiatric symptoms; see Supplementary Table 1 for number of individuals in the study populations by outcome). When considering the mental health disorders outcome, the study population consisted of 5,306,132 individuals, and when considering the general medical conditions outcome, the study population consisted of 3,517,630 individuals. In total, 2,462,877 individuals had a prior hospital contact for either a mental disorder or a general medical condition (48.9% female, mean age at start (SD) 33.4 (21.2) years) (for cohort characteristics of the study populations of the primary outcomes, see Supplementary Tables 2–4). During the study period, 1,832,328 (54.8%) individuals tested positive for SARS-CoV-2, 1,222,202 (36.6%) only tested negative for SARS-CoV-2, and 289,176 (8.6%) were not tested for SARS-CoV-2 (see Supplementary Fig. 2 for a graphical representation of the number of individuals in each group over time). In total, 9409 (0.3%) individuals were hospitalized with COVID-19, of whom 416 were admitted to ICU (Supplementary Table 5). During the study period, 353,517 (10.6%) individuals had a hospital contact for any mental disorder or general medical condition, of which 83,232 previously tested positive for SARS-CoV-2.

### SARS-CoV-2 positive test results and the rate of any first mental disorder
Compared with individuals not tested for SARS-CoV-2, rates of any first mental disorder were higher after positive SARS-CoV-2 tests (hazard ratio (HR) 1.12, 95% confidence interval (CI) 1.10–1.15, p-value < 0.001) and after negative SARS-CoV-2 tests (HR 1.37, 95% CI 1.35–1.40, p-value < 0.001) (Table 1 and Supplementary Table 6). However, with an HR significantly below one, positive SARS-CoV-2 tests were not associated with higher mental disorder rates compared with individuals with negative SARS-CoV-2 tests (HR 0.82, 95% CI 0.80–0.83, p-value < 0.001). The rates of mental disorder rates did not depend on the age group (Fig. 1 and Supplementary Table 7). The mental disorder rates did not significantly increase with the number of SARS-CoV-2 reinfections (Table 1, Supplementary Table 8). Moreover, there was no significant association between time since testing and mental disorders (Table 1, Supplementary Table 9).

### SARS-CoV-2 positive test results and the rate of any first general medical condition
Compared with individuals not tested, rates of any first general medical condition were higher after positive SARS-CoV-2 tests (HR 1.90, 95% CI 1.87–1.93, p-value < 0.001) and after negative SARS-CoV-2 tests (HR 1.88, 95% CI 1.86–1.90, p-value < 0.001) (Table 1). However, when compared with negative SARS-CoV-2 tests, positive SARS-CoV-2 tests were not overall associated with clinically relevant differences (HR 1.01, 95% CI 1.00–1.02, p-value 0.024).

In age-stratified analyses compared with negative SARS-CoV-2 tests, general medical condition rates were higher for the age groups between 18 and 69 years, lower for the 80+-year-olds (HR 0.88, 95% CI 0.82–0.94, p-value < 0.001), and with no significant difference for the <18 and 70–79-year-olds after positive SARS-CoV-2 tests (Fig. 1 and Supplementary Table 7). Compared with individuals with negative SARS-CoV-2 tests, the general medical condition rates increased with the number of SARS-CoV-2 reinfections (one infection: HR 1.04, 95% CI 1.02–1.05, p-value < 0.001; two or more infections: HR 1.20, 95% CI 1.16–1.24, p-value < 0.001) (Table 1 and Supplementary Table 8). Compared with negative SARS-CoV-2 tests, the rates of general medical condition after a positive SARS-CoV-2 test were highest 1-month post-COVID-19 (HR 1.10, 95% CI 1.06–1.13, p-value < 0.001) and remained significantly elevated until 30 months post-COVID-19 (Table 1 and Supplementary Table 9).

**Table 1 | Hazard ratios and 95% confidence intervals of the association between SARS-CoV-2 test, number of SARS-CoV-2 positive tests, time since SARS-CoV-2 test, admission to hospital with COVID-19, number of readmissions with COVID-19, duration of COVID-19-admission and mental disorders or general medical conditions**

| | Mental disorders[a] | | | General medical conditions[b] | | |
|---|---|---|---|---|---|---|
| | Cases, No. | HR (95% CI)[c] | p-value | Cases, No. | HR (95% CI)[c] | p-value |
| **SARS-CoV-2 test** | | | | | | |
| No SARS-CoV-2 test | 32,049 | 1.00 [reference] | – | 77,865 | 1.00 [reference] | – |
| SARS-CoV-2 negative | 70,638 | 1.37 (1.35–1.40) | <0.001 | 168,329 | 1.88 (1.86–1.90) | <0.001 |
| SARS-CoV-2 positive | 30,292 | 1.12 (1.10–1.15) | <0.001 | 74,638 | 1.90 (1.87–1.93) | <0.001 |
| Positive vs. negative | – | 0.82 (0.80–0.83) | <0.001 | – | 1.01 (1.00–1.02) | 0.024 |
| **Number of positive SARS-CoV-2 tests** | | | | | | |
| SARS-CoV-2 negative | 70,638 | 1.00 [reference] | – | 168,329 | 1.00 [reference] | – |
| 1 positive test | 28,360 | 0.83 (0.82–0.84) | <0.001 | 70,251 | 1.02 (1.01–1.03) | <0.001 |
| 2 positive tests | 1899 | 0.80 (0.77–0.84) | <0.001 | 4309 | 1.17 (1.13–1.20) | <0.001 |
| ≥3 positive tests | 33 | 0.91 (0.64–1.27) | 0.569 | 78 | 1.61 (1.29–2.01) | <0.001 |
| **Time since positive SARS-CoV-2 test** | | | | | | |
| SARS-CoV-2 negative | 70,638 | 1.00 [reference] | – | 168,329 | 1.00 [reference] | – |
| <1 month | 1742 | 0.84 (0.80–0.88) | <0.001 | 4497 | 1.10 (1.06–1.13) | <0.001 |
| 1–2 months | 3463 | 0.82 (0.79–0.85) | <0.001 | 8428 | 1.02 (1.00–1.05) | 0.095 |
| 3–5 months | 5067 | 0.82 (0.80–0.85) | <0.001 | 12,155 | 1.03 (1.01–1.06) | 0.002 |
| 6–11 months | 9888 | 0.83 (0.81–0.85) | <0.001 | 24,982 | 1.05 (1.03–1.07) | <0.001 |
| 12–17 months | 7974 | 0.82 (0.80–0.85) | <0.001 | 19,384 | 1.03 (1.01–1.05) | 0.007 |
| 18–23 months | 1283 | 0.78 (0.74–0.83) | <0.001 | 3059 | 1.08 (1.04–1.12) | <0.001 |
| 24–29 months | 718 | 0.78 (0.72–0.84) | <0.001 | 1788 | 1.07 (1.02–1.12) | 0.004 |
| ≥30 months | 157 | 0.84 (0.72–0.99) | 0.032 | 345 | 1.01 (0.91–1.12) | 0.855 |
| **Hospital admission with COVID-19** | | | | | | |
| No COVID-19-admission[d] | 100,111 | 1.00 [reference] | – | 241,417 | 1.00 [reference] | – |
| COVID-19-admission without ICU[e] | 751 | 1.88 (1.75–2.03) | <0.001 | 1401 | 2.53 (2.40–2.67) | <0.001 |
| COVID-19-admission with ICU[e] | 68 | 2.49 (1.96–3.15) | <0.001 | 149 | 5.09 (4.34–5.98) | <0.001 |
| **Readmissions with COVID-19** | | | | | | |
| No COVID-19-admission[d] | 100,111 | 1.00 [reference] | – | 241,417 | 1.00 [reference] | – |
| 1 COVID-19 admission | 748 | 1.92 (1.78–2.06) | <0.001 | 1446 | 2.70 (2.57–2.85) | <0.001 |
| ≥2 COVID-19 admissions | 71 | 2.20 (1.75–2.78) | <0.001 | 104 | 3.78 (3.12–4.58) | <0.001 |
| **Duration of admission with COVID-19** | | | | | | |
| No COVID-19-admission[d] | 100,111 | 1.00 [reference] | – | 241,417 | 1.00 [reference] | – |
| 1–2 bed days | 314 | 2.07 (1.85–2.32) | <0.001 | 571 | 2.68 (2.47–2.91) | <0.001 |
| 3–6 bed days | 250 | 1.57 (1.38–1.77) | <0.001 | 519 | 2.35 (2.16–2.56) | <0.001 |
| ≥7 bed days | 255 | 2.30 (2.03–2.60) | <0.001 | 460 | 3.10 (2.82–3.39) | <0.001 |

HR hazard ratio, CI confidence interval, ICU intensive care unit.

[a]Including mental and behavioral disorders due to psychoactive substance use, schizophrenia spectrum, mood, and anxiety disorders. The diagnosis codes are summarized in Supplementary Table 27. Results for mental disorders are derived from a study population of n = 4,896,347 individuals, and the first analysis of SARS-CoV-2 tests also includes 409,785 individuals not tested for SARS-CoV-2.

[b]Including selected neurological, respiratory, gastrointestinal, circulatory, kidney, endocrine, hematological, musculoskeletal, and dermatological disorders, as well as neuropsychiatric symptoms. The diagnosis codes are summarized in Supplementary Table 27. Results for general medical conditions are derived from a study population of n = 3,221,166 individuals, and the first analysis of SARS-CoV-2 tests also includes 296,464 individuals not tested for SARS-CoV-2.

[c]The estimates are HRs with 95% CI from Cox proportional hazards model stratified by age and adjusted for confounders (sex, Charlson Comorbidity Index (CCI), parental CCI, parental mental health disorders, employment status, income, highest level of education).

[d]The reference group "no COVID-19-admission" consisted of individuals without admission to a hospital with SARS-CoV-2 infection, i.e., all individuals with negative or positive test results but no admission to hospital.

[e]The categorization of COVID-19-admission and ICU is summarized in Supplementary Table 30.

## SARS-CoV-2 positive test results and time-varying effects

Compared with negative SARS-CoV-2 tests, the rates of first mental disorders and general medical conditions after a positive SARS-CoV-2 test changed over the study period as illustrated by the time-varying hazard ratios in Fig. 2 (Supplementary Table 10). The time-varying hazard ratios show that compared to negative SARS-CoV-2 tests, the rates of mental disorders were consistently lower throughout the study period while the rates of general medical conditions were initially lower but then elevated through most of 2021 for individuals with positive SARS-CoV-2 tests.

## SARS-CoV-2 positive test results and the rates of specific general medical conditions

Compared with negative SARS-CoV-2 tests, rates after a positive SARS-CoV-2 test were higher specifically for neuropsychiatric symptoms (HR 1.08, 95% CI 1.06–1.10, p-value < 0.001), musculoskeletal disorders (HR 1.07, 95% CI 1.05–1.10, p-value < 0.001), and dermatological disorders (HR 1.06, 95% CI 1.02–1.10, p-value 0.002) (Supplementary Table 6). The HRs of incident diagnoses diminished with increasing age for most disorders (Supplementary Table 7 and Supplementary Fig. 3). The HRs for all specific disorders increased by number of reinfections (Fig. 3,

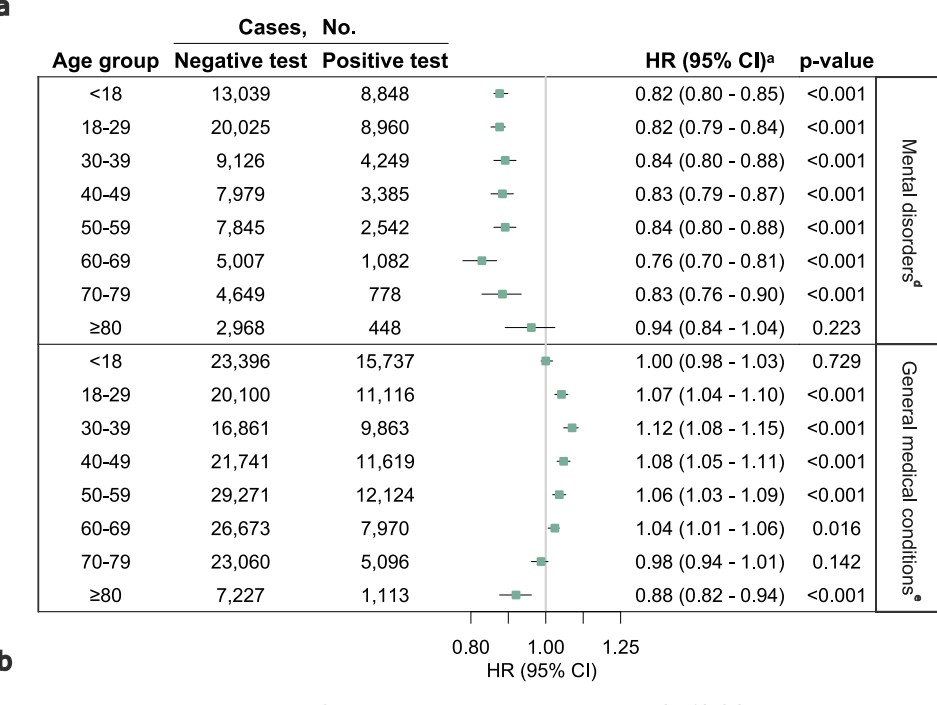

**a**

| Age group | Negative test | Positive test | | HR (95% CI)ᵃ | p-value | |
|---|---|---|---|---|---|---|
| <18 | 13,039 | 8,848 | | 0.82 (0.80 - 0.85) | <0.001 | Mental disordersᵈ |
| 18-29 | 20,025 | 8,960 | | 0.82 (0.79 - 0.84) | <0.001 | |
| 30-39 | 9,126 | 4,249 | | 0.84 (0.80 - 0.88) | <0.001 | |
| 40-49 | 7,979 | 3,385 | | 0.83 (0.79 - 0.87) | <0.001 | |
| 50-59 | 7,845 | 2,542 | | 0.84 (0.80 - 0.88) | <0.001 | |
| 60-69 | 5,007 | 1,082 | | 0.76 (0.70 - 0.81) | <0.001 | |
| 70-79 | 4,649 | 778 | | 0.83 (0.76 - 0.90) | <0.001 | |
| ≥80 | 2,968 | 448 | | 0.94 (0.84 - 1.04) | 0.223 | |
| <18 | 23,396 | 15,737 | | 1.00 (0.98 - 1.03) | 0.729 | General medical conditionsᵉ |
| 18-29 | 20,100 | 11,116 | | 1.07 (1.04 - 1.10) | <0.001 | |
| 30-39 | 16,861 | 9,863 | | 1.12 (1.08 - 1.15) | <0.001 | |
| 40-49 | 21,741 | 11,619 | | 1.08 (1.05 - 1.11) | <0.001 | |
| 50-59 | 29,271 | 12,124 | | 1.06 (1.03 - 1.09) | <0.001 | |
| 60-69 | 26,673 | 7,970 | | 1.04 (1.01 - 1.06) | 0.016 | |
| 70-79 | 23,060 | 5,096 | | 0.98 (0.94 - 1.01) | 0.142 | |
| ≥80 | 7,227 | 1,113 | | 0.88 (0.82 - 0.94) | <0.001 | |

**b**

| Age group | Admissionᶜ | Cases, No. | | HR (95% CI)ᵃ | p-value | |
|---|---|---|---|---|---|---|
| <40 | No admission | 63,987 | | 1.00 [reference] | .. | Mental disordersᵈ |
| | Without ICU | 244 | | 1.52 (1.34 - 1.72) | <0.001 | |
| | With ICU | 16 | | 2.35 (1.44 - 3.84) | 0.001 | |
| 40-59 | No admission | 21,582 | | 1.00 [reference] | .. | |
| | Without ICU | 154 | | 1.80 (1.54 - 2.11) | <0.001 | |
| | With ICU | 15 | | 1.64 (0.99 - 2.72) | 0.056 | |
| 60-79 | No admission | 11,253 | | 1.00 [reference] | .. | |
| | Without ICU | 232 | | 2.69 (2.36 - 3.07) | <0.001 | |
| | With ICU | 31 | | 3.25 (2.28 - 4.62) | <0.001 | |
| ≥80 | No admission | 3,289 | | 1.00 [reference] | .. | |
| | Without ICU | 121 | | 1.85 (1.54 - 2.22) | <0.001 | |
| | With ICU | 6 | | 3.42 (1.54 - 7.63) | 0.003 | |
| <40 | No admission | 96,611 | | 1.00 [reference] | .. | General medical conditionsᵉ |
| | Without ICU | 437 | | 2.46 (2.24 - 2.71) | <0.001 | |
| | With ICU | 25 | | 5.07 (3.43 - 7.51) | <0.001 | |
| 40-59 | No admission | 74,318 | | 1.00 [reference] | .. | |
| | Without ICU | 381 | | 2.59 (2.34 - 2.87) | <0.001 | |
| | With ICU | 56 | | 7.01 (5.39 - 9.11) | <0.001 | |
| 60-79 | No admission | 62,296 | | 1.00 [reference] | .. | |
| | Without ICU | 439 | | 2.85 (2.60 - 3.13) | <0.001 | |
| | With ICU | 64 | | 4.99 (3.90 - 6.37) | <0.001 | |
| ≥80 | No admission | 8,192 | | 1.00 [reference] | .. | |
| | Without ICU | 144 | | 1.90 (1.61 - 2.24) | <0.001 | |
| | With ICU | ≤5ᵇ | | .. | .. | |

Exposure —●— SARS-CoV-2 positive —●— Without ICU —●— With ICU

Supplementary Table 8). The HRs for circulatory, endocrine, hematological, kidney, and respiratory disorders were elevated only during the initial few months following a positive SARS-CoV-2 test, with little to no significant increases thereafter (Supplementary Table 9). In contrast, the HR for neuropsychiatric symptoms remained significantly higher for positive compared with negative SARS-CoV-2 tests across most time intervals after testing.

**SARS-CoV-2 positive test results compared with prescriptions for anti-infective agents regarding rates of long-term sequelae**
Redeeming a prescription for any anti-infective agent was associated with higher rates of mental disorders (HR 1.41, 95% CI 1.38–1.44, p-value < 0.001) and general medical conditions (HR 1.65, 95% CI 1.62–1.67, p-value < 0.001) compared to no prescription among individuals with negative SARS-CoV-2 tests (Fig. 4 and Supplementary

**Fig. 1 | Hazard ratios and 95% confidence intervals of the association between SARS-CoV-2 status, age group, admission to hospital and mental disorders or general medical conditions. a** Positive SARS-CoV-2 test compared to negative SARS-CoV-2 test. **b** Admission to hospital with COVID-19 compared to no admission with COVID-19. HR hazard ratio, CI confidence interval, ICU intensive care unit. [a]The estimates are HRs with 95% CIs from Cox proportional hazards model stratified by age and adjusted for confounders (sex, Charlson Comorbidity Index (CCI), parental CCI, parental psychiatric history, employment status, income, and highest level of education). All statistical tests were two-sided without correction for multiple comparisons. [b]Results from ≤5 patients are displayed as "≤5" to ensure data privacy. [c]The reference group was individuals not admitted to the hospital with COVID-19, i.e., all individuals with negative or positive test results but no admission to hospital with COVID-19. The definition of admission to the hospital and ICU is summarized in Supplementary Table 30. [d]Including mental and behavioral disorders due to psychoactive substance use, schizophrenia spectrum, mood, and anxiety disorders. The diagnosis codes are summarized in Supplementary Table 27. Results for mental disorders are derived from a study population of n = 4,896,347 individuals. [e]Including selected neurological, respiratory, gastrointestinal, circulatory, kidney, endocrine, hematological, musculoskeletal, and dermatological disorders, as well as neuropsychiatric symptoms. The diagnosis codes are summarized in Supplementary Table 27. Results for general medical conditions are derived from a study population of n = 3,221,166 individuals.

Table 11). Individuals with positive SARS-CoV-2 tests did not have higher rates of mental disorders (HR 0.63, 95% CI 0.62–0.65, p-value < 0.001) and general medical conditions (HR 0.72, 95% CI 0.70–0.73, p-value < 0.001) compared with individuals with anti-infective prescriptions (and only negative SARS-CoV-2 tests). These findings were consistent across different types of anti-infectives and specific general medical conditions.

### Hospitalization with COVID-19 and rates of long-term mental disorders or general medical conditions

Compared with non-hospitalized individuals, compromising both those with positive and negative SARS-CoV-2 tests, hospitalized COVID-19 patients had elevated rates of mental disorders (HR 1.88, 95% CI 1.75–2.03, p-value < 0.001) and general medical conditions (HR 2.53, 95% CI 2.40–2.67, p-value < 0.001) (Table 1 and Supplementary Table 12). Compared with the non-hospitalized individuals, COVID-19-related ICU admission was associated with further elevated rates of mental disorders (HR 2.49, 95% CI 1.96–3.15, p-value < 0.001) and general medical conditions (HR 5.09, 95% CI 4.34–5.98, p-value < 0.001). Compared with the general population, all age groups showed higher rates of mental disorders and general medical conditions after COVID-19 admission, and highest among 60–79-year-olds (Fig. 1 and Supplementary Table 13). A dose-response relationship was observed between the rate of sequelae and number of admissions, duration of admission (Table 1, Supplementary Tables 14 and 15).

### Hospitalization with COVID-19 and time-varying effects

Compared to non-hospitalized individuals, the rates of mental disorders and general medical conditions for patients admitted with COVID-19 varied over the study period as illustrated by the time-varying hazard ratios in Fig. 2 (Supplementary Table 16). The effect of hospital admission with COVID-19 on sequelae were highest around the turn of the years 2020 and 2021.

### Hospitalization with COVID-19 and the rates of specific general medical conditions

Compared to the general population, rates of all specific general medical conditions, except musculoskeletal disorders, were higher among patients admitted with COVID-19 (Fig. 3 and Supplementary Table 12). The rates of respiratory disorders were particularly elevated (without ICU: HR 3.09, 95% CI 2.92–3.27, p-value < 0.001; with ICU: HR 7.77 (6.77–8.92, p-value < 0.001). The hazard ratios tended to decrease with increasing age (Supplementary Table 13 and Supplementary Fig. 3).

### Hospitalization with COVID-19 compared with other infectious diseases regarding rates of long-term sequelae

Compared with those not hospitalized with infections, hospital admissions with non-COVID-19 infections, particularly pulmonary infections, were associated with higher rates of mental disorders and general medical conditions (Fig. 4, Supplementary Tables 17 and 18). Individuals hospitalized with COVID-19 had similar rates of mental disorders as those hospitalized for non-COVID-19 infections (HR 1.03, 95% CI 0.94–1.13, p-value 0.517) and non-COVID-19 pulmonary infections (HR 0.87, 95% CI 0.79–0.96, p-value 0.005). However, the rates of general medical conditions were higher for COVID-19 compared with non-COVID-19 infections (HR 1.15, 95% CI 1.09–1.23, p-value < 0.001), but lower for COVID-19 compared with non-COVID-19 pulmonary infections (HR 0.70, 95% CI 0.65–0.75, p-value < 0.001) among individuals hospitalized with infections.

### Sensitivity analyses

For more information on sensitivity analyses, see Supplementary Materials.

### Number of SARS-CoV-2 tests and the rates of long-term sequelae

To investigate the impact of testing behavior, we further stratified by the number of conducted tests in the comparison between positive and negative SARS-CoV-2 PCR tests (Table 2 and Sensitivity analysis 2). For general medical conditions, the HRs for positive compared with negative tests remained close to non-significant, and only slightly increased with more tests conducted. However, testing behavior seemed to have a larger impact on the observed associations with mental disorders. Here, the HR increased with more tests conducted and to a non-clinically relevant difference for individuals with ≥15 SARS-CoV-2 tests, where the HR was 0.94 (95% CI 0.91–0.96).

### Type of SARS-CoV-2 virus and the rates of long-term sequelae

Compared to individuals with negative SARS-CoV-2 PCR tests, positive tests were not associated with higher rates of mental disorders for all SARS-CoV-2 variants, while the general medical condition rates were only elevated for the virus-types preceding Omicron (Original: HR 1.21, 95% CI 1.14–1.28; Alpha: HR 1.17, 95% CI 1.11–1.22, p-value < 0.001; Delta: HR 1.09, 95% CI 1.05–1.13, p-value < 0.001; Omicron: HR 0.98, 95% CI 0.96–1.01, Omicron subtypes: HR 1.02, 95% CI 1.01–1.04, p-value 0.001) (Fig. 2, Sensitivity analysis 3–4). Compared with the general population, the rates of mental disorders and general medical conditions were increased for all virus-types among individuals admitted to the hospital with COVID-19 (Fig. 2, Sensitivity analysis 8–9).

### Vaccinated vs non-vaccinated and the rates of long-term sequelae

Compared to individuals with negative SARS-CoV-2 PCR tests, positive tests were only associated with increased rates of general medical conditions among individuals with less than 3 vaccination doses (no vaccinations: HR 1.12, 95% CI 1.09–1.14, p-value < 0.001, 1 vaccination dose: HR 1.14, 95% CI 1.08–1.20, p-value < 0.001, 2 vaccination doses: HR 1.05, 95% CI 1.03–1.07, p-value < 0.001, 3 or more vaccination doses: HR 0.99, 95% CI 0.98–1.01, p-value 0.389) (Sensitivity analysis 5). Compared with the general population, the rates of mental disorders and general medical conditions were increased for patients admitted to hospital with COVID-19 irrespective of number of vaccinations (Sensitivity analysis 10).

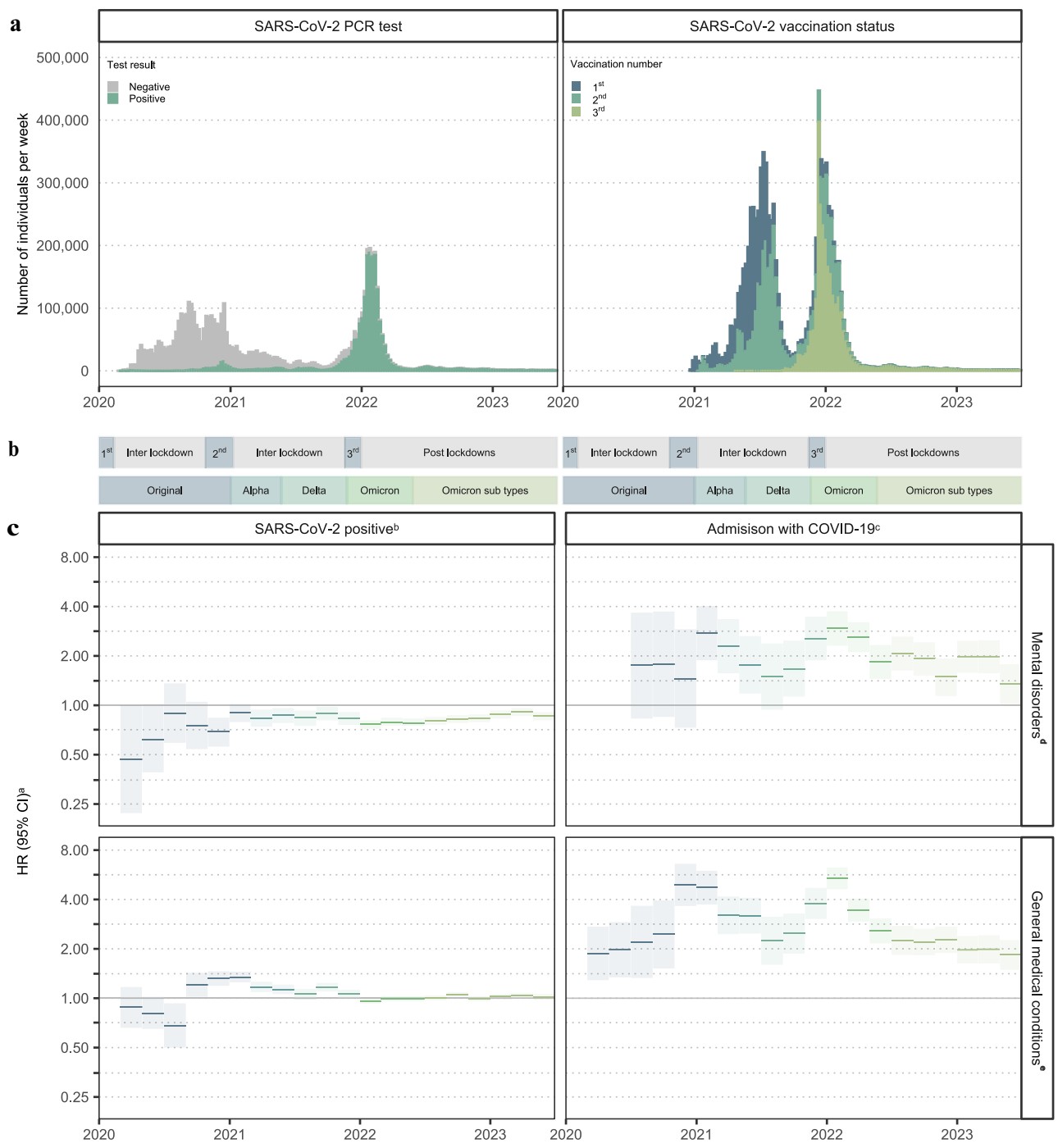

**Fig. 2 | Timeline of the COVID-19 pandemic, number of individuals tested and vaccinated.** Hazard ratios and 95% confidence intervals of SARS-CoV-2 status, calendar period, admission status and mental disorders or general medical conditions. **a** Number of individuals per week with a SARS-CoV-2 test and number of individuals per week vaccinated against SARS-CoV-2. **b** Timeline of lockdown and inter-lockdown periods and dominating SARS-CoV-2 type. **c** Hazard ratios and 95% confidence intervals of SARS-CoV-2 status, calendar time in 2-month intervals, admission status and mental disorders or general medical conditions. Results from ≤5 patients are censored to ensure data privacy. HR, hazard ratio; CI, confidence interval; 1st, 1st lockdown; 2nd, 2nd lockdown; 3rd, 3rd lockdown. [a]The estimates are HRs with 95% CIs from Cox proportional hazards model stratified by age and adjusted for confounders (sex, Charlson Comorbidity Index (CCI), parental CCI, parental psychiatric history, employment status, income, and highest level of education). The underlying time scale, calendar time, was divided into 2-month intervals. We compared the exposure to the reference within the different calendar periods. [b]The exposure was individuals with positive SARS-CoV-2 tests and the reference group was individuals with negative SARS-CoV-2 tests. [c]The exposure was individuals admitted with COVID-19 and the reference group was individuals not admitted to the hospital with COVID-19, i.e., individuals with negative or positive test results but no admission to hospital with COVID-19. The definition of admission to the hospital is summarized in Supplementary Table 30. [d]Including mental and behavioral disorders due to psychoactive substance use, schizophrenia spectrum, mood, and anxiety disorders. The diagnosis codes are summarized in Supplementary Table 27. Results for mental disorders are derived from a study population of $n = 4{,}896{,}347$ individuals. [e]Including selected neurological, respiratory, gastrointestinal, circulatory, kidney, endocrine, hematological, musculoskeletal, and dermatological disorders, as well as neuropsychiatric symptoms. The diagnosis codes are summarized in Supplementary Table 27. Results for general medical conditions are derived from a study population of $n = 3{,}221{,}166$ individuals.

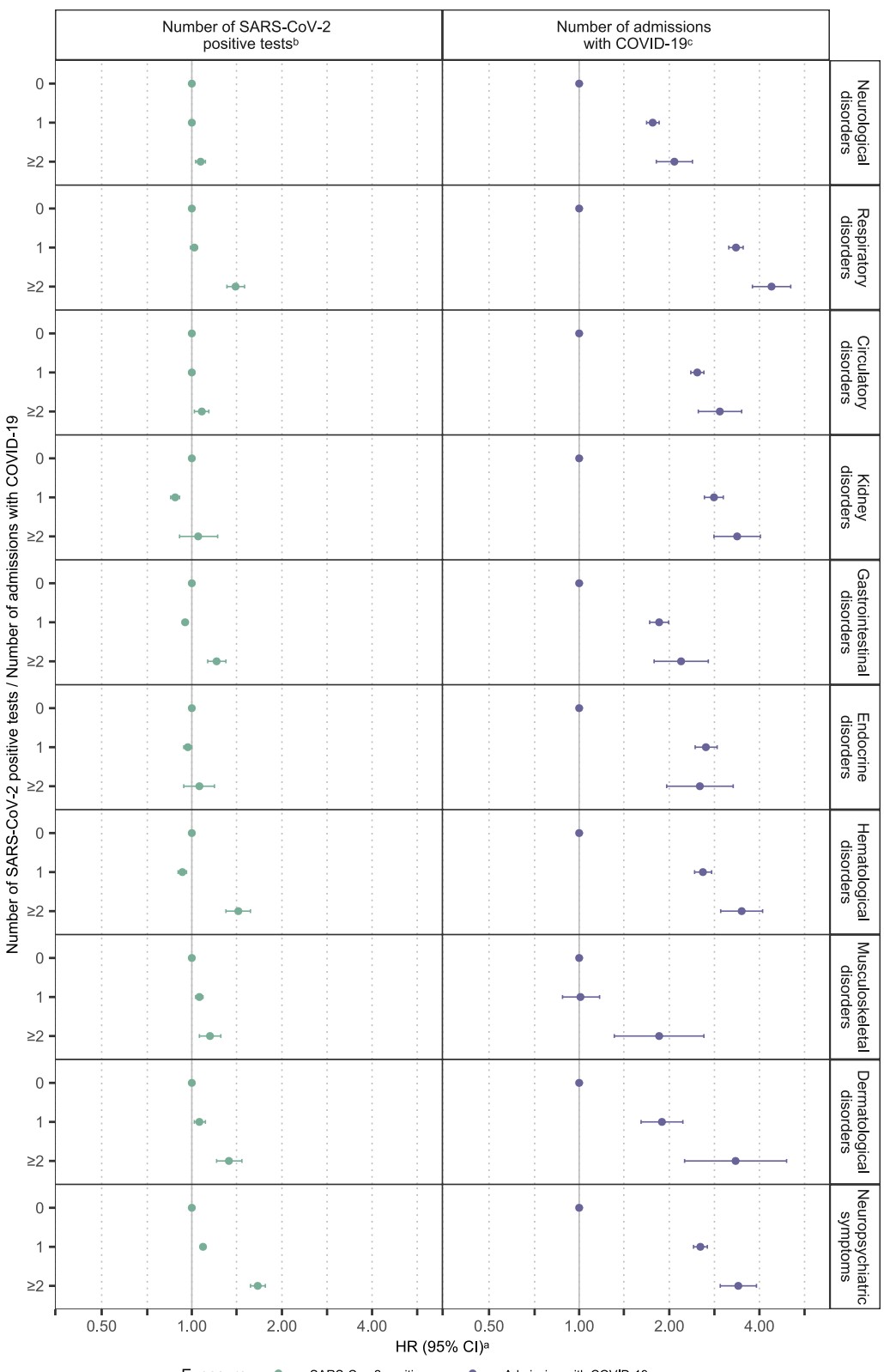

### CRP levels and the rates of long-term sequelae

Compared with SARS-CoV-2 negative individuals with peak CRP ≤ 4 mg/L, peak CRP ≥ 100 mg/L was associated with higher general medical condition rates among individuals with positive (HR 1.43, 95% CI 1.35–1.51, p-value < 0.001) and negative (HR 1.41, 95% CI 1.36–1.46) SARS-CoV-2 tests (Supplementary Tables 19 and 20 and Supplementary Fig. 4). Among individuals with positive SARS-CoV-2

tests, there was no significant association between CRP levels and mental disorder rates and increasing CRP values were only associated with increasing HR of general medical conditions for the age groups between 40 and 79 years (Supplementary Tables 21 and 22, Supplementary Figs. 5 and 6). Among individuals hospitalized with COVID-19, there was no significant association between CRP levels and rates of long-term mental disorders or general medical

**Fig. 3 | Hazard ratios and 95% confidence intervals of the association between the number of SARS-CoV-2 infections, admission to hospital and specific general medical conditions.** The specific general medical conditions are neurological ($n = 4,323,502$), respiratory ($n = 4,915,770$), circulatory ($n = 4,531,873$), kidney ($n = 5,315,598$), gastrointestinal ($n = 4,943,199$), endocrine ($n = 5,193,087$), hematological ($n = 5,249,230$), musculoskeletal ($n = 5,230,816$), dermatological disorders ($n = 5,272,108$), and neuropsychiatric symptoms ($n = 5,125,359$). Source data are provided as a Source Data file. HR, hazard ratio; CI, confidence interval.

[a]The estimates are HRs with 95% CIs from Cox proportional hazards model stratified by age and adjusted for confounders (sex, Charlson Comorbidity Index (CCI), parental CCI, parental psychiatric history, employment status, income, and highest level of education). The results were not adjusted for multiple testing. [b]The reference group was individuals with negative SARS-CoV-2 tests. [c]The reference group was individuals not admitted to the hospital with COVID-19, i.e., all individuals with negative or positive test results but no admission to hospital with COVID-19. The definition of admission to the hospital is summarized in Supplementary Table 30.

| | Mental disorders[b] | | | | General medical conditions[c] | | | |
|---|---|---|---|---|---|---|---|---|
| | Cases, No. | HR (95% CI)[a] | p-value | | Cases, No. | HR (95% CI)[a] | p-value | |
| **Prescription for anti-infective agents[d]** | | | | | | | | |
| No infection | 32,782 | 1.00 [reference] | .. | | 85,560 | 1.00 [reference] | .. | |
| Any prescription for anti-infective agents | 8,951 | 1.41 (1.38 - 1.44) | <0.001 | | 22,209 | 1.65 (1.62 - 1.67) | <0.001 | |
| SARS-CoV-2 positive | 18,394 | 0.89 (0.87 - 0.91) | <0.001 | | 48,231 | 1.18 (1.16 - 1.19) | <0.001 | |
| SARS-CoV-2 positive vs. prescription | .. | 0.63 (0.62 - 0.65) | <0.001 | | .. | 0.72 (0.70 - 0.73) | <0.001 | |
| **Admission with non-COVID-19 infections[e]** | | | | | | | | |
| No admission | 80,928 | 1.00 [reference] | .. | | 207,045 | 1.00 [reference] | .. | |
| Any non-COVID infection | 2,779 | 2.03 (1.95 - 2.11) | <0.001 | | 5,980 | 2.41 (2.35 - 2.48) | <0.001 | |
| COVID-19-admission | 567 | 2.09 (1.93 - 2.27) | <0.001 | | 1,331 | 2.79 (2.64 - 2.94) | <0.001 | |
| COVID-19 vs. other infection | .. | 1.03 (0.94 - 1.13) | 0.517 | | .. | 1.15 (1.09 - 1.23) | <0.001 | |
| **Admission with non-COVID-19 pulmonary infections[e]** | | | | | | | | |
| No admission | 94,197 | 1.00 [reference] | .. | | 231,699 | 1.00 [reference] | .. | |
| Any non-COVID pulmonary infection | 913 | 2.32 (2.17 - 2.48) | <0.001 | | 1,919 | 3.84 (3.67 - 4.02) | <0.001 | |
| COVID-19-admission | 722 | 2.02 (1.87 - 2.17) | <0.001 | | 1,479 | 2.68 (2.55 - 2.83) | <0.001 | |
| COVID-19 vs. other pulmonary infection | .. | 0.87 (0.79 - 0.96) | 0.005 | | .. | 0.70 (0.65 - 0.75) | <0.001 | |

0.50 1.00 2.00 0.50 1.00 2.00
HR (95% CI) HR (95% CI)

Exposure —●— SARS-Cov-2 positive —●— Admission with COVID-19

**Fig. 4 | Hazard ratios and 95% confidence intervals of the association between SARS-CoV-2 test, prescriptions for anti-infective agents, admission with COVID-19, admissions with non-COVID-19 (pulmonary) infections and mental disorders or general medical conditions.** HR, hazard ratio; CI, confidence interval. [a]The estimates are HRs with 95% CIs from Cox proportional hazards model stratified by age and adjusted for confounders (sex, Charlson Comorbidity Index (CCI), parental CCI, parental mental health disorders, employment status, income, highest level of education). All statistical tests were two-sided without correction for multiple comparisons. [b]Including mental and behavioral disorders due to psychoactive substance use, schizophrenia spectrum, mood, and anxiety disorders. The diagnosis codes are summarized in Supplementary Table 27. Results for mental disorders are derived from a study population of $n = 4,896,347$ individuals. [c]Including selected neurological, respiratory, gastrointestinal, circulatory, kidney, endocrine, hematological, musculoskeletal, and dermatological disorders, as well as neuropsychiatric symptoms. The diagnosis codes are summarized in Supplementary Table 27. Results for general medical conditions are derived from a study population of $n = 3,221,166$ individuals. [d]Excluding individuals who redeemed a prescription for any anti-infective agent within a year before the study start to rule

out recurring infections. The ATC codes for anti-infective agents are summarized in Supplementary Table 31. The reference group, *"no infection"*, included all individuals without a prescription for an anti-infective agent and with negative test results. The exposure group, *"any prescription for anti-infective agents"*, comprised individuals with a prescription for an anti-infective agent and negative SARS-CoV-2 test results. The exposure group, *"SARS-CoV-2 positive"*, included all individuals with a positive SARS-CoV-2 test result, regardless of whether they had prescriptions for anti-infective agents. [e]Excluding individuals who had been admitted for non-COVID-19 (pulmonary) infections within ten years before the study start to rule out recurring infections. The definition of hospital admission is summarized in Supplementary Table 30, and the ICD-10 codes for infections are listed in Supplementary Table 32. The reference group, *"no admission"*, included individuals without hospital admission with COVID-19 or any other infection. The exposure group, *"any non-COVID-19 (pulmonary) infection"*, comprised individuals admitted to the hospital with a non-COVID-19 (pulmonary) infection. The exposure group, *"COVID-19 admission"*, included individuals admitted to the hospital with COVID-19, regardless of whether they had also been admitted to the hospital with a non-COVID-19 (pulmonary) infection.

conditions (Supplementary Tables 23 and 26, Supplementary Figs. 7 and 8).

### Analyses with five years washout period for prior hospital contacts

Post hoc, instead of excluding individuals with hospital contacts for pre-existing disorders at any point in time before study start, we conducted additional analyses only excluded individuals with a hospital contact within five years before study start (Sensitivity analysis 7 and 12). This excludes 215,506 and 1,298,288 individuals with hospital

contacts for mental disorders and general medical conditions, respectively. The results were generally unchanged.

### Discussion

Individuals with positive and individuals with negative SARS-CoV-2 tests had higher rates of mental disorders and general medical conditions than individuals not tested. The rates of mental disorders and general medical conditions increased with reinfections and disease severity as indicated by COVID-19 admission status and duration. However, when comparing individuals tested positive for SARS-CoV-2

**Table 2 | Hazard ratios and 95% confidence intervals of the association between the number of SARS-CoV-2 tests and mental disorders or general medical conditions**

| Number of SARS-CoV-2 tests | Mental disorders[a] | | | General medical conditions[b] | | |
|---|---|---|---|---|---|---|
| | Cases, No. | HR (95% CI)[c] | p-value | Cases, No. | HR (95% CI)[c] | p-value |
| **0–3 tests** | | | | | | |
| Negative | 33,327 | 1.00 [reference] | – | 82,055 | 1.00 [reference] | – |
| Positive | 3605 | 0.76 (0.74–0.79) | <0.001 | 10,471 | 0.97 (0.95–0.99) | 0.006 |
| **4–9 tests** | | | | | | |
| Negative | 22,701 | 1.00 [reference] | – | 49,151 | 1.00 [reference] | – |
| Positive | 10,888 | 0.80 (0.78–0.82) | <0.001 | 25,191 | 0.97 (0.95–0.98) | <0.001 |
| **10–14 tests** | | | | | | |
| Negative | 6724 | 1.00 [reference] | – | 15,413 | 1.00 [reference] | – |
| Positive | 6106 | 0.87 (0.84–0.91) | <0.001 | 13,874 | 1.03 (1.01–1.06) | 0.006 |
| **≥15 tests** | | | | | | |
| Negative | 7886 | 1.00 [reference] | – | 21,710 | 1.00 [reference] | – |
| Positive | 9693 | 0.94 (0.91–0.96) | <0.001 | 25,102 | 1.05 (1.03–1.07) | <0.001 |
| p-value of effect modifier | – | – | <0.001 | – | – | <0.001 |

HR hazard ratio, CI confidence interval.

[a]Including mental and behavioral disorders due to psychoactive substance use, schizophrenia spectrum, mood, and anxiety disorders. The diagnosis codes are summarized in Supplementary Table 27. Results for mental disorders are derived from a study population of n = 4,896,347 individuals.

[b]Including selected neurological, respiratory, gastrointestinal, circulatory, kidney, endocrine, hematological, musculoskeletal, and dermatological disorders, as well as neuropsychiatric symptoms. The diagnosis codes are summarized in Supplementary Table 27. Results for general medical conditions are derived from a study population of n = 3,221,166 individuals.

[c]The estimates are HRs with 95% CIs from Cox proportional hazards model stratified by age and adjusted for confounders (sex, Charlson Comorbidity Index (CCI), parental CCI, parental mental health disorders, employment status, income, highest level of education). The results for positive test results were compared to negative test results within each group.

tests with individuals tested negative for SARS-CoV-2 tests, the rates of mental disorders were not increased, and the associations with general medical conditions displayed only negligible effects for virus types preceding Omicron and for individuals with less than 3 vaccinations, with no overall major biological nor public health impact. Rates of mental disorders and general medical conditions were comparably increased for individuals hospitalized for COVID-19 and individuals hospitalized for other infections.

Prior survey-based studies examining self-reported post-COVID-19 symptoms included individuals with PCR tests taken outside hospital context and found higher prevalence of neuropsychiatric symptoms such as dizziness, painful muscles, general tiredness/fatigue, ageusia/anosmia/dysgeusia, and memory problems after positive compared to negative tests[3,22,23]. However, most prior studies with clinical outcomes included only individuals with hospital contacts, for example PCR tests from outpatient laboratories[4,5,24], even in studies assessing sequelae among non-hospitalized individuals. Thereby previous studies excluded the milder and asymptomatic cases with SARS-CoV-2. Further, the self-reported survey-based studies are subject to reporter bias, while we were able to include all individuals PCR tested for SARS-CoV-2 in all of Denmark. We corroborated the findings of another Danish study of non-hospitalized individuals that compared positive to negative SARS-CoV-2 PCR tests and found that after 6 months the absolute risk post-COVID of severe post-acute complications was low[15]. Our analyses showed that the hazard ratios of sequelae varied over time and that, in the current situation with widespread vaccination and dominant Omicron subtypes, COVID-19 poses a lower risk of post-acute sequelae than in the beginning of the pandemic. This could partly explain why prior studies that were conducted earlier in the pandemic estimated higher risks than in our study. Moreover, the different timeframes and heterogeneous definitions of long COVID symptoms in prior studies might blur the understanding of COVID-19 impact and post-acute sequelae development[20,25].

This study did not find post-infection sequelae to be specific for COVID-19, as the rates of mental disorders or general medical conditions were similarly elevated after COVID-19 compared to non-COVID-19 infections. Individuals with prescriptions for anti-infective agents had higher rates of sequelae than those tested positive for SARS-CoV-2, reflecting that anti-infective use may reflect more severe acute illness. Acknowledging that the risk of post-acute sequalae associated with COVID-19 is similar to that of other infections is important for prevention of unnecessary anxiety in the public perception. Accurate and balanced risk assessments are important for broadening our understanding of post-infection sequelae in general and for resource allocation in the health care sector, where our study strongly suggests that resources towards sequelae after severe infections must be balanced more broadly. The severity of the infection, and to some degree the levels of inflammatory response induced by the infection as measured by CRP levels in outpatients, was one of the main drivers for increased risks of post-infection sequalae. Although we did not find associations between CRP levels and the rates of sequelae, among individuals hospitalized with COVID-19 or other infections, we did find that severity measures such as requiring ICU treatment were associated with increased risk of post-infection sequelae.

In Denmark, PCR tests were available from the beginning of the pandemic and were predominantly used until they were gradually phased out during spring and summer 2022[26]. In May 2021, a nationwide *corona passport* was introduced to facilitate a safe reopening of society, requiring either a recent negative SARS-CoV-2 test or full vaccination for access to social and educational activities. Notably, there was equal access to testing facilities[27] and reduced economic barriers in Denmark compared to other countries, may have led to higher detection rates of mild cases in our study than in studies from other countries. The decreased rates of mental disorders observed among individuals with positive compared to negative tests could have several explanations, especially since the effect diminished to a non-clinically significant level among individuals tested 15 times or more. First, in the clinical treatment of patients with COVID-19, examination for general medical conditions may be prioritized before mental disorders were considered. Second, the rates of mental disorders

depended on the number of tests conducted, suggesting an unexplained dependency on the testing behavior in the evaluation of mental health outcomes. Third, access to social activities increases the risk of contracting COVID-19 and strengthens social relationships, which is known to support mental health[28]. Finally, an undiagnosed mental disorder might influence the incentive to get tested.

## Strengths and limitations

The strengths of this study were firstly, the use of the nationwide and validated Danish registers that enabled analyses on an entire population with complete information on exposure (PCR tests, hospitalization), outcomes (diagnoses in an inpatient or outpatient setting, including emergency room visits, although not from primary care), and confounders on the individual level (including sex, age, comorbidities, socioeconomics, and likewise for parents)[29–38]. Secondly, we analyzed an extensive list of pre-specified endpoints on mental disorders and general medical conditions during acute COVID-19 and up to 40 months post COVID-19. Thirdly, by comparing multiple infectious diseases, the study highlights differences and similarities between COVID-19 and other infections. Lastly, SARS-CoV-2 PCR testing during the COVID-19 pandemic was used by more than 92% of the population and enabled appropriate control groups and low misclassification bias. The numbers and results of PCR tests and admissions were consistent with the official Danish numbers[39].

The limitations of this study include first, that potential surveillance bias occurred due to attention on sequelae among COVID-19 survivors compared with other infectious diseases. Second, we could only include outcome diagnoses made in inpatient or outpatient settings, including emergency room visits, since information on sequelae treated in primary care or symptoms not requiring professional care was not available in the Danish registers. Therefore, we might only capture the more severe end of the spectrum of sequalae that required inpatient or outpatient treatment, while differential effects of predominantly mild sequelae of COVID-19 or other infectious diseases were not detected by our analyses. Third, the 40-month follow-up captured most long-term risks, but not slowly progressing diseases. Fourth, despite biases towards specific demographics and high false negative rates (symptomatic COVID-19 patients testing negative and being included in the control group), PCR tests were the most reliable SARS-CoV-2 detection method[40,41]. Fifth, the study population consisted of predominantly healthy individuals, as we excluded anyone previously diagnosed with any of the outcome disorders at any point since the start of the registers (1968 for mental disorders and 1977 for general medical conditions) to ensure analysis of first-time outcome diagnoses. However, a post hoc sensitivity analysis using a washout period of only 5 years for prior conditions did not change the results, indicating that healthy user bias likely had minimal impact on our findings. Sixth, testing behavior, such as incentives to get a SARS-CoV-2 test, were not captured in the registers and therefore not included in the analyses, although we demonstrated that testing behavior, as captured by the number of SARS-CoV-2 tests, did impact the associations with mental disorders. Unobservable factors may also have influenced testing behavior and test results, and these influences may have varied over the course of the pandemic. Last, testing behavior, socioeconomic and demographic factors likely differ between countries, affecting the psychosocial aspects of post-COVID-19 disorders, but the biological effects of SARS-CoV-2 infection on the subsequent risk of sequelae are expected to be largely comparable across countries.

In conclusion, COVID-19 infection requiring hospitalization was associated with increased rates of sequelae compared to individuals not hospitalized. However, the rates of sequelae after COVID-19 requiring hospitalization were comparable to those observed after hospitalization for non-COVID-19 infections. Moreover, compared with negative SARS-CoV-2 tests, individuals with a positive SARS-CoV-2 test did not display rates of sequelae that were elevated to a clinically relevant extent. Rates of general medical conditions after a positive compared with a negative SARS-CoV-2 PCR test were only elevated for virus-types preceding Omicron and for individuals with less than 3 vaccinations. Thus, sequelae are not exclusive to COVID-19 survivors; they also occur after other infectious diseases of similar severity like influenza and bacterial pneumonia. These unique COVID-19 data broaden our understanding of post-infectious sequelae and highlights the need to investigate preventive and treatment measurements more broadly than only in a COVID-19 perspective.

## Methods

This study was pre-registered on the Open Science Framework (osf.io/9qgjt). This study was approved by the Danish Data Protection Agency and the Danish Health and Medicine Authority. Following the implementation of the General Data Protection Regulation (EU) 2016/679 in Denmark on 23 May 2018, register-based studies are no longer required to be reported to or approved by the Danish Data Protection Authority and do not require additional ethical approval or informed consent.

### Study population

We utilized nationwide registers covering the entire population of Denmark as of March 1, 2020 (5.8 million inhabitants). The registers provided data on each resident, including sex, age, socioeconomic status, hospital contacts, and prescriptions. Data from the different Danish registers were linked using the unique personal identification number in the Danish Civil Registration System, which also enabled linkage to parents[42]. Individuals were followed from March 1, 2020, until the onset of the disorders of interest or censored in case of death, emigration, or end of follow-up on June 30, 2023. For the primary analysis, we included all individuals, including those not tested, and considered March 1, 2020, the start of follow-up. For other analyses, only individuals with a SARS-CoV-2 PCR test were included, with the first test date as the start of follow-up.

### Exposure to SARS-CoV-2

*Confirmed COVID-19* was defined by a positive SARS-CoV-2 polymerase chain reaction (PCR) by nasopharyngeal/tracheal test. Data were available on all SARS-CoV-2 test results performed in official testing facilities in Denmark from February 2, 2020, through the Microbiology Database (MiBa)[43]. Access to health care is free in Denmark, and during the pandemic there was free and equal access to testing facilities[27]. PCR tests were the best method for detecting SARS-CoV-2[40], and we did not include results from other types of tests such as self-testing, as a positive self-test required subsequent confirmation with PCR testing.

### Assessment of outcomes

Diagnoses from inpatient, outpatient, or emergency room contacts were collected from the Danish Psychiatric Central Research Register and the Danish National Patient Register[38,44]. The registers were coded using the *International Classification of Diseases, Eighth Revision (ICD-8)* from January 1969 and *Tenth Revision (ICD-10)* from January 1994. Patients with pre-existing diagnoses were excluded within each outcome diagnosis. For example, when investigating kidney disorders, patients with kidney disorders before the start of follow-up were excluded, but not patients with psychiatric diagnoses such as PTSD (see Supplementary Table 1 for number of individuals, and Supplementary Table 27 for diagnosis codes).

**Primary outcome.** Any first-time mental disorder (including mental and behavioral disorders due to psychoactive substance use, psychotic, mood, and anxiety disorders) or general medical condition (including selected neurological, respiratory, gastrointestinal, circulatory, kidney, endocrine, hematological, musculoskeletal, and

dermatological disorders, as well as neuropsychiatric symptoms) (Supplementary Table 27). The pre-specified outcomes were selected based on our prior work on neurological and psychiatric sequelae of COVID-19[16,18], and from the prior literature of identified associations[4–7,15,19,45–54]. The date of the first hospital contact defined illness onset.

**Secondary outcomes.** A first diagnosis within the following specific categories: mental, neurological, respiratory, gastrointestinal, circulatory, kidney, endocrine, hematological, musculoskeletal, and dermatological disorders, and neuropsychiatric symptoms (Supplementary Table 27)[5,16,18,55–57].

### Statistical analysis
We performed Cox proportional hazards regression, stratified by age and with calendar time as the underlying time scale to capture the most complex effects during the pandemic, for example virus variants, lockdowns, and vaccination recommendations. We reported hazard ratios (HRs) with 95% confidence intervals (CIs). Statistical analyses were conducted with the *survival* package in R, version 4.4.1, with significance level set at a two-sided *p*-value < 0.05 (see Supplementary Methods Section for more details). The analyses were adjusted for confounders related to COVID-19 and sequelae, and the confounders were defined at the start of follow-up: Sex, socioeconomic status (employment status, income quantile, and educational attainment level)[33,58], medical susceptibility to disease (Charlson Comorbidity Index (CCI) (Supplementary Table 28))[59], parental medical history (parental CCI, and parental psychiatric diagnosis) (see Supplementary Table 29 for confounder definitions). Missing values were managed by conducting complete case analyses (missing values 0.2% due to unknown employment status and income).

*The primary analyses* compared individuals tested positive for SARS-CoV-2 with individuals not tested, or tested negative for SARS-CoV-2. We identified the test results in a hierarchal time-varying manner, where initially everyone was in the no test group and when tested moved to the negative or positive group. Subsequent negative tests were ignored in the positive group. With this definition, the same individuals could not be in different groups at the same time.

*In the secondary analyses*: we investigated how the COVID-19 sequelae were affected by disease severity, measured by (i) COVID-19-reinfections, (ii) hospital admission, (iii) duration of hospital admission, (iv) number of admissions, (v) intensive care unit (ICU) admission, and (vi) peak CRP levels (see Supplementary Table 30 and Supplementary Methods Section for more details on exposure definitions). CRP was collected from the Register of Laboratory Results for Research covering the large clinical biochemical and immunological laboratories[60]. When considering SARS-CoV-2 PCR tests, CRP measurements taken within two weeks (14 days) prior to the test date were included in the analysis, and in the case of admissions with COVID-19, only CRP measurements taken within two days before or during the admission were considered. We assessed the trajectory of the illness by measuring outcomes based on time after infection: <1 month, 1–2 months, 3–5 months, 6–11 months, and more than 12 months. We investigated time-varying effects by dividing calendar time into 2-month intervals and using stratified analyses to allow different baseline hazards and different hazard ratios in each interval. We compared COVID-19 to non-COVID-19 infections separately for hospital and non-hospital-treated infections (see Supplementary Methods Section for more details on exposure definitions). For non-hospital-treated infections, SARS-CoV-2 positive tests were compared with, (i) any anti-infective agent prescription, and (ii) subtypes of anti-infectives (antibacterial, antiviral, and antimycotic agents) (see Supplementary Table 31 for ATC codes). All individuals with a positive SARS-CoV-2 test result were assigned to the positive group, regardless of whether they had prescriptions for anti-infective agents.

Prescriptions with anti-infective agents were collected from the Danish National Prescription Register, which contained information on all redeemed prescriptions since 1995 grouped by ATC codes[61]. For hospital-treated infections, COVID-19-admissions were compared with: (i) any infection treated in hospital, (ii) any pulmonary infection treated in hospital, and (iii) subtypes of pulmonary infections (influenza, bacterial pneumonia, and other pulmonary infections) (see Supplementary Table 32 for diagnosis codes). Admissions with COVID-19 and non-COVID-19 infections were identified using the Danish National Patient Register. All individuals admitted to the hospital with COVID-19 were included in the COVID-19 admission group, regardless of whether they were also admitted for other infectious diseases.

### Sensitivity analysis
In sensitivity analyses, the robustness of results was assessed by repeating the analyses with various adjustments for confounders: (i) no adjustment, (ii) stratification for age in 10-year bands, (iii) stratified for age and adjusted for sex, iv) stratified for age, adjusted for sex, CCI, parental CCI, parental mental health, employment status, income, the highest level of education. Given the changes in SARS-CoV-2 contagiousness, testing behavior, and vaccination uptake throughout the pandemic, we also examined (i) the number of conducted tests, (ii) dominating SARS-CoV-2 variants, (iii) lockdown periods, (iv) vaccination status, and (v) immigration status at the individual level (Supplementary Tables 33 and 34 and Supplementary Fig. 9). Vaccination data was extracted from the Danish Vaccination Register[62]. Post hoc, we conducted an additional sensitivity analysis to investigate how exclusion criterion affected results by only excluding individuals with a hospital contact for a pre-existing diagnosis 5 years before the start of the study.

### Reporting summary
Further information on research design is available in the Nature Portfolio Reporting Summary linked to this article.

## Data availability
We used the following registers: the Microbiology Database, the Danish National Patient Register, the Danish Psychiatric Central Research Register, the Population Education Register, the Income Statistics Register, and the Danish National Prescription Register. The data are available from Statistics Denmark, https://www.dst.dk/en/TilSalg/Forskningsservice/Dataadgang. To access health data in Denmark, applications must be submitted to the Danish Data Protection Agency, the Danish National Board of Health, and Statistics Denmark. For more information, see https://www.itgovernance.eu/da-dk/eu-gdpr-compliance-dk, https://sundhedsdatastyrelsen.dk/da/english, and https://dst.dk/en. Source data are provided with this paper.

## Code availability
Code available at: https://github.com/Biological-and-Precision-Psychiatry/COVID-19-post-infection-sequelae[63]. The managing of data, analyses, and illustrations were made in R version 4.4.1. Analyses using the Cox proportional hazards model were performed with the *coxph* function from the *survival* package version 3.7-0. Additional packages used (version): *data.table* (1.16.2), *dplyr* (1.1.4), *lubridate* (1.9.3), *foreach* (1.5.2), *doParallel* (1.0.17), and *emmeans* (1.10.2).

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

## Acknowledgements

We thank Elisabeth Anne Wreford Andersen for invaluable input for the pre-registration of this study and the data analysis. The research was funded by an unrestricted grant from the Novo Nordisk Foundation (grant number NNF21OC0067769) (M.E.B. and D.K.) and the Lundbeck Foundation (grant number 349-2020-658) (M.E.B. and D.K.). The funding sources had no role in the design, data, analysis, interpretation, writing of the article, or decision to publish the findings.

## Author contributions

M.E.B. and D.K. received funding for the study. All authors participated in the design of the study. C.S.G. and R.H.B.C. verified the underlying data. C.S.G. conducted the statistical analysis, wrote the first draft of the manuscript, and all authors contributed to editing and approving the final version.

## Competing interests

The authors declare no competing interests.
