## [Transparent Peer Review file · Nature Communications]

Post-infection sequelae of COVID-19 and other infectious diseases – a nationwide Danish study with 40-month follow-up

Corresponding Author: Professor Michael Benros

Version 0:

Reviewer comments:

Reviewer #1

(Remarks to the Author)

Using the nationwide Danish registers, this large study (3,343,706 individuals) used SARS-CoV-2 PCR test results and COVID-19 hospitalizations from March 2020 to June 2023, and estimated Hazard ratios of mental, neurological, and general medical conditions, as well as of neuropsychiatric symptoms. The authors found that compared with negative tests, positive SARS-CoV-2 PCR tests were associated with lower risk of mental disorders and minimally higher rates of general medical conditions. This large study is well conducted and an important contribution to the research on long-COVID symptoms but needs major revisions. Please see my comments below:

Comment 1: Methods: exposure to SARS-CoV-2 paragraph; did the study include only tests from the registry MiBa? What about self-testing? Please add a sentence that you did not use self-tests and how this could have influenced the results, for example selection bias.

Comment 2: Study population-flow chart: only those with PCR-test? Only self-test after February 2022 in Norway, the same in Denmark? Low number of participants with PCR-test after omicron? Who are these participants with PCR-test after Omicron? Selection bias.

Comment 3: Too short discussion, half a page... extend the discussion. There are several other studies who have included non-hospitalized and tested for SARS-CoV-2 (Ballering, van Zon et al. 2022, Ellingjord-Dale, Brunvoll et al. 2024, Ellingjord-Dale, Nygaard et al. 2024),

Ballering, A. V., S. K. R. van Zon, T. C. Olde Hartman and J. G. M. Rosmalen (2022). "Persistence of somatic symptoms after COVID-19 in the Netherlands: an observational cohort study." *Lancet* 400(10350): 452-461.

Ellingjord-Dale, M., S. H. Brunvoll and A. Soraas (2024). "Prospective Memory Assessment before and after Covid-19." *N Engl J Med* 390(9): 863-865.

Ellingjord-Dale, M., A. B. Nygaard, N. C. Støer, R. Bø, N. I. Landrø, S. H. Brunvoll, M. Istre, K. T. Kalleberg, J. A. Dahl, L. Geng, K. Tsilidis, E. Riboli, G. Ursin and A. Søråas (2024). "Temporal trajectories of long-COVID symptoms in adults with 22 months follow-up in a prospective cohort study in Norway." *Int J Infect Dis* 149: 107263.

Comment 4: Line 204. "hazard ratio of general medical conditions increased with CRP in individuals with positive SARS CoV-2 tests, though not in patients hospitalized with COVID-19". Please discuss in the discussion section why you did not find any difference? Who was the comparison group? If hospitalized they have already a high CRP?

Comment 5: line 240, please define pre-existing outcome disorders.

Comment 6: line 239-242: 2.5 million people were excluded from the study having pre-existing outcome disorder. Please add a sentence or two on how this large exclusion of people could have affected the results, under- or overestimation?

Healthy bias effect as limitation. Selection bias etc.

Comment 7: I could not find any sensitivity-analysis on the 2.5 million excluded from the study?
Please, do a sensitivity analysis including these large group of people.

Comment 8: line 241-242- Please rephrase this sentence. Could the authors be more specific than stating that “some effect estimates were not...an unselected population”? “Some” effect estimates? Which ones? In my opinion, this goes for the whole study population and all the effects. By “unselected” I assume you mean the general population in Denmark? Please rephrase.

Comment 9: Conclusion, line 245-46: Unprecise fist sentence-. “Hospitalized COVID-19 individuals have higher rates of sequelae, whereas non-hospitalized COVID-19 individuals do not” Compared to who? Please be more specific.

Comment 10: line 245-246, “Hospitalized COVID-19 individuals have higher rates of sequelae, whereas non-hospitalized COVID-19 individuals do not” Higher rates, should it not be “risk of sequelae” and not “rates” since HR is a risk-estimate? Please be more specific.

Comment 11: line 246-47. “Sequelae are not exclusive to COVID-19 survivors but also occur after other infectious diseases with similar severity”. What do the authors mean “by similar severity”? Hospitalized? Severity according to what? Please me more specific.

Comment 12: line 247-48, “However, the unique data on COVID-19, can help ...of infections”. I assume that the authors mean” COVID-19 infection compared to other infections (SARS-CoV-2-negative). Please be more precise.

Comment 13: line 239-241. Excluding pre-existing outcome disorders? How many? What do the authors mean by pre-existing outcome disorders? Please be more specific and define.

Comment 14: Line 260-Methods: “Individuals were followed from March 1, 2020, until the onset of the disorders of interest and censored in case of death, emigration...” I assume that it should say ;...or censored in case of death, emigration..”

Comment 15: line 279-280. Please be more precise in the definition of primary outcomes: what kind of mental disorder, please give examples? The same goes for general medical conditions; please specify.

Comment 16: line 292-93, Statistical analyses. This sentence is confusing “This method has been proven to minimize bias when estimating vaccination effectiveness”. Do the study measure vaccination effectiveness as the main objective? And cox regression is widely used as a survival analysis, time to event, risk analysis etc., so it is a bit misleading to state that it is known for minimizing bias in vaccination effectiveness. Please remove this sentence.

Comment 17: Line 293-94; the authors mention that they reported Hazard rate ratios (HR), shouldn't it be hazard ratio? The effect estimate should be referred to as hazard ratio throughout the manuscript.

Comment 18: line 296-297 “The analyses were adjusted for the following confounders related to COVID-19 and sequelae at start of follow-up”. Confusing sentence: what are the exposures and what is the outcome in the analyses?
What do the authors mean by confounders related to sequelae at start of follow-up? Are there special confounders for sequelae at start versus at follow-up?

Comment 19: Please reduce the number of tables (27 tables).
There are too many sensitivity analyses. Why do the authors need to do all the analyses by SARS-CoV-2 status and admission for COVID-19? Is it possible to only include the comparison btw negative and positive and to compare SARS-CoV-2 status by calendar period, virus variant, lockdown and vaccination? The main comparison is positive vs negative? Also reduce the number of supplementary tables (1-19) Too many on SARS-CoV-2 test and hospitalization. Try to focus on the main objective.

Comment 20: What do the extended data Figures add to the result? Are they necessary? I could not find the referred to in the text.

Comment 21: Methods, line 314-16. Why did not the authors divide the time after infection into more categories after 12 months. Was not the follow-up time 40 months?

Comment 22: Please add in the Methods section whether self-testing was included or not

Comment 23: Sensitivity analyses, line 330-31, adjustments for confounders. Please mention which confounders it was adjusted for in the text and not only referring to the supplementary tables. It is also a bit unclear which specific additional sensitivity analyses that were done for example different SARS-CoV-2 variants etc. Please refer to all the sensitivity analyses (11 analyses) in the text.

Comment 24: Supplementary Figure 1; A bit confusing in the flow chart when it comes to the final study population. I missed

a final study population at the end of the flow chart. There are for example exclusions boxes further down in the flow chart without any information on the final study sample after exclusion.

Comment 25: general comment on the headings in the tables and figures is to remove PCR since all tests are PCR-tests and no self-tests are included?

Comment: Supplementary Table 1. Why do you show study population characteristics for people not included in the study? Is this a kind of sensitivity analysis?

Comment 26: Table 1. The authors should be more precise in their use of wording, In the heading it should be hazard ratio (HR) instead of relative rates. Further, in the footnotes it should be Hazard ratio instead of hazard rate ratio. In addition, the heading is too long and is therefore difficult to understand. However, the authors should mention the confidence intervals in the title.

I think that the heading is rather confusing, Relative rates, relative to what? It is difficult to understand by the heading what is the exposure and what is the outcome?

What is presented in Table 1 is the hazard ratios of the association between SARS-CoV-2 test, number of positive SARS-CoV-2 tests, time since test etc. (exposures) and risk of mental disorders or general medical condition. To make Table 1 more accurate, an alternative heading could be "Hazard ratios and 95% confidence intervals of the association between SARS-CoV-2 test, number of positive SARS-CoV-2 tests, time since test, etc. and mental disorders or general medical condition"

Comment 27: Table 1. Please provide info in the footnote how you define the outcome; mental disorder and general medical condition.

Comment 28: Figure 1. See comment for Table 1. The heading is confusing because the order of exposure and outcome is not accurate, the exposure should come before the outcome. Again, a hazard ratio of what? What are the exposures and outcomes?

Comment 29: What is the reference group in Figure 1a? It is really confusing that the outcome comes before the exposures. You should follow the same order/setup as in Table 1 (where the exposures are far left, before the outcomes at the right-hand side of the table. Again, the heading in the figure should follow the following setup. Hazard ratios and 95% confidence intervals of the association between age group, SARS-CoV-2 status, admission to hospital and mental disorders or general medical disorders. This figure should also include a footnote defining the outcomes such as mental disorders and general medical conditions.

Comment 30: Figure 1b, the exposures should be at the left-hand side of the table and the outcome to the right.

Comment 31: Figure 2. The title is too confusing as it is with the headings in the previous tables and figures. Define the exposures and outcome in the heading, see suggestions above for the previous tables/figures. When I look at the figure it is hard to understand what are the exposures and what are the outcomes? What is the reference? It should be indicated by a square in the figure. What is the difference between the x-axis "number of SARS-CoV-2 positive test and the y-axis Number of SARS-CoV-2 infections? These same comments go for Figure 3 and 4 as the comments for the previous comments.

Comment 32: Figure 5 could be added to the supplementary materials. The headings are very unprecise and do not fit the figures.

General comment to the supplement: Please use Hazard ratios instead of rates which is not very specific in all the headings in all the supplementary tables.

Reviewer #2

(Remarks to the Author)

See attached file for our referee report.

Reviewer #3

(Remarks to the Author)

Reviewer #4

(Remarks to the Author)

Review comments: COVID-19 and all post-infection sequelae on a nationwide scale with 40-months follow-up

As indicated in the title, this paper analyses post-infection sequelae on a nation-wide scale with 40-months follow. The paper uses a wide range of Danish health registers spanning the whole population in the period March 2020 – June 2023. Outcomes are mental, neurological and general medical condition as well as neuropsychiatric conditions. The authors follow the Danish population as a cohort starting with all persons being not tested for Sars-CoV-19 where after they can change status to negative and/or positive based on PCR tests. Both those not tested, test-negatives and test-positives are compared. Cox proportional hazard models are used.

The authors find that those tested positive for SARS-CoV-2 had lower hazard ratios (HR) of mental disorders while hospitalized Covid-19 patients had a very slight increased hazard ratio (HR 1.01) of all disorders except for a few compared with the general population, but that this elevated HR was not different from that following other infections. The increased HR for test-positives was modified by SARS-CoV-2 variant and vaccination status.

Overall impression

The overall impression of the paper is that this is an extremely comprehensive paper looking at a wide range of symptoms following a test result, either negative or positive. A wide range of factors that can modify or influence outcomes are taken into account, including hospital admission, readmission, vaccination, co-morbidity (?) and virus variants.

The paper uses Danish high-quality registers yielding information of a very large number of tested persons and their outcomes. Overall, statistical methods are appropriate.

The scale of the study population and the range of the analyses is impressive.

The paper presents interesting data and provides novel findings on the subject of long COVID, in particular regarding outcomes after other infections that have been little studied.

Figures and graphical abstract are well-designed.

However, there are four main issues of concern related to this paper: The paper is extremely comprehensive; Discussion is vague and limited; there is much bias in results that is not dealt with nor discussed; and the large sample size allows the paper to draw conclusions with statistical power that has little biological impact.

Comprehensiveness: The paper aims to identify post-infection sequelae related both to mental health issues, general medical conditions and neurological questions. 'General medical conditions' cover most disease groups in the ICD8 and ICD10 diagnosis systems. Thus, the authors wish to identify most sequelae (or differences in occurrence between SARS-CoV-19 test groups), not only in a specified time period (e.g. within one month corresponding to acute Covid or later than that corresponding to Long-Covid), but over a wide time frame from less than 1 month to more than 18 months after test. Not only do the authors estimate HRs between those tested positive and negative, but also between these groups and those not tested. The authors also include number of tests, hospital admissions (no admissions and admissions with and without intensive care), readmissions and length of admissions and exposures. Other exposures or modifying factors are age, CRP (C-reactive protein), vaccinations, and time period/SARS-CoV-2 strain.

The paper is not hypothesis driven, but rather explorative.

Thus, the authors obtain a huge amount of results which is reflected in the volume of supplementary material (81 pages(!)). However, only few of these results are discussed. E.g. the authors describe the study population in detail (supplementary table 2). But these pieces of information are not really used nor discussed in the paper. Any differences between the groups?

On the other hand, so many results cannot be realistically discussed in one paper.

There is no doubt that the paper shows interesting results. Of those, particularly those concerning comparisons with sequelae after other infections than Covid-19 appear interesting. The natural history of SARS-CoV-2 is not fully described, and there are many remaining questions regarding SARS-CoV-2 sequelae. However, there is also, at present, a large volume of papers published on different aspects of Long-Covid, including nation-wide questionnaire-based papers from Denmark that are not quoted. Overall, such papers deal with aspects of Long-Covid rather than the overall picture.

The main impression of this paper is therefore that it would serve as a fine overview paper and starting point for testing of specific hypotheses regarding sequelae of Covid-19, if little was known at this time. The authors rightly point at problems with previous papers that such papers use different comparison groups, different time frames for follow-up, different symptom groupings, etc. Therefore, the authors used the same methodology to look at a range of exposures and outcomes as explained above. But there are also many methodological issues related to this paper, and the question is therefore whether it is too late in the exploration of the natural history of Covid-19 to produce a very high amount of data that are little explained, discussed or put into context.

Discussion. This section is very short, just touching upon differences on mental disorders and general medical conditions between those tested for SARS-CoV-2 (positive and negative) and those not tested. But first, the quoted/discussed results

are not completely identical to what is written in the abstract nor in the table (in Discussion it says 'However, positive compared with negative SARS-CoV-2 tests showed no association with mental disorders...' while in Abstract and Table 1 'Compared with negative tests, positive SARS-CoV-2 PCR tests were associated with lower rates of mental disorders (hazard Ratio 0.82, 95% CI 0.80 – 0.83)). And second, this difference is not discussed. Why is it beneficial to mental health to being tested positive to SARS-CoV-2 compared to being tested negative? Third, why should the development of mental disorders be explained by CRP? It is not according to results, but why do the authors believe that CRP would explain mental health disorders? This is not discussed. Also, CRP peaks as indicated in Fig. 3 for SARS-CoV-2 positive are in general below 10 mg/ml, but this is within the normal range (<10 mg/ml). Also, a CRP of 800 as marked as end point for CRP graphs (Figure 3) are unseen in daily practice (question of this is a realistic value). In general, the discussion of own results only cover one page, of which half of the page discusses what the authors did differently from other papers.

Bias: The authors use the national Danish health registers of high quality. However, there are a number of biases associated with this use. The Strengths and limitations section includes a list of possible biases related to this study, but the possible biases and their impact on results are not discussed. Thus, being tested positive for SARS-CoV-2 results in a lower HR for mental health issues than those tested negative. This is hard to understand. Could this be a result of bias? Who are being tested multiple times relative to few times? Also, as the authors rightly point out, all study persons with pre-existing outcome disorders since the start of the registers were excluded, and the study population therefore consisted of very healthy individuals. Why was not paid more attention to this fact in the Discussion, Abstract or elsewhere? This is quite important as the results may not be fully generalizable to the general population because of this bias. In other respects the strength of the paper is that it is population-based and the reader clearly gets the impression that this paper gives a full picture of Covid-19 sequelae in a general population.

Sample size and interpretation: Being population-based over several years, the paper includes an impressively high number of study persons and events. This also gives statistical strength. However, the authors underline this too much when they state (Results and Table 1) that 'Positive SARS-Cov-2 PCR tests were associated with only slightly higher general medical condition rates compared with negative SARS-Cov-2 PCR tests (HR 1.01, 95% CI 1.00 – 1.02, p-value 0.024)'. This may be significant (how can it be so when the CI includes 1.00?), but an increased HR of 1% (HR 1.01) has no biological nor public health impact. A HR of 1.01 is essentially the same as a HR of 1.00.

To sum up – The findings of the paper are so comprehensive, yet subject to bias and little discussed that it is difficult to extract the important findings. An overall paper of all sequelae over a large period of time with many confounders or effect modifiers appears of less use now than some years ago.

A suggestion would be to divide the paper in a number of papers addressing different aspects of Covid-19 sequelae that are then discussed in more detail plus an overview paper describing the common methodology with a clear delimitation of the study population (how large a fraction of the Danish population would not have a pre-existing comorbidity of the given types) and with a discussion of possible bias.

Specific comments

In its present form, the paper could benefit from improved text structure. It is often challenging to follow the authors' line of thought in each section, and the main points are sometimes unclear. Enhancing the flow of the text would make the content easier to follow and increase the overall readability.

Given the impressive size of the paper with supplementary material it is not possible to touch on all parts of the paper including methodology. But the following lists a number of specific points to the paper in its present form in addition the general comments above:

Title

1. Suggest revising the title to make it more informative by clearly stating the paper's main finding.

Abstract

1. Lines 37–38. The methods and results related to the comparison of sequelae with other infections should be elaborated upon in the abstract, as this represents one of the primary objectives of the paper.

Introduction

1. Lines 47-48: This statement requires a citation.
2. Lines 50-51: Is the risk of adverse health outcomes the same for all individuals with COVID-19, regardless of acute illness severity (hospitalized vs. non-hospitalized)? Please elaborate.
3. Lines 59–60: The statement, "In older adults, the sequelae resembled those of viral lower respiratory tract illness," is unclear. Please specify by providing additional context.
4. Lines 65-67. "However, the findings of prior studies depended on whether the control group was individuals with negative tests, no positive test, or various non COVID-19 infections". Needs a citation.
5. Lines 50-71. This section is informative and well-documented, but the structure could be improved. Adding sentences to clarify the relevance of the information to the paper would enhance readability and overall coherence.

Results

1. Consider reporting only the 95% CIs in the results section, as they provide the most relevant measure for assessing statistical significance, making the inclusion of p-values unnecessary.
2. Line 102. What do the authors mean by "depend on time since testing"?
3. Lines 114-116. "Temporally, the general medical condition rates were highest 1-month post-COVID-19 (HR 1.15 1.10,

95% CI 1.06-1.13, p-value <0.001) and remained elevated 18 months post-COVID-19 (HR 1.07, 95% CI 1.04-1.10, p-value <0.001)". Which groups are you comparing? Are the rates higher among those individuals who tested positive compared to those who tested negative?

4. Lines 120-122. "The HR of general medical conditions was flat for CRP values in the age groups younger than 40 years and increasing in increasing CRP for the age groups between 40 and 79 years". This sentence is unclear. Please clarify what is meant.

Discussion

1. Lines 209-211. It would be relevant to mention that not all previous studies focused solely on hospital contacts.

2. Lines 216-218. "Sensitivity analyses revealed that, in the current situation with widespread vaccination and dominant Omicron subtypes, COVID-19 poses a lower risk to humans than previously". Do the authors mean a lower risk of acute or post-acute symptoms, and does it include all symptoms?

3. The main section of the discussion focuses mainly on listing the key findings of the paper, which is appropriate. However, the authors do not sufficiently discuss or contextualize their findings in relation to other studies on long COVID, aside from a single Danish study. This comparison with broader research could provide more depth and perspective, and would strengthen the discussion by relating their results to the existing evidence in the literature.

4. Line 235. Why did the authors choose to exclude sequelae treated in primary care?

5. Line 247. What other infections? Please give examples.

6. Lines 247-248. "However, the unique data on COVID-19 can help broaden our understanding of infections and other disorders in general". This is a valid point however; this aspect is not even mentioned in the general discussion.

Methods

1. Did the authors have access to diagnoses registered in the primary care sector? If no, please describe.

2. Line 296. Please explain why the authors chose to adjust for those specific confounders.

Figures and tables

1. Figure 3 contains a lot of information, making it difficult to read. Suggest to select only a few key specific disorders to present. Additionally, expanding the figure caption with more details would improve understanding.

2. Supplementary Table 20 should be moved to the main paper, as it provides important context and details on the specific diagnoses examined, which are valuable for understanding the study's results.

Reviewer #5

(Remarks to the Author)

Version 1:

Reviewer comments:

Reviewer #1

(Remarks to the Author)

I would like to thank the authors for very thorough responses to my comments related to their manuscript "COVID-19 post-infection sequelae are comparable to sequelae observed after other infections of similar severity – a nationwide Danish study with 40-month follow-up". I have no further comments and I recommend this manuscript to be accepted for publication.

Reviewer #2

(Remarks to the Author)

See attachment for referee report.

Reviewer #3

(Remarks to the Author)

Reviewer #4

(Remarks to the Author)

Comments to revised version of paper entitled 'COVID-19 post-infection sequelae are comparable to sequelae observed after other infections of similar severity – a nationwide Danish study with 40-month follow-up'

Overall impression

The authors have made significant improvements to the paper, enhancing its readability and overall flow.

It is clearly an improvement of the paper that the title has changed to highlight what the authors believe is the main conclusion of the paper, namely that symptoms after Covid-19 are not different from those of other similar infections. This has addressed one of the main issues with the paper as previously indicated, that it addressed many different issues and research questions.

However, the paper still presents a very comprehensive range of findings, sometimes lacking focus. While the introduction has been greatly improved and states some of the main points, it remains difficult to fully grasp the paper's primary objective. This is also reflected in the abstract, where only one conclusion is listed (see specific comments below).

In our primary review we noted that this study did not appear to be hypothesis driven, but rather explorative. The authors replied that the hypotheses were to investigate if there are long-term sequelae after SARS-CoV-2 and whether these are specific or similar to the long-term sequelae after other infections. However, this is not a hypothesis per se. We believe it would help the reader to fully understand the paper if the Introduction section included a section of what exactly is unclear (in brief) related to Long-covid (types of outcome/diagnose groups, time period after infection, variant, measures of Covid-19 severity etc.) and then list the primary aim (which is done) and then specify secondary aims clearly. As listed now in the Introduction, many exposures, effect modifiers and outcomes are listed in a way that is difficult to grasp.

The Results section is better structured in the revised text. The analyses are undoubtedly of high quality but some findings are described too vaguely, making it challenging to fully understand the key takeaways.

The discussion has also improved, with many main findings now addressed. However, some results and potential biases remain undiscussed. In our primary review we noted that there were major methodological issues related to the paper which the authors in their response asked us to identify. These were listed in the subsequent text of the original review. We do not question the qualifications of the authors nor the involved statisticians and believe that analyses have been calculated correctly. However, there are, inherently, a number of epidemiological issues regarding mainly bias and interpretation of results as indicated in the primary review that need discussion, and we believe that this has not been fully done according to the specific comment.

Besides this, the generalizability of the findings to other populations should be considered in the discussion. A broader discussion of how this study contributes to the existing body of literature on long COVID is missing, though it is briefly mentioned in the conclusion.

Thus, given the very comprehensive material, we believe, besides addressing the specific points as indicated, the readability of the text would improve by focus more on specific questions and mainly the question of comparability of outcomes of Covid-19 compared with other infections as indicated in the new title.

Finally, in the revised version there appear some linguistic and typing errors (e.g. P. 13 L 284 'Sequalae' and P. 3 L. 62 'Studies comparing positive with negative SARS-CoV-2 63 PCR tests among non-hospitalized individuals found higher 6-month post-COVID risks of certain 64 hospital diagnoses, such as venous thromboembolism, [but?] the risk was not higher of serious complications like ischemic stroke, encephalitis, or psychoses'). We suggest that the text should be scrutinized for these.

Specific comments

Abstract

1. What is the overall conclusion of the study? Four sentences (lines 35 – 42) list the main findings. But what are the conclusions of those?
2. L. 35: 'Compared with negative tests, positive SARS-CoV-2 PCR tests were not associated with clinically relevant increased rates of mental disorders or general medical conditions. Rates of general medical conditions after a positive compared with a negative SARS-CoV-2 PCR test were only elevated for virus-types preceding Omicron and for individuals with less than 3 vaccinations.' Aren't those two sentences not contradictory to each other?
3. "Compared with the general population, the rates were most elevated among hospitalized COVID-19 patients, and particularly with ICU treatment". Rates of what? Needs clarification.

Introduction

1. The introduction has improved significantly; apart from what is written above we have only one minor comment. Line 52: "...which is of utmost importance for healthcare planning and the society." In what way is it important?
2. Line 50. What other infections? It would be informative to give a few examples.

Results

1. Are individuals with a prior hospital contact for mental disorders and/or general medical conditions excluded from your study population? It is unclear in the text. If they are excluded, the introduction should state this more clearly to clarify that the results are based on a healthy study population. Consider moving Supplementary Figure 1 to the main paper to give an overview of the study population.
2. Line 118: "...were decreased during all periods of time since testing". This sentence is unclear. Please elaborate.
3. Lines 157-160. This section is unclear. Are you comparing test-positive individuals who had been prescribed an anti-infective drug with test-negative individuals who also had been prescribed an anti-infective drug?
4. Lines 167+170. Compared to COVID-19 infected, non-hospitalized individuals?

5. Lines 199-205. Suggest excluding these results (including Table 2) from the main paper, as they seem redundant. Consider moving them to supplementary materials.
6. Lines 222-231. These results are very unclear. It is difficult to determine the exact comparison being made. Which groups are included in the analysis, and what CRP levels are being compared?

Discussion

1. It is highlighted that this study was able to include all PCR tests for SARS-CoV-2 conducted in Denmark, which was important for capturing mild/asymptomatic COVID-19 cases, unlike previous studies. However, since data from the primary care sector was not included in your analysis, there is a significant risk that many individuals in the study population sought medical care for mental disorders and/or general medical conditions in the primary sector rather than in hospitals, which is likely the case for most mild/asymptomatic cases. This may have affected the estimation of rates and represents a possible sampling bias that should be discussed further.
2. Lines 281-284. What could be the explanation for CRP levels not being associated with a higher risk of post-infection sequelae in hospitalized patients as opposed to non-hospitalized patients?
3. Lines 286-292. This section should be included in the methods section and not in the discussion.
4. The results regarding the impact of prescribed anti-infective drugs on the HR of post-infection sequelae are not discussed. This should be mentioned, as there are many biases that could have impacted the results.
5. Under strengths and limitations, it should be noted initially that the study did not have access to data from the primary care sector. Although it is stated that the study had complete data, this is not entirely accurate due to the absence of primary care data.
6. Is it possible that a proportion of the test-negative individuals or those who were never tested had tested positive using a home antigen test, thereby increasing the risk of misclassification bias?

Methods

1. The use of CRP measurements from the Register of Laboratory Results for Research should be described in more detail. How were the CRP levels linked to the SARS-CoV-2 test? Did the CRP measurement have to be conducted within a certain time frame relative to the test date to be included in the study? If so, how did you account for the potential time difference between a SARS-CoV-2 test and a measured PCR value?

Figures and tables

1. Figure 3 is very hard to comprehend, as well as the section describing some of the main results (lines 152-161)

Reviewer #5

(Remarks to the Author)

Version 2:

Reviewer comments:

Reviewer #2

(Remarks to the Author)

Please see the attached file.

Reviewer #3

(Remarks to the Author)

Reviewer #4

(Remarks to the Author)

The authors have submitted a second revised version of the paper, including a rebuttal letter addressing all raised reviewer items point by point.

In general, we find that the authors have sufficiently addressed the raised issues in this second review. Now the aims, methods, results and conclusions of the study stand out quite more clear. I believe the lack of focus for many of the findings have now been explained so the paper now stands more focused.

We have no major comments to the present revision of the paper.

Reviewer #5

(Remarks to the Author)

Point by point response to the reviewers

Manuscript #: NCOMMS-24-86598-T

Title: COVID-19 post-infection sequelae are comparable to sequelae observed after other infections of similar severity – a nationwide Danish study with 40-month follow-up

REVIEWER REPORTS

Reviewer #1 (Remarks to the Author):

Using the nationwide Danish registers, this large study (3,343,706 individuals) used SARS-CoV-2 PCR test results and COVID-19 hospitalizations from March 2020 to June 2023, and estimated Hazard ratios of mental, neurological, and general medical conditions, as well as of neuropsychiatric symptoms. The authors found that compared with negative tests, positive SARS-CoV-2 PCR tests were associated with lower risk of mental disorders and minimally higher rates of general medical conditions. This large study is well conducted and an important contribution to the research on long-COVID symptoms but needs major revisions. Please see my comments below:

Authors' reply: We are grateful for the reviewer's very positive evaluation of our paper and thoughtful suggestions, which has strengthened our manuscript.

Comment 1: Methods: exposure to SARS-CoV-2 paragraph; did the study include only tests from the registry MiBa? What about self-testing? Please add a sentence that you did not use self-tests and how this could have influenced the results, for example selection bias.

Authors' reply: Since a positive self-test required subsequent confirmation with PCR testing, as PCR testing is regarded as the most reliable method for detecting COVID-19, we argue that relying solely on PCR test results to define 'confirmed COVID-19' minimizes bias more effectively than including other test types. Including self-test data, conversely, would introduce greater risk of bias due to their lower reliability.

We have added a clarifying sentence in the *Exposure to SARS-CoV-2* paragraph of the Methods section: "PCR tests were the best method for detecting SARS-CoV-2,⁴⁰ and we did not include results from other type of tests such as self-testing, as a positive self-test required subsequent confirmation with PCR testing."

Comment 2: Study population-flow chart: only those with PCR-test? Only self-test after February 2022 in Norway, the same in Denmark? Low number of participants with PCR-test after omicron? Who are these participants with PCR-test after Omicron? Selection bias.

Authors' reply: We realize that we have not successfully described the testing strategy and behavior in Denmark during the pandemic, nor why there were lower rates of mental disorders for individuals tested positive compared to negative. First, we have added the results from the sensitivity analysis stratified by the number of SARS-CoV-2 tests in Table 2 in the main paper, and highlighted the results in the 'Sensitivity Analysis' section of the Results:

"Number of SARS-CoV-2 tests and the rates of long-term sequelae

In the comparison between positive and negative SARS-CoV-2 PCR tests, we stratified by the number of conducted tests (Table 2 and sensitivity analysis 2). The rates of mental disorders were lower for positive compared with negative tests regardless of the number of tests, though the HR increased with more tests conducted. For general medical conditions, the HRs for positive

compared with negative tests remained close to non-significant, and only slightly increased with more tests conducted.”

Second, we have extended the discussion to provide a more thorough analysis of the results:

“In Denmark, PCR tests were available from the beginning of the pandemic and were predominantly used until they were gradually phased out during spring and summer 2022.²⁶ In May 2021, a nationwide *corona passport* was introduced to facilitate a safe reopening of society, requiring either a recent negative SARS-CoV-2 test or full vaccination for access to social and educational activities. It should be noted that equal access to testing facilities²⁷ and reduced economic barriers in Denmark compared to other countries might result in higher detection rates of mild cases in our study. The lower rates of mental disorders observed among individuals with positive compared to negative tests could have several explanations. First, COVID-19 survivors were possibly examined for general medical conditions before mental disorders were considered. Second, the rates of mental disorders depended on the number of tests conducted, suggesting an unexplained dependency on the testing behavior in the evaluation of mental health outcomes. Third, access to social activities both increases the risk of contracting COVID-19 and strengthens social relationships, which are known to support mental health.²⁸ Fourth, an undiagnosed mental disorder might influence the incentive to get tested. Fifth, the study population consisted of very healthy individuals, since we excluded those who, at any point since the start of the registers, were diagnosed with any pre-existing disorder that was studied as outcomes in this research. However, a post hoc sensitivity analysis with a 5-year only washout period for prior conditions did not change results, and thus healthy user bias did not seem to have a major impact on the findings.”

Third, we have added the following to the limitations of the discussion:

“Last, testing behavior, such as incentive to get a SARS-CoV-2 test, was not captured in the registers, and could explain why positive tests compared with negative tests was associated with lower rates of mental disorders.”

Comment 3: Too short discussion, half a page... extend the discussion. There are several other studies who have included non-hospitalized and tested for SARS-CoV-2 (Ballering, van Zon et al. 2022, Ellingjord-Dale, Brunvoll et al. 2024, Ellingjord-Dale, Nygaard et al. 2024), Ballering, A. V., S. K. R. van Zon, T. C. Olde Hartman and J. G. M. Rosmalen (2022). "Persistence of somatic symptoms after COVID-19 in the Netherlands: an observational cohort study." *Lancet* 400(10350): 452-461. Ellingjord-Dale, M., S. H. Brunvoll and A. Soraas (2024). "Prospective Memory Assessment before and after Covid-19." *N Engl J Med* 390(9): 863-865. Ellingjord-Dale, M., A. B. Nygaard, N. C. Støer, R. Bø, N. I. Landrø, S. H. Brunvoll, M. Istre, K. T. Kalleberg, J. A. Dahl, L. Geng, K. Tsilidis, E. Riboli, G. Ursin and A. Søråas (2024). "Temporal trajectories of long-COVID symptoms in adults with 22 months follow-up in a prospective cohort study in Norway." *Int J Infect Dis* 149: 107263.

Authors' reply: We thank the reviewer for suggesting these references to survey-based studies on self-reported symptoms related to long COVID. While our study focuses on clinical outcomes rather than self-reported questionnaires, we have incorporated the references into the extended discussion in line with the reviewer's previous comments:

“Prior survey-based studies examining self-reported post-COVID-19 symptoms included individuals with PCR tests taken out of hospitals and found higher prevalence of neuropsychiatric symptoms such as dizziness, painful muscles, general tiredness/fatigue, ageusia/anosmia/dysgeusia, and memory problems after positive compared to negative tests.^{3,23,24}”

Comment 4: Line 204. “hazard ratio of general medical conditions increased with CRP in individuals with positive SARS CoV-2 tests, though not in patients hospitalized with COVID-19”. Please discuss in the discussion section why you did not find any difference? Who was the comparison group? If hospitalized they have already a high CRP?

Authors’ reply: We appreciate the reviewer’s keen observation. In order to make the associations with CRP clearer, we have combined the results on CRP and moved them down to a separate paragraph under sensitivity analyses in the result section. Moreover, we have refined our discussion by adding a paragraph in the Discussion addressing the results related to CRP, and putting the findings in context, while keeping the overall focus in the manuscript on the most important findings, as emphasized by the reviewers.

Comment 5: line 240, please define pre-existing outcome disorders.

Authors’ reply: We have rephrased the sentence to avoid the term ‘*outcome disorder*’. Note, that the sentence has been moved from ‘*Limitations*’ to the general discussion of results:

“The study population consisted of very healthy individuals, since we excluded those who, at any point since the start of the registers, were diagnosed with any pre-existing disorder that was studied as outcomes in this research.”

Comment 6: line 239-242: 2.5 million people were excluded from the study having pre-existing outcome disorder. Please add a sentence or two on how this large exclusion of people could have affected the results, under- or overestimation? Healthy bias effect as limitation. Selection bias etc.

Authors’ reply: We did not exclude 2.5 million people from all analyses and apologize for this misunderstanding. We excluded individuals with prior hospital contact for specific outcome disorders. So for mental disorders, we excluded 910,236 individuals, and for general medical conditions, we excluded 2,585,417 individuals. We have added Supplementary Table 1, 3, and 4 with overviews of the number of individuals in the populations and revised the first paragraph of the RESULTS section to make this clearer.

We highlight this sentence from the Statistical analysis paragraph in the Methods:

“Patients with pre-existing diagnoses were excluded within each specific outcome diagnosis. For example, when investigating kidney disorders, patients with kidney disorders before the start of follow-up were excluded, but not patients with psychiatric diagnoses.”

Also, we have added “*due to healthy user effect*” to the sentence in question:

“As a result, the effect estimates were not generalizable to the general population due to healthy user bias..”

Moreover, we conducted sensitivity analyses only excluding individuals with a hospital contact within five years before study start, which reaches the same conclusions as the main analyses, which we have now displayed in a separate section under sensitivity analyses and discussed in the discussion section.

Comment 7: I could not find any sensitivity-analysis on the 2.5 million excluded from the study? Please, do a sensitivity analysis including these large group of people.

Authors’ reply: We appreciate the reviewer’s insightful feedback, which led us to reassess the exclusion criteria for the study population. In response, we conducted a new sensitivity analysis, which yielded similar results. This suggests that the healthy user bias is minimal. The details of the analysis, along with the corresponding additions to the manuscript, are outlined below.

Since we aim to estimate rates of incident disorders, we want to avoid a prevalent design. Therefore, we exclude individuals with a hospital contact for a pre-existing diagnosis within 5 years prior to the start of the study.

Also, we have added information about the analysis in the Methods section:

“Post hoc, we conducted an additional sensitivity analysis to investigate how exclusion criterion affected results by only excluding individuals with a hospital contact for a pre-existing diagnosis 5 years before the start of the study.”

This novel post.hoc analysis with only a 5-year washout period, excluded 659,295 and 1,732,096 individuals with hospital contacts for mental disorders and general medical conditions, respectively. The results are noted in the paragraph titled “Analyses with five years washout period only for prior hospital contacts” under the *sensitivity analyses* in the Results section:

“Post hoc, instead of excluding individuals with hospital contacts for pre-existing disorders at any point in time before study start, we conducted additional analyses only excluded individuals with a hospital contact within five years before study start (Sensitivity analysis 8 and 14). Thus, excluding 659,295 and 1,732,096 individuals with hospital contacts for mental disorders and general medical conditions, respectively. The results were generally unchanged. The rates of mental disorders were decreased, and the rates of general medical conditions were not significantly different for positive compared to negative tests. The rates were still increased for mental disorders and general medical conditions after admission with COVID-19 compared to no admission with COVID-19.”

The following is added to the Discussion:

“However, a post hoc sensitivity analysis addressing the exclusion criterion of the study population did not change the results.”

Comment 8: line 241-242- Please rephrase this sentence. Could the authors be more specific than stating that “some effect estimates were not...an unselected population”? “Some” effect estimates? Which ones? In my opinion, this goes for the whole study population and all the effects. By “unselected “I assume you mean the general population in Denmark? Please rephrase.

Authors’ reply: We have revised and rephrased the sentence in the discussion section to the following: “the study population consisted of very healthy individuals, since we excluded those who, at any point since the start of the registers, were diagnosed with any pre-existing disorder that was studied as outcomes in this research. However, a post hoc sensitivity analysis with a 5-year only washout period for prior conditions did not change results, and thus healthy user bias did not seem to have a major impact on the findings.”

Comment 9: Conclusion, line 245-46: Unprecise fist sentence-. “Hospitalized COVID-19 individuals have higher rates of sequelae, whereas non-hospitalized COVID-19 individuals do not” Compared to who? Please be more specific.

Authors’ reply: We thank the reviewer for highlighting the need for improved clarity in this sentence and we have now revised the sentence in the Conclusion to enhance its clarity.

Comment 10: line 245-246, “Hospitalized COVID-19 individuals have higher rates of sequelae, whereas non-hospitalized COVID-19 individuals do not” Higher rates, should it not be “risk of sequelae” and not “rates” since HR is a risk-estimate? Please be more specific.

Authors’ reply: Respectfully, we are convinced that it is correct to write rates instead of risks.

A hazard ratio is the ratio of hazards, and hazards are rates (and not risks). Since there are several other competing risks, hazard ratios are not proportional to risks. For instance, see this paper highlighting that, for all other outcomes than all-cause mortality, there is not a one-to-one correspondence between rates and risks (Andersen PK, Geskus RB, de Witte T, Putter H. *Competing risks in epidemiology: possibilities and pitfalls. Int J Epidemiol. 2012 Jun;41(3):861-70. doi: 10.1093/ije/dyr213*).

Comment 11: line 246-47. “Sequelae are not exclusive to COVID-19 survivors but also occur after other infectious diseases with similar severity”. What do the authors mean “by similar severity”? Hospitalized? Severity according to what? Please be more specific.

Authors’ reply: As the reviewer correctly points out, by ‘similar severity’ we are referring to admission status. To improve clarity, we have rephrased the Conclusion as follows:

“Sequelae are not exclusive to COVID-19 survivors; they also occur after other infectious diseases (for example influenza and bacterial pneumonia) of similar severity, as measured by admission status.”

Comment 12: line 247-48, “However, the unique data on COVID-19, can help ...of infections”. I assume that the authors mean” COVID-19 infection compared to other infections (SARS-CoV-2-negative). Please be more precise.

Authors’ reply: We thank the reviewer for pointing this out and have now rephrased the sentence for more clarity.

Comment 13: line 239-241. Excluding pre-existing outcome disorders? How many? What do the authors mean by pre-existing outcome disorders? Please be more specific and define.

Authors’ reply: We have rephrased the sentence in the response to comment 5.

Comment 14: Line 260-Methods: “Individuals were followed from March 1, 2020, until the onset of the disorders of interest and censored in case of death, emigration...” I assume that it should say ;...or censored in case of death, emigration..”

Authors’ reply: The reviewer is correct, and we have now replaced ‘and’ with ‘or’ in the appropriate part of the Methods.

Comment 15: line 279-280. Please be more precise in the definition of primary outcomes: what kind of mental disorder, please give examples? The same goes for general medical conditions; please specify.

Authors’ reply: We have now expanded the text to include information from the Supplementary Table. Specifically, we have added the details of the mental disorders and the general medical condition categories:

“Any first-time mental disorder (including mental and behavioral disorders due to psychoactive substance use, psychotic, mood, and anxiety disorders) or general medical condition (including selected neurological, respiratory, gastrointestinal, circulatory, kidney, endocrine, hematological, musculoskeletal, and dermatological disorders, as well as neuropsychiatric symptoms) based on the prior literature on identified associations.”

Comment 16: line 292-93, Statistical analyses. This sentence is confusing “This method has been proven to minimize bias when estimating vaccination effectiveness”. Do the study measure

vaccination effectiveness as the main objective? And cox regression is widely used as a survival analysis, time to event, risk analysis etc., so it is a bit misleading to state that it is known for minimizing bias in vaccination effectiveness. Please remove this sentence.

Authors' reply: We appreciate the reviewer's careful reading, and we have removed this sentence.

Comment 17: Line 293-94; the authors mention that they reported Hazard rate ratios (HR), shouldn't it be hazard ratio? The effect estimate should be referred to as hazard ratio throughout the manuscript.

Authors' reply: The term 'hazards' can be used synonymously with 'hazard rate.' However, to avoid confusion between rates and risks, we typically prefer the term 'hazard rate ratio.' In response to the request, we have rephrased the text to use the more commonly accepted term 'hazard ratio.'

Comment 18: line 296-297 "The analyses were adjusted for the following confounders related to COVID-19 and sequelae at start of follow-up". Confusing sentence: what are the exposures and what is the outcome in the analyses?

What do the authors mean by confounders related to sequelae at start of follow-up? Are there special confounders for sequelae at start versus at follow-up?

Authors' reply: The confounders are defined based on their value at the 'start of follow-up.' For example, if a person is 25 years old at the start of follow-up, we use 25 years as their age throughout the study period. Therefore, the confounders are not influenced by when the sequelae are assessed. We have rephrased the sentence in the Statistical Analysis section of the Methods for clarity:

"The analyses were adjusted for confounders related to the exposure COVID-19 and the outcome sequelae, and the confounders were defined at the start of follow-up: Sex, socioeconomic status..."

Comment 19: Please reduce the number of tables (27 tables).

There are too many sensitivity analyses. Why do the authors need to do all the analyses by SARS-CoV-2 status and admission for COVID-19? Is it possible to only include the comparison btw negative and positive and to compare SARS-CoV-2 status by calendar period, virus variant, lockdown and vaccination? The main comparison is positive vs negative?

Also reduce the number of supplementary tables (1-19) Too many on SARS-CoV-2 test and hospitalization. Try to focus on the main objective.

Authors' reply: In the main manuscript there is one table only. We have adhered to our carefully defined, pre-registered study protocol, which was uploaded to OSF prior to analyses. The protocol was developed based on experience from several similar studies on infections and psychiatric/neurological disorders. We believe the additional analyses are both necessary and appropriate for evaluating the severity, temporality, and other key aspects of the research question, ensuring a comprehensive investigation, which have been lacking so far. We prefer not to deviate from the study protocol, as our goal is to maintain the highest possible level of scientific rigor.

However, since the main comparison is positive vs negative, we agree that sensitivity analyses could be made solely on positive vs negative, and not hospitalization. Instead of removing the sensitivity analyses on hospitalization, we have refined the structure in the Supplementary material and hope that this is sufficient to ensure focus on the main point.

Comment 20: What do the extended data Figures add to the result? Are they necessary? I could not find the referred to in the text.

Authors' reply: The extended data figures are cited in the Results section as '*Extended Data Fig. (NUMBER)*'. We have reevaluated the inclusion of all Extended Data Figures. To ensure a higher focus on the main point of the paper, we deleted Extended Data Fig. 1 and 4 and moved Extended Data Fig. 2 to Supplementary Figure 2 and 4.

Comment 21: Methods, line 314-16. Why did not the authors divide the time after infection into more categories after 12 months. Was not the follow-up time 40 months?

Authors' reply: Additional categories beyond 12 months include 12–17 months and 18+ months. To describe the temporal pattern in greater detail, we have extended the last category in 6-month increments, resulting in the following groups: 18–23 months, 24–29 months, and 30+ months. In the latest interval (30+ months), the number of cases is lower than in previous intervals, leading to broader confidence intervals. The rates of mental disorders remained decreased across all time intervals, while the rates of general medical conditions increased for all intervals except the latest (30+ months). We have now incorporated these findings into the results section with the subheading '*SARS-CoV-2 positive test results and the rate of any first general medical condition*':

“Temporally, the general medical condition rates were highest 1-month post-COVID-19 (HR 1.10, 95% CI 1.06-1.13, p-value <0.001) and remained significantly elevated until 30 months post-COVID-19 (HR 1.01, 95% CI 0.91-1.12, p-value 0.855) for positive compared with negative SARS-CoV-2 tests.”

Comment 22: Please add in the Methods section whether self-testing was included or not

Authors' reply: We have added a sentence in response to comment 1.

Comment 23: Sensitivity analyses, line 330-31, adjustments for confounders. Please mention which confounders it was adjusted for in the text and not only referring to the supplementary tables. It is also a bit unclear which specific additional sensitivity analyses that were done for example different SARS-CoV-2 variants etc. Please refer to all the sensitivity analyses (11 analyses) in the text.

Authors' reply: We have specified the different levels confounder adjustments in the '*Sensitivity Analysis*' section of the Methods:

“In sensitivity analyses, the robustness of results was assessed by repeating the analyses with various adjustments for confounders: i) no adjustment, ii) stratification for age in 10-year bands, iii) stratified for age and adjusted for sex, iv) stratified for age, adjusted for sex, CCI, parental CCI, parental mental health, employment status, income, the highest level of education.”

All sensitivity analyses are mentioned in the text, but we have now added Roman numerals to clarify the distinctions:

“Given the changes in SARS-CoV-2 contagiousness, testing behavior, and vaccination uptake throughout the pandemic, we also examined i) the number of conducted tests, ii) calendar periods, iii) dominating SARS-CoV-2 variants, iv) lockdown periods, v) vaccination status, and vi) immigration status at the individual level”.

Comment 24: Supplementary Figure 1; A bit confusing in the flow chart when it comes to the final study population. I missed a final study population at the end of the flow chart. There are for

example exclusions boxes further down in the flow chart without any information on the final study sample after exclusion.

Authors’ reply: The flowchart provides a comprehensive visualization of the study populations used in each analysis, but we understand that trying to include everything in one figure may be confusing. Therefore, we have created subfigures, now available in Supplementary Figure 1. For example, part (a) of the figure illustrates how we arrived at the study population:

“a Study population.

Comment 25: general comment on the headings in the tables and figures is to remove PCR since all tests are PCR-tests and no self-tests are included?

Authors’ reply: We have now removed ‘PCR’ from ‘SARS-CoV-2 PCR test’ in all headings and figure/table legends.

Comment: Supplementary Table 1. Why do you show study population characteristics for people not included in the study? Is this a kind of sensitivity analysis?

Authors’ reply: By ‘people not included in the study,’ we assume the reviewer refers to individuals who were not tested for SARS-CoV-2. We present them in the population characteristics, as they are included in the primary analysis, where we compare individuals with positive, negative, and no tests. We included the untested individuals to demonstrate the sensitivity of results based on the choice of reference/control group. However, the untested individuals are not included in any other analysis.

Comment 26: Table 1. The authors should be more precise in their use of wording, In the heading it should be hazard ratio (HR) instead of relative rates. Further, in the footnotes it should be Hazard ratio instead of hazard rate ratio. In addition, the heading is too long and is therefore difficult to understand. However, the authors should mention the confidence intervals in the title. I think that the heading is rather confusing, Relative rates, relative to what? It is difficult to understand by the heading what is the exposure and what is the outcome? What is presented in Table 1 is the hazard ratios of the association between SARS-CoV-2 test, number of positive SARS-CoV-2 tests, time since test etc. (exposures) and risk of mental disorders or general medical condition. To make Table 1 more accurate, an alternative heading could be “Hazard ratios and 95% confidence intervals of the association between SARS-CoV-2 test, number of positive SARS-CoV-2 tests, time since test, etc. and mental disorders or general medical condition”

Authors’ reply: We have followed the advice of the reviewer and updated the title of Table 1 to the following:

“Hazard ratios and 95% confidence intervals of the association between SARS-CoV-2 test, number of SARS-CoV-2 positive tests, time since SARS-CoV-2 test, admission to hospital with COVID-19, number of readmissions with COVID-19, duration of COVID-19-admission and mental disorders or general medical conditions.”

Comment 27: Table 1. Please provide info in the footnote how you define the outcome; mental disorder and general medical condition.

Authors' reply: We have added two footnotes to Table 1 and Figure 1 describing how the outcomes are defined:

^a Including mental and behavioral disorders due to psychoactive substance use, schizophrenia spectrum, mood, and anxiety disorders. The diagnosis codes are summarized in Supplementary Table 25. Results for mental disorders are derived from a study population of n = 5,147,288 individuals.

^b Including selected neurological, respiratory, gastrointestinal, circulatory, kidney, endocrine, hematological, musculoskeletal, and dermatological disorders, as well as neuropsychiatric symptoms. The diagnosis codes are summarized in Supplementary Table 25. Results for general medical conditions are derived from a study population of n = 4,074,487 individuals."

Comment 28: Figure 1. See comment for Table 1. The heading is confusing because the order of exposure and outcome is not accurate, the exposure should come before the outcome. Again, a hazard ratio of what? What are the exposures and outcomes?

Authors' reply: We have updated the title of Figure 1 following the advice of the reviewer.

Comment 29: What is the reference group in Figure 1a? It is really confusing that the outcome comes before the exposures. You should follow the same order/setup as in Table 1 (where the exposures are far left, before the outcomes at the right-hand side of the table. Again, the heading in the figure should follow the following setup. Hazard ratios and 95% confidence intervals of the association between age group, SARS-CoV-2 status, admission to hospital and mental disorders or general medical disorders. This figure should also include a footnote defining the outcomes such as mental disorders and general medical conditions.

Authors' reply: We have updated the titles of all the Figures following the advice of the reviewer. The titles are now on the form:

"Hazard ratios and 95% confidence intervals of the association between (EXPOSURE), (GROUP VARIABLE, e.g., age group, admission status) and (OUTCOME)."

Also, we have added footnotes, cf. the response to the reviewer's comment 27.

Comment 30: Figure 1b, the exposures should be at the left-hand side of the table and the outcome to the right.

Authors' reply: We have moved the outcome to the right-hand side of the plot in Figure 1a and 1b.

Comment 31: Figure 2. The title is too confusing as it is with the headings in the previous tables and figures. Define the exposures and outcome in the heading, see suggestions above for the previous tables/figures. When I look at the figure it is hard to understand what are the exposures and what are the outcomes? What is the reference? It should be indicated by a square in the figure. What is the difference between the x-axis "number of SARS-CoV-2 positive test and the y-axis Number of SARS-CoV-2 infections? These same comments go for Figure 3 and 4 as the comments for the previous comments.

Authors' reply: We have updated the titles based on the reviewer's previous comments.

The reference groups are defined in the foot note and not in the plot to avoid overcrowding it. The values of the y-axis depend on the facet. In the left-hand side, the y-axis denotes number of positive SARS-CoV-2 tests, and in the right-hand side, the y-axis denotes the number of admissions with COVID-19. We have updated the y-axis label to reflect this information: “*Number of SARS-CoV-2 positive tests / Number of admissions with COVID-19*”.

Comment 32: Figure 5 could be added to the supplementary materials. The headings are very unprecise and do not fit the figures.

Authors’ reply: In our opinion Figure 5 (now referred to as Figure 4) helps understand how the hazard ratios have changed during the pandemic which is necessary to understand the complexity of the research question. We hope that the heading is more precise after updating it cf. the reviewer’s previous comments.

General comment to the supplement: Please use Hazard ratios instead of rates which is not very specific in all the headings in all the supplementary tables.

Authors’ reply: ‘Rate’ and ‘hazard’ are well-defined terms, but in response to the reviewer’s request, we have changed to ‘hazard ratios’ where appropriate.

Reviewer #2 (Remarks to the Author):

General comment

This nationwide follow-up of long-term post-acute COVID-19 sequelae aims to compare rates of mental, neurological and general medical conditions in people tested positive for COVID-19 with people diagnosed with other infections. The study is based on very interesting and rich data, and has great potential. However, it lacks some clarity and focus in presentation, and there are also methodological issues that should be looked into.

Authors' reply: We sincerely thank the reviewer for their encouraging words about our work. We are delighted that the reviewer found the study valuable and appreciate their thoughtful feedback, which has helped us better articulate our findings with greater clarity and focus in the presentation.

Main comments

1) Motivation and clarity of presentation: The introduction would benefit from a clear and logical buildup to the study's aims. Now it reads more like a discussion, with scattered arguments and insufficient focus. The last paragraph in particular requires a stronger outline of the study's objectives and the subsequent structure.

Authors' reply: We have revised the introduction to sharpen the focus and moved the detailed content to the discussion section. Also, we have added the primary objective in the last paragraph of the introduction: "Our primary objective was to identify the association between COVID-19 and subsequent mental disorders and general medical conditions."

2) CRP results: The CRP results/discussion appears somewhat disconnected from the main focus of the manuscript. This topic may detract from the overall narrative. Consider its necessity, and if retained it needs better placement.

Authors' reply: We agree that the presentation of the CRP results in the manuscript could be improved. As we have explicitly stated in our pre-registered study protocol, uploaded to OSF, that we would investigate the link to CRP, we believe it would be inappropriate to remove these results from the paper entirely. However, we agree that it needs to be down played to not distract from the main findings and needs a better placement. Thus, we have now moved the CRP main results down to a separate part under sensitivity analyses to accommodate the need for a better placement and additionally made the following changes to create a more intuitive flow in results:

- **Removed Extended Data Fig. 1**
- **Moved Extended Data Fig. 2 to the Supplementary material (now referred to as Supplementary Figure 2 and 4)**
- **Moved Figure 3 to Extended Data Figures and removed the secondary outcomes (now referred to as Extended Data Fig. 3)**

3) Comparison to other infections: These are very interesting results which should be given more place/attention. We suggest a stronger focus on this comparison in the analysis and discussion. We also suggest cutting some of the secondary analysis as it makes the scope of the paper too wide.

Authors' reply: We sincerely appreciate the reviewer's positive feedback and their suggestion to give these results on the comparison to other infections more attention. We agree that this comparison provides very valuable insights, and we have now expanded the analysis and

discussion to emphasize these findings more clearly, including highlighting it more clearly in the abstract.

Additionally, to include the important comparison between COVID-19 and other infections, we have altered the former Figure 4 (now Figure 3) in the main paper as recommended by Reviewer 4. Here, we include information from Supplementary Tables 19-21 (now referred to as Supplementary Tables 22-24), more specifically: i) positive test compared with prescription for anti-infective agent, and admission with COVID-19 compared with admission with non-COVID-19 (pulmonary) infections. And we only include the results for the primary outcomes as also suggested.

Given that the secondary analyses are outlined in the analyses plan of the pre-published study protocol, we do consider it most appropriate to keep these analyses. as we believe these analyses strengthen the evidence regarding the association between COVID-19 and sequelae, particularly in relation to factors such as severity and temporality.

4) Selection into testing: The manuscript does not sufficiently explain or explore the selection mechanism into PCR-testing (tested/did not test), tested positive, or tested negative. This is important - who was tested, tested positive, and not tested, when? There is a need for descriptive statistics that explore/explain the difference between these groups to evaluate the validity of the comparison.

Authors' reply: The cohort characteristics requested by the reviewer are found in the Supplementary Table 1, now referred to as Supplementary Table 2. In the text, we have now explicitly mentioned that this table has cohort characteristics:

“(for cohort characteristics of the study populations of the primary outcomes, see Supplementary Table 2–4).”

Moreover, we have now included sensitivity analyses on the impact of the number of SARS-CoV-2 tests conducted.

5) Group composition over time: The groups - non-tested, positive and negative - change their composition over time (cf. line 306-308)? How to think about comparisons between these groups then? Could the changing composition of the groups introduce biases in the analyses? And, when comparing non-tested to positive, for example, could the same individual in principle be in both groups? Please explain and describe group composition, and under what assumptions the methods used are robust changes in group composition. Also, consider a graphical presentation of the number of individuals in each group over time (x-axis).

Authors' reply: It is correctly understood that the groups change their composition over time, since the exposure is time dependent. Importantly, however, the same individuals cannot be in the different groups at the same time. In the Cox proportional hazards model, hazard rates are compared at specific time points, thus not comparing the same individual to itself. We have now added a brief clarification to ‘*The primary analyses*’ paragraph in the Methods section:

“With this definition, the same individuals could not be in different groups at the same time.”

Also, we have included a new Extended Data Fig. 1 with a graphical presentation of the number of individuals in each group over time. This figure is referred to in the first paragraph of the Results section:

“During the study period, 1,832,328 (54.8%) individuals tested positive for SARS-CoV-2, 1,222,202 (36.6%) only tested negative for SARS-CoV-2, and 289,176 (8.6%) were not tested for SARS-CoV-2 (see Extended Data Fig. 1 for a graphical representation of the number of individuals in each group over time).”

Smaller comments:

1) Descriptive statistics - Table 1: It would be helpful to include a table of descriptive statistics for the analytical sample, broken down by the different groups in the study.

Authors’ reply: Please see our response to Main comment 4.

2) Figure 1: Could you clarify if/where the moderating effects of adjustment variables can be seen? Did results vary by gender, SES, or other observable characteristics?

Authors’ reply: We adjusted for sex, SES, and other relevant confounders in the analyses; however, we do not present results stratified by these variables, as there based on the reviewer comments, seemed to be more than plenty of analyses and sensitivity analyses already included in the manuscript, and as this was not part of the pre-published study protocol.

3) Figure 2: There are quite a few tests here—are the confidence intervals adjusted for multiple testing? If not, it would be good to explicitly state this.

Authors’ reply: The results were not adjusted for multiple testing, and we have now added this information in a footnote to Figure 2.

4) Figure 3: This figure is somewhat difficult to interpret in its current form. To improve clarity, consider either explaining it more thoroughly or moving it to the supplementary material. If CPR is not central to the analysis, you might also consider omitting it altogether.

Authors’ reply: To enhance readability as suggested by the reviewer, we have excluded diagnoses while also expanding the figure caption, for improved clarity. Also, the figure is moved to the extended data figures (now referred to as Extended Data Fig. 3).

5) Figure 4: Would it be possible to streamline this figure? Perhaps the most important (panel a) could remain in the main text, with the others moved to the supplementary material.

Authors’ reply: To include the important comparison between COVID-19 and other infections, we have altered the former Figure 4, now Figure 3. For details, please see our response to reviewer 2’s main comment 3.

6) Figure 5: This figure presents a lot of information—would it be possible to distill the key message into a simpler figure while moving the full version to the supplementary material?

Authors’ reply: In our opinion Figure 5 (now referred to as Figure 4) helps understand how the hazard ratios have changed during the pandemic which is necessary to understand the complexity of the research question.

7) **Contribution:** The paper already makes a strong contribution, but there may be an opportunity to further highlight its uniqueness—for example, by emphasizing the length of follow-up or comparisons to other infectious diseases.

Authors' reply: The reviewer's observation is much appreciated, and we appreciate the opportunity to further highlight the uniqueness of the study.

The following is added to the abstract:

“Using the nationwide registers, we analyzed all SARS-CoV-2 PCR test results, prescriptions for anti-infective agents, and hospitalizations with COVID-19 or other infections in Denmark from March 2020 to June 2023, resulting in up to 40 months of follow-up.”

The following is added to the Strengths section of the Discussion:

“Thirdly, by comparing multiple infectious diseases, the study highlights the differences between COVID-19 and other infections.”

Other comments - arranged in the order of the manuscript:

● **Line 32:** The text mentions four disease types but reports results for only two (lines 33-34). Could you clarify the reasoning behind this choice? Are the others not considered as relevant in this context?

Authors' reply: In our paper, general medical conditions include both neurological and neuropsychiatric symptoms. We chose to explicitly write them in the abstract as they are of high interest to many researchers but see how this can be confusing to the reader. To keep consistency, we have rephrased the sentence and now only mention “*mental disorders and general medical conditions*”.

● **Line 45:** I wasn't able to locate the source of the 10-20% figure in the referenced commentary. It might strengthen the argument to provide a more solid empirical backing for this number.

Authors' reply: The 10-20% is mentioned in lines 10-11 in the referenced commentary: “... *long COVID is estimated to occur in 10-20% of cases...*”. We have added an additional reference to this, in which 12.7% of patients continued to experience at least one significant symptom 90-150 days after their COVID-19 diagnosis (Ballering, A. V., S. K. R. van Zon, T. C. Olde Hartman and J. G. M. Rosmalen (2022). “*Persistence of somatic symptoms after COVID-19 in the Netherlands: an observational cohort study.*” *Lancet* 400(10350): 452-461).

● **Line 47:** The introduction of CRP feels somewhat abrupt. Providing a bit more background would help contextualize its relevance. More broadly, the discussion of CPR seems somewhat detached from the main focus—consider whether it is essential to the paper.

Authors' reply: We have revised the introduction to better motivate the analysis of CRP.

● **Line 50:** Compared to what? Clarifying the reference point would improve readability.

Authors' reply: The reviewer has raised an excellent point, and we sincerely appreciate the opportunity to address it. We agree that explicitly stating the reference group is crucial for a clear understanding of the background literature. A key challenge in the current literature is the inconsistent use of control groups in prior studies. We had already highlighted and critiqued this in the Introduction, which has now been rephrased:

“Prior studies showed various results depending on the time of follow-up, admission status of COVID-19 patients, and whether the control group was individuals with negative tests, no positive test, or various non-COVID-19 infections. Large scale studies and meta-analyses found that individuals with COVID-19 compared with various control groups had higher risks of numerous adverse health outcomes.”

● **Lines 50-71:** This section reads more like a discussion rather than a framing of the study. It could benefit from a sharper focus, with some of the more detailed content moved to the discussion section.

Authors’ reply: We have revised the introduction in relation to this and main comment 1.

● **Lines 73-81:** The section would benefit from a clearer focus and a more structured presentation of the study’s overall aims. As written, the argument could flow more logically to help guide the reader.

Authors’ reply: We thank the reviewer for the suggestion and have revised the introduction accordingly.

● **Line 75:** Compared to what? Each other or no tests? Clarification would help the reader follow the argument.

Authors’ reply: We have rephrased the sentence: *“We compared individuals with positive, negative and no SARS-CoV-2 PCR test results to each other...”*.

● **General:** The introduction would benefit from a clearer structure, with a stronger motivation and a more logical buildup to the research questions. Right now, it leans too much toward discussion rather than setting the stage for the study. And the final paragraph could more explicitly outline what follows.

Authors’ reply: We have revised the introduction cf. the reviewer’s previous comments in order to setting the stage for the study.

● **Lines 84-85:** Since the Methods section is placed at the end, it would be helpful to briefly introduce what is meant by “mental disorders or general medical conditions.”

Authors’ reply: The reviewer’s suggestion is highly appreciated, and we have revised accordingly:

“... mental disorders (including mental and behavioral disorders due to psychoactive substance use, psychotic, mood, and anxiety disorders) and ... general medical conditions (including selected neurological, respiratory, gastrointestinal, circulatory, kidney, endocrine, hematological, musculoskeletal, and dermatological disorders, as well as neuropsychiatric symptoms) ...”

● **Lines 94-103:** These are very interesting results.

Authors’ reply: We appreciate the praise of our work.

● **Line 100:** The reference to an “age effect” might not be necessary. Also, “effect” may not be the most suitable term here—consider rewording if you decide to keep this point.

Authors' reply: We have rephrased the sentence in the relevant paragraph: *“The rates of mental disorder rates did not depend on the age group.”*

● **Lines 110-126:** The shift in comparison groups (from not-tested to negative tests) in the HRs is confusing for the reader. It could help to make this transition clearer. One possible approach is to explicitly state the comparison at the beginning of the sentence. For example, on line 110: *“Compared to individuals with a negative SARS-CoV-2 test, the number of medical conditions...”*

Authors' reply: We have revised according to the reviewer's comment and now state the comparison in the beginning of the sentence.

● **Line 127:** A new subsection heading here might help distinguish general medical conditions from the following content.

Authors' reply: We have added a subheading to clarify that this section focuses on specific diagnoses: *“SARS-CoV-2 positive test results and the rates of specific general medical conditions”*.

● **Lines 158-178:** Very interesting results.

Authors' reply: This comment is highly appreciated.

● **Line 182:** Consider rewording for clarity: *“Compared to individuals with negative SARS-CoV-2 tests, positive tests were associated with lower rates of mental disorder for all SARS-CoV-2 variants...”*

● **Line 191:** Same suggestion as for line 182.

Authors' reply: We have revised according to the reviewer's comments.

● **Line 218:** It looks like a word might be missing here—should it be *“assumed”*?

Authors' reply: We have reworded for clarity: *“This could also explain why prior studies that were conducted earlier in the pandemic estimated higher risks than in our study.”*

● **General note on discussion:** What are the implications of negative tests being associated with higher rates of mental illness than positive tests? Could this suggest selection effects in testing behavior? It would be helpful to explore this further.

Authors' reply: In our opinion, testing behavior could explain these contra intuitive findings that confirm our previous work (*Nersesjan V, Christensen RHB, Kondziella D, Benros ME. COVID-19 and Risk for Mental Disorders Among Adults in Denmark. JAMA psychiatry, 2023;80(8):778–86.*). In that study, we discuss some of the potential mechanisms, and we have now added these to the Limitations paragraph of the Discussion:

“Last, testing behavior, such as incentive to get a SARS-CoV-2 test, was not captured in the registers, and could explain why positive tests compared with negative tests was associated with lower rates of mental disorders.”

Moreover, we have added sensitivity analyses on the number of tests conducted and associations with rates of long-term sequelae.

● **Line 229:** If 92% of individuals were tested, does that mean the “non-tested” comparison group comprises only 8% of the sample? Would it be possible to conduct statistical tests on selection into different groups (testing positive, testing negative, and not testing)? Are there notable differences based on observables and disease history? Additionally, since there were more negative tests early in the pandemic (as seen in Fig. 5), does this affect the composition of the groups?

Authors’ reply: The exposure is time-varying, causing the composition of the groups to change throughout the pandemic. We have added this clarification to the Abstract:

“We used Cox Proportional Hazards models with time-varying exposures to estimate the rates of mental disorders and general medical conditions.”

● **Table 1:** There may be a typographical error in the number of admissions in the readmissions analysis—should it be 1,064 instead of 10,686?

Authors’ reply: The reviewer is absolutely correct, and we appreciate them pointing out this typo, which we have now corrected.

Reviewer #3 (Remarks to the Author):

Authors’ reply: We thank the reviewer for their feedback and support the initiative.

Reviewer #4 (Remarks to the Author):

Review comments: COVID-19 and all post-infection sequelae on a nationwide scale with 40-months follow-up

As indicated in the title, this paper analyses post-infection sequelae on a nation-wide scale with 40-months follow. The paper uses a wide range of Danish health registers spanning the whole population in the period March 2020 – June 2023. Outcomes are mental, neurological and general medical condition as well as neuropsychiatric conditions. The authors follow the Danish population as a cohort starting with all persons being not tested for Sars-CoV-19 where after they can change status to negative and/or positive based on PCR tests. Both those not tested, test-negatives and test-positives are compared. Cox proportional hazard models are used.

The authors find that those tested positive for SARS-CoV-2 had lower hazard ratios (HR) of mental disorders while hospitalized Covid-19 patients had a very slight increased hazard ratio (HR 1.01) of all disorders except for a few compared with the general population, but that this elevated HR was not different from that following other infections. The increased HR for test-positives was modified by SARS-CoV-2 variant and vaccination status.

Overall impression

The overall impression of the paper is that this is an extremely comprehensive paper looking at a

wide range of symptoms following a test result, either negative or positive. A wide range of factors that can modify or influence outcomes are taken into account, including hospital admission, readmission, vaccination, co-morbidity (?) and virus variants.

The paper uses Danish high-quality registers yielding information of a very large number of tested persons and their outcomes. Overall, statistical methods are appropriate.

The scale of the study population and the range of the analyses is impressive.

The paper presents interesting data and provides novel findings on the subject of long COVID, in particular regarding outcomes after other infections that have been little studied.

Figures and graphical abstract are well-designed.

Authors' reply: We are truly honored by the reviewer's kind words and positive evaluation of our work, and thankful for the detailed and thoughtful feedback, which led to meaningful revisions.

However, there are four main issues of concern related to this paper: The paper is extremely comprehensive; Discussion is vague and limited; there is much bias in results that is not dealt with nor discussed; and the large sample size allows the paper to draw conclusions with statistical power that has little biological impact.

Authors' reply: We thank the reviewer for these points for further improvements, which we have all addressed in the below, and believed that it has helped strengthening the paper even more.

Comprehensiveness: The paper aims to identify post-infection sequelae related both to mental health issues, general medical conditions and neurological questions. 'General medical conditions' cover most disease groups in the ICD8 and ICD10 diagnosis systems. Thus, the authors wish to identify most sequelae (or differences in occurrence between SARS-CoV-19 test groups), not only in a specified time period (e.g. within one month corresponding to acute Covid or later than that corresponding to Long-Covid), but over a wide time frame from less than 1 month to more than 18 months after test. Not only do the authors estimate HRs between those tested positive and negative, but also between these groups and those not tested. The authors also include number of tests, hospital admissions (no admissions and admissions with and without intensive care), readmissions and length of admissions and exposures. Other exposures or modifying factors are age, CRP (C-reactive protein), vaccinations, and time period/SARS-CoV-2 strain.

The paper is not hypothesis driven, but rather explorative.

Thus, the authors obtain a huge amount of results which is reflected in the volume of supplementary material (81 pages(!)). However, only few of these results are discussed. E.g. the authors describe the study population in detail (supplementary table 2). But these pieces of information are not really used nor discussed in the paper. Any differences between the groups?

On the other hand, so many results cannot be realistically discussed in one paper.

There is no doubt that the paper shows interesting results. Of those, particularly those concerning comparisons with sequelae after other infections than Covid-19 appear interesting. The natural history of SARS-CoV-2 is not fully described, and there are many remaining questions regarding SARS-CoV-2 sequelae. However, there is also, at present, a large volume of papers published on different aspects of Long-Covid, including nation-wide questionnaire-based papers from Denmark that are not quoted. Overall, such papers deal with aspects of Long-Covid rather than the overall picture.

Authors' reply: We thank the reviewer for recognizing that it is a very ambitious paper, and that it is comprehensive to include so many results in one paper, and for stating that there is

no doubt that the paper shows interesting results. We are convinced that it is essential to consider these results simultaneously for a comprehensive understanding.

We have now made major changes to the structure and table of contents in the Supplementary material, and hope that this helps the reader gain overview of all the results. We have now also cited more papers in the introduction, including the questionnaire-based papers as suggested by the reviewer. Regarding if there are differences between the groups, there are some differences; for example, more young people test positive than negative. However, we account for these differences in the analyses by adjusting for relevant confounders.

We respectfully disagree with the comment that this study is exploratory rather than hypothesis driven. The hypotheses are, as stated in the manuscript, to investigate if there are long-term sequelae after SARS-CoV-2, and if it is specific or similar to the long-term sequelae after other infections, which we clearly show with the different comparison groups. We have followed our carefully defined pre-registered study protocol, which is publicly available on OSF. This protocol was developed based on insights from several prior studies on infections and psychiatric/neurological disorders. We thus did not include most or all disease groups from the ICD-8 and ICD-10 diagnosis systems. Instead, we pre-specified outcomes based on previously identified associations in the literature, as outlined in the methods section.

We believe that the additional analyses are both necessary and appropriate for evaluating the severity, temporality, and other key aspects of the research question, ensuring a thorough and robust investigation.

The main impression of this paper is therefore that it would serve as a fine overview paper and starting point for testing of specific hypotheses regarding sequelae of Covid-19, if little was known at this time. The authors rightly point at problems with previous papers that such papers use different comparison groups, different time frames for follow-up, different symptom groupings, etc. Therefore, the authors used the same methodology to look at a range of exposures and outcomes as explained above. But there are also many methodological issues related to this paper, and the question is therefore whether it is too late in the exploration of the natural history of Covid-19 to produce a very high amount of data that are little explained, discussed or put into context.

Authors' reply: We respectfully disagree with the suggestion that there are major methodological issues in the present paper, but we would appreciate it if the reviewer could highlight specific methodological concerns so that we can address them directly. All analyses have been conducted by statisticians including a senior statistician.

In our view, the use of different comparison groups in prior studies (such as individuals not tested versus those with negative tests) raises concerns and blurs the current state-of-the-art on the associations. As demonstrated in our paper, the association between a positive test and mental or general medical conditions is highly dependent on the reference group chosen. Using '*individuals not tested*' as the reference group tends to overestimate the association between COVID-19 and sequelae. We believe that we have emphasized the importance of selecting appropriate comparison groups, providing the overview of the directions of associations comprehensively and on a high-quality nationwide cohort, which have been lacking until now.

Given the rapid evolution of COVID-19, exploring differences in calendar time is crucial for interpreting the associations, which further highlights the urgent need for the present study in

addition to that no previous study have been able to comprehensively display the associations and the specificity of the associations as we were able to due to the unique nationwide Danish registers.

Discussion. This section is very short, just touching upon differences on mental disorders and general medical conditions between those tested for SARS-CoV-2 (positive and negative) and those not tested. But first, the quoted/discussed results are not completely identical to what is written in the abstract nor in the table (in Discussion it says ‘However, positive compared with negative SARS-CoV-2 tests showed no association with mental disorders...’ while in Abstract and Table 1 ‘Compared with negative tests, positive SARS-CoV-2 PCR tests were associated with lower rates of mental disorders (hazard Ratio 0.82, 95% CI 0.80 – 0.83)).

Authors’ reply: To ensure consistency, we have rephrased the Abstract: “Compared with negative tests, positive SARS-CoV-2 PCR tests were not associated with clinically relevant increased rates of mental disorders or general medical conditions” and the Discussion: “However, positive compared with negative SARS-CoV-2 tests were associated with lower rates of mental disorders...”

And second, this difference is not discussed. Why is it beneficial to mental health to being tested positive to SARS-CoV-2 compared to being tested negative?

Authors’ reply: This is a valid point also pointed about by another reviewer, why we have added a paragraph discussing these results in the Discussion. Please, see our response to reviewer 1’s comment 2.

Third, why should the development of mental disorders be explained by CRP? It is not according to results, but why do the authors believe that CRP would explain mental health disorders? This is not discussed. Also, CRP peaks as indicated in Fig. 3 for SARS-CoV-2 positive are in general below 10 mg/ml, but this is within the normal range (<10 mg/ml). Also, a CRP of 800 as marked as end point for CRP graphs (Figure 3) are unseen in daily practice (question of this is a realistic value). In general, the discussion of own results only cover one page, of which half of the page discusses what the authors did differently from other papers.

Authors’ reply: This concern was also stated by another reviewer, why we have added a paragraph discussing the results on CRP in the discussion. Please, see our response to reviewer 1’s comment 4.

CRP is a marker of inflammatory responses and thus the severity of infection, which is why we hypothesized that CRP is associated with sequelae. Based on our analysis, we were not able to dismiss the null hypothesis that CRP is not associated with sequelae.

Note, Figure 3 is now referred to as Extended Data Fig. 3. (see our response to Reviewer 2’s smaller comment 4).

CRP <10 mg/L are common for individuals tested positive for SARS-CoV-2 (left side of Extended Data Fig. 3), meaning that in general non-hospitalized individuals with COVID-19 have low values of CRP. On the other hand, these values are less common for individuals admitted with COVID-19 (right side of Extended Data Fig. 3). The x-axis in Extended Data Fig. 3 is on log-scale, and as seen in the density plot, there is practically no individuals with CRP values around 800 mg/L.

Bias: The authors use the national Danish health registers of high quality. However, there are a

number of biases associated with this use. The Strengths and limitations section includes a list of possible biases related to this study, but the possible biases and their impact on results are not discussed. Thus, being tested positive for SARS-CoV-2 results in a lower HR for mental health issues than those tested negative. This is hard to understand. Could this be a result of bias? Who are being tested multiple times relative to few times?

Authors' reply: We have elaborated on the possible biases for the real-world data including testing patterns and their impact on results by extending the Discussion cf. the reviewer's previous comments.

Also, as the authors rightly point out, all study persons with pre-existing outcome disorders since the start of the registers were excluded, and the study population therefore consisted of very healthy individuals. Why was not paid more attention to this fact in the Discussion, Abstract or elsewhere? This is quite important as the results may not be fully generalizable to the general population because of this bias. In other respects the strength of the paper is that it is population-based and the reader clearly gets the impression that this paper gives a full picture of Covid-19 sequelae in a general population.

Authors' reply: This is a valid point also pointed out by another reviewer and urged us to conduct a sensitivity analysis yielding similar results. Please, see our response to reviewer 1's comment 7.

Sample size and interpretation: Being population-based over several years, the paper includes an impressively high number of study persons and events. This also gives statistical strength. However, the authors underline this too much when they state (Results and Table 1) that 'Positive SARS-Cov-2 PCR tests were associated with only slightly higher general medical condition rates compared with negative SARS-Cov-2 PCR tests (HR 1.01, 95% CI 1.00 – 1.02, p-value 0.024)'. This may be significant (how can it be so when the CI includes 1.00?), but an increased HR of 1% (HR 1.01) has no biological nor public health impact. A HR of 1.01 is essentially the same as a HR of 1.00.

Authors' reply: This is an extremely important comment and we completely agree that the effect is significant but not clinically relevant (as seen by the lower bound of the confidence interval being rounded to 1.00 with two decimal places). We think that we have been honest in our communication of results by writing "... minimally higher rates of general medical conditions...", "... associated with only slightly higher general medical condition rates...", and "... and the associations with general medical conditions displayed only negligible effects". To further highlight the lack of relevant effects we reworded the abstract: "*Compared with negative tests, positive SARS-CoV-2 PCR tests were not associated with clinically relevant increased rates of mental disorders or general medical conditions*" and the first paragraph of the discussion "... and the associations with general medical conditions displayed only minor effects, with no biological nor public health impact."

To sum up – The findings of the paper are so comprehensive, yet subject to bias and little discussed that it is difficult to extract the important findings. An overall paper of all sequelae over a large period of time with many confounders or effect modifiers appears of less use now than some years ago.

A suggestion would be to divide the paper in a number of papers addressing different aspects of Covid-19 sequelae that are then discussed in more detail plus an overview paper describing the common methodology with a clear delimitation of the study population (how large a fraction of the

Danish population would not have a pre-existing comorbidity of the given types) and with a discussion of possible bias.

Authors' reply: As this is the first paper comprehensively investigating COVID-19 sequelae with multiple comparison groups to both show that there are increased risks but not a specificity making it unique as the risks are comparable to the risks after other infections with comparable severity, we believe that it is indeed still very relevant. We of course agree that it would have been even more relevant earlier on in the pandemic, where it would have had a massive societal and health care planning impact immediately, it now gives an overview of the current status and what kind of study should be conducted as early as possible in the next pandemic. The comment about that the study is so comprehensive that it could be published as several studies, is of editorial and not scientific origin, why we leave this decision to the editor. In our opinion, dividing the paper in several papers could lead to unconstructive inflation of the number of published papers, we have followed the prepublished study protocol, and we believe that having it all together provides the much needed overview as a decisive paper on the associations with long-term sequelae.

Specific comments

In its present form, the paper could benefit from improved text structure. It is often challenging to follow the authors' line of thought in each section, and the main points are sometimes unclear. Enhancing the flow of the text would make the content easier to follow and increase the overall readability.

Authors' reply: We appreciate the reviewer's feedback, which has helped enhance both the text and structure of the manuscript. We have incorporated these suggestions to improve the presentation throughout.

Given the impressive size of the paper with supplementary material it is not possible to touch on all parts of the paper including methodology. But the following lists a number of specific points to the paper in its present form in addition the general comments above:

Title

1. Suggest revising the title to make it more informative by clearly stating the paper's main finding.

Authors' reply: As suggested by the reviewer we have changed the title to "*COVID-19 post-infection sequelae are comparable to sequelae observed after other infections of similar severity – a nationwide Danish study with 40-month follow-up.*"

But if the editor prefers we can change it back to the previous title that was more neutral:

"COVID-19 and all post-infection sequelae on a nationwide scale with 40-months follow-up"

Abstract

1. Lines 37–38. The methods and results related to the comparison of sequelae with other infections should be elaborated upon in the abstract, as this represents one of the primary objectives of the paper.

Authors' reply: We appreciate the opportunity to improve the abstract and have added the following sentences: "*Compared with individuals with anti-infective prescriptions and negative SARS-CoV-2 tests, individuals with positive SARS-CoV-2 tests had lower rates of mental disorders and general medical conditions.*" "*However, compared with patients hospitalized with non-COVID-19 pulmonary infections, patients hospitalized with COVID-19 had lower rates of mental disorders or general medical conditions.*"

Introduction

1. Lines 47-48: This statement requires a citation.

Authors' reply: We have now provided citations.

2. Lines 50-51: Is the risk of adverse health outcomes the same for all individuals with COVID-19, regardless of acute illness severity (hospitalized vs. non-hospitalized)? Please elaborate.

Authors' reply: The risk is not the same and we have elaborated on it throughout the manuscript.

3. Lines 59–60: The statement, "In older adults, the sequelae resembled those of viral lower respiratory tract illness," is unclear. Please specify by providing additional context.

Authors' reply: In the revision of the introduction, we have deleted the sentence.

4. Lines 65-67. "However, the findings of prior studies depended on whether the control group was individuals with negative tests, no positive test, or various non COVID-19 infections". Needs a citation.

Authors' reply: We have inserted references. No prior study has investigated it all in one study like ours, so therefore it is reference 4-19.

5. Lines 50-71. This section is informative and well-documented, but the structure could be improved. Adding sentences to clarify the relevance of the information to the paper would enhance readability and overall coherence.

Authors' reply: We have revised the introduction.

Results

1. Consider reporting only the 95% CIs in the results section, as they provide the most relevant measure for assessing statistical significance, making the inclusion of p-values unnecessary.

Authors' reply: According to the author guidelines, p-values are required when referring to findings as 'significant.' We agree that confidence intervals are essential for interpreting the results, particularly given the large sample size. Therefore, we report both confidence intervals and p-values in line with the traditional reporting style.

2. Line 102. What do the authors mean by "depend on time since testing"?

Authors' reply: We have reworded for readability: "The mental disorder rates did not significantly increase with the number of SARS-CoV-2 reinfections and were decreased during all periods of time since testing."

3. Lines 114-116. "Temporally, the general medical condition rates were highest 1-month post-COVID-19 (HR 115 1.10, 95% CI 1.06-1.13, p-value <0.001) and remained elevated 18 months post-COVID-19 (HR 116 1.07, 95% CI 1.04-1.10, p-value <0.001)". Which groups are you comparing? Are the rates higher among those individuals who tested positive compared to those who tested negative?

Authors' reply: We compare individuals with positive SARS-CoV-2 tests with individuals with negative SARS-CoV-2 tests and have therefore added this to the sentence: *"for positive compared with negative SARS-CoV-2 tests"*.

4. Lines 120-122. "The HR of general medical conditions was flat for CRP values in the age groups younger than 40 years and increasing in increasing CRP for the age groups between 40 and 79 years". This sentence is unclear. Please clarify what is meant.

Authors' reply: We have now included the CRP findings together in a section under sensitivity analyses for more clarity."

Discussion

1. Lines 209-211. It would be relevant to mention that not all previous studies focused solely on hospital contacts.

Authors' reply: We have included the references of three survey-based studies examining self-reported symptoms that used PCR tests taken out of the hospital. Please, also see our response to reviewer 1's comment 3.

2. Lines 216-218. "Sensitivity analyses revealed that, in the current situation with widespread vaccination and dominant Omicron subtypes, COVID-19 poses a lower risk to humans than previously". Do the authors mean a lower risk of acute or post-acute symptoms, and does it include all symptoms?

Authors' reply: We have modified for clarity.

3. The main section of the discussion focuses mainly on listing the key findings of the paper, which is appropriate. However, the authors do not sufficiently discuss or contextualize their findings in relation to other studies on long COVID, aside from a single Danish study. This comparison with broader research could provide more depth and perspective, and would strengthen the discussion by relating their results to the existing evidence in the literature.

Authors' reply: We appreciate the reviewer's insightful comment, which was also mentioned by the other reviewers. We have now expanded the discussion to better contextualize our findings in relation to existing studies, including a wider range of literature on long COVID beyond the single Danish study mentioned.

4. Line 235. Why did the authors choose to exclude sequelae treated in primary care?

Authors' reply: The reason for not including sequelae treated in primary care is that this information is unfortunately not available in the Danish registers. We have rephrased: *"Second, we only included diagnoses made in inpatient or outpatient settings, including emergency room visits, since information on sequelae treated in primary care or symptoms not requiring professional care was not available in the Danish registers"*.

5. Line 247. What other infections? Please give examples.

Authors' reply: We have added some examples: *"Sequelae are not exclusive to COVID-19 survivors; they also occur after other infectious diseases (for example influenza and bacterial pneumonia) of similar severity, as measured by admission status."*

6. Lines 247-248. “However, the unique data on COVID-19 can help broaden our understanding of infections and other disorders in general”. This is a valid point however; this aspect is not even mentioned in the general discussion.

Authors’ reply: We have added a section in the discussion about the difference between COVID-19 and other infections. Please, see our response to reviewer 2’s main comment 2.

Methods

1. Did the authors have access to diagnoses registered in the primary care sector? If no, please describe.

Authors’ reply: This information is not available in the Danish registers cf. the previous comment.

2. Line 296. Please explain why the authors chose to adjust for those specific confounders.

Authors’ reply: We chose to adjust for the confounders sex, socioeconomic status (measured by employment status, income quantile, and educational attainment level), medical susceptibility to disease (measured by the Charlson Comorbidity Index), and parental medical history (measured by parental Charlson Comorbidity Index and parental psychiatric diagnosis). These confounders were selected because we believe they are directly related with both the likelihood of contracting COVID-19 and the risk of developing mental disorders or general medical conditions. We have now elaborated on this point in the referenced section of the text:

“Sex, socioeconomic status (employment status, income quantile, and educational attainment level),^{33,58} medical susceptibility to disease (Charlson Comorbidity Index (CCI) (Supplementary Table 26)),⁵⁹ parental medical history (parental CCI, and parental psychiatric diagnosis).”

Figures and tables

1. Figure 3 contains a lot of information, making it difficult to read. Suggest to select only a few key specific disorders to present. Additionally, expanding the figure caption with more details would improve understanding.

Authors’ reply: Please, see our response to Reviewer 2’s comment 4.

2. Supplementary Table 20 should be moved to the main paper, as it provides important context and details on the specific diagnoses examined, which are valuable for understanding the study's results.

Authors’ reply: To include the important comparison between COVID-19 and other infections, we have altered the former Figure 4, now Figure 3. For details, please see our response to reviewer 2’s main comment 3.

Reviewer #5 (Remarks to the Author):

Authors’ reply: We thank the reviewer for their feedback and support the initiative.

Point by point response to the reviewers

Manuscript #: NCOMMS-24-86598-A

Title: COVID-19 post-infection sequelae are comparable to sequelae observed after other infections of similar severity – a nationwide Danish study with 40-month follow-up

REVIEWER COMMENTS

Reviewer #1 (Remarks to the Author):

I would like to thank the authors for very thorough responses to my comments related to their manuscript "COVID-19 post-infection sequelae are comparable to sequelae observed after other infections of similar severity – a nationwide Danish study with 40-month follow-up". I have no further comments and I recommend this manuscript to be accepted for publication.

Authors' reply: We thank the reviewer for the positive evaluation of our revised manuscript and for the evaluation that it was ready for publication after the first round of revision.

Reviewer #2 (Remarks to the Author):

The authors have done several substantial improvements to the paper since the last version. For example the introduction reads better, with a clearer focus. Also, the CRP-results are better placed. All in all, the paper presents much better.

Authors' reply: We thank the reviewer for acknowledging that the revised manuscript is substantially improved regarding the main points from the reviewers in the first review.

Still, there are issues that are not fully resolved.

General comment

The topic of post-Covid remains a contentious issue, both within and outside of academia. In this context, the paper has the potential to make a significant contribution, but it may also attract intense scrutiny. One such point of contention could arise from the finding that positive Covid test results are associated with lower levels of mental health issues compared to negative tests. This conclusion is difficult to accept and may lead skeptical readers to question the whole endeavour, and whether there is an underlying bias in the selected analytical methods. Furthermore, given the numerous comparisons made throughout the study, this particular finding could be perceived as a statistical artifact rather than a reliable result. While the authors briefly address the limitations related to mental illness and potential selection bias in testing, a mere sentence may not sufficiently alleviate the concerns of more critical readers. Important note: My reading of the paper is that there are no important differences in COVID-19 post-infection sequelae, both when comparing testing positive/negative, and hospitalization compared to hospitalization for other infections. If my reading is correct, the worry is that this important result could get lost by results that really are not valid due to sample composition issues or multiple testing (example: lower rates of mental disorder among those testing positive). I recommend a revision that presents results in ways that reduces the risk of the contribution being undermined by such worries/criticisms. Therefore, I strongly encourage the

authors to ensure that the main findings are robust, and the issues outlined below represent my strongest concerns.

Authors' reply: Through scrutiny and multiple sensitivity analyses, we have ensured that the main findings are robust. Moreover, in the revised version, we have further aimed to improve the presentation of the findings to reduce the risk of misinterpretation and to make it even clearer to the reader that there were no important differences in post-infection sequelae, both when comparing testing positive/negative, and hospitalization compared to hospitalization for other infections. This includes emphasizing that a positive SARS-CoV-2 test is not associated with a higher risk of mental disorders than a negative test, making the message clearer while maintaining accurate terminology. Moreover, we have highlighted the impact of testing behavior regarding the association with mental disorders, although this does not affect the overall conclusion of the paper.

Focus of presentation

While there has been noticeable improvement in the clarity of the writing, the extensive amount of results and supplementary material still makes the paper challenging to evaluate and comprehend. Streamlining both the main manuscript and the supplementary content would greatly enhance its accessibility and facilitate a clearer understanding for readers.

Authors' reply: We have structured the results in both the main manuscript and in the supplementary materials in the following order for ease of reference:

First:

- **Positive test**
- **Positive test vs anti-infective prescriptions**

Second:

- **COVID-19 admission**
- **COVID-19 admission vs other infections**

Third:

- **Sensitivity analyses for positive test**
- **Sensitivity analyses for COVID-19 admission**

We further consider the comprehensiveness of the paper to be a strength, particularly as we adhered to the pre-published analysis plan to provide the reader with a complete and transparent presentation of the findings. Although a comprehensive presentation may complicate the narrative, we believe it ultimately provides a more accurate reflection of reality.

Selection into testing

This issue was also brought up in the previous report. Thank you for referring to the Supplementary Table 2 showing descriptives for the different groups. (Also, Extended Data Figure 1 and Sensitivity analysis 3 was useful for understanding this topic.) However, the response offered was insufficient. There is a need for a more explicit discussion on how the selection into testing may introduce bias into the results, and to what extent this is an issue. For example, see Extended Data Figure 1: the large part of positive tests are at a very late stage of the pandemic when the Omicron virus was active and societal restrictions were lifted. This can be illustrated from "Sensitivity analysis 5: Lockdown" (page 63): For mental disorders, approxi-

mately 90% of the positive tested group are from the “Post lockdown” period. Hence, if I understand correctly, the overall estimate is almost entirely driven by this period. Are there confounders (and observable imbalance) related to who were not testing positive until the pandemic was over? I find it hard to believe there was not. Maybe more healthy and educated individuals were able to shield themselves better during the pandemic, or more healthy individuals were less afraid to expose themselves to infection risks? Were not almost all individuals vaccinated at this point? I challenge the authors to take this issue more seriously.

Authors’ reply: We understand this comment as addressing selection into treatment (which in our analyses is testing) which may lead to *confounding bias* in the terminology of the causal inference literature (cf. section 8.3 of https://mi-guelhernan.org/s/hernanrobins_WhatIf_2jan25.pdf). We have responded point by point below and elaborated on this in the fourth paragraph of the Discussion section, starting with:

“In Denmark, PCR tests were available from the beginning of the pandemic...”

Regarding the reviewer’s question whether the overall estimate was driven by the post lockdown period: The time-averaged overall HR can be viewed as a weighted average of the time-specific HRs (see Supplementary Table 3) using the number of cases as weights. While the number of cases of positive tests occur late in the period (mostly omicron), the number of cases for negative cases is much more uniform with a median around June 2021. The overall HR is therefore most representative of the time around the beginning of 2022 and not for a post-lockdown period as mentioned by the reviewer. The time-specific HRs are illustrated in Figure 2 (formerly Figure 4). Details regarding the proportional hazard assumption are provided in the response to the next comment, where we have added a paragraph in the Results section describing the time-varying effects.

Regarding the reviewer’s question about covariates related to the chance of avoiding a positive test until the end of the pandemic: Our analyses consider testing behavior (no testing, testing negative, or testing positive) as the exposure. Consequently, we are not able to model testing behavior per se. We include covariates in our Cox regression models because we assume that they act as confounders, i.e., covariates associated with both the exposure and the outcome. Thus, only variables directly associated with the outcome are included. Adjusting for variables that are only weakly associated with the outcome confers a risk of inducing bias (Judea Pearl, Invited Commentary: Understanding Bias Amplification, *American Journal of Epidemiology*, Volume 174, Issue 11, 1 December 2011, Pages 1223–1227, <https://doi.org/10.1093/aje/kwr352>). The reviewers specifically mention health status and education as potential confounders, and we note that we have adjusted for Charlson’s comorbidity index and education level in the analyses.

Regarding the reviewer’s remark about selection into testing: Another aspect is that selection into testing is of little importance when comparing negative and positive tests (i.e., exposures). While behavior may affect the decision to be tested in the first place, this aspect is minimal once interpretation is based on analyses that compare those who were tested; that is, in the comparison of negative and positive tests. Potential biases that might arise when comparing positive and negative tests are discussed in detail (more than 15 lines) in the Discussion section starting with the sentence “The less elevated rates

of mental disorders observed among individuals with positive compared to negative tests could have several explanations...”.

Change in group composition

This issue was also brought up in the previous report, and is closely related to the previous comment. The concern raised in the previous report was that group composition changes substantially over time, which complicates interpretation of hazard ratios. The authors respond that: Importantly, however, the same individuals cannot be in the different groups at the same time. In the Cox proportional hazards model, hazard rates are compared at specific time points, thus not comparing the same individual to itself. And, in the manuscript: We identified the test results in a hierarchical time-varying manner, where initially everyone was in the no test group and when tested moved to the negative or positive group. Subsequent negative tests were ignored in the positive group. With this definition, the same individuals could not be in different groups at the same time. Thank you for this clarification. But, the change in group composition over time could undermine the proportional hazards assumption. Specifically, the Cox model assumes that the hazard ratio between groups is constant over time — an assumption that may not hold if individuals with differing risk profiles enter or exit groups at different times. I believe this can to some extent be informed by “Sensitivity analysis 5: Lock-down” and “Sensitivity analysis 3 (secondary outcomes): Calendar period”, but a more explicit/formal treatment/discussion of the implications of changing sampling composition and the potential violation of the proportional hazards assumption is needed.

Authors’ reply: We thank the reviewer for this question regarding the proportional hazards (PH) assumption, which guided us to emphasize in the revised manuscript that the time-varying hazard ratio is a key finding.

Time-varying exposures: The use of time-varying exposures is a standard approach with the Cox model, which causes the change of group composition over time in our study. Time-varying exposures are a simple and well-understood version of a multi-state Cox model, and time-varying exposures do not make the model more or less sensitive to violations of the PH assumption (Therneou, T.M., and Grambsch, P-M. (2000). Modeling survival data: Extending the Cox model. Springer) (Andersen, P.K., and Ravn, H. (2023). Models for Multi-State Survival Data: Rates, Risks, and Pseudo-Values. Chapman & Hall/CRC). This is distinct from time-varying coefficients, since a coefficient that varies significantly with time represents a deviation from the PH assumption for that variable (Therneou, T.M., and Grambsch, P-M. (2000). Modeling survival data: Extending the Cox model. Springer, New York. (Page 16 of this online resource by Therneou <https://cran.r-project.org/web/packages/survival/vignettes/timedep.pdf>).

Time-averaged hazard ratios: The PH assumption implies that the hazard ratios (HRs) remain constant over the chosen time scale in the Cox model. In our analyses, we use calendar time as the underlying time scale to build robustness towards fluctuations and abrupt changes due to lockdowns or behavioral changes (please see (Clayton, D., and Hills, M. (1993). Statistical Models in Epidemiology. Oxford Science Publications.) Chapter 24 for a discussion of baseline time scale choice). For our models, the PH as-

assumption implies that the HR is constant over calendar time. In practice, this assumption is unlikely to be fully satisfied (Stensrud MJ, Hernán MA. Why Test for Proportional Hazards? JAMA. 2020;323(14):1401–1402. doi:10.1001/jama.2020.1267).

Therefore, the reported HRs presented in, for example Table 1 and the main text, should be interpreted as time-averaged estimates over the study period.

Time-varying hazard ratios: Due to availability of substantial amounts of data, we were successful in modelling the time-varying nature of the HRs explicitly for different exposures and outcomes. This is shown in Figure 2 (formerly Figure 4) and in several supplementary tables. Importantly, the time-averaged HRs presented in Table 1 remain valid, useful single-number summaries of the change in rates averaged over the study period, while the time-varying HRs presented in Fig. 2 (formerly Fig. 4) provide a more nuanced view of how these associations evolve over time.

To highlight the time-varying HRs in the manuscript, we have moved Figure 4 to Figure 2, and we have inserted the following in the Results section:

“SARS-CoV-2 positive test results and time-varying effects

Compared with negative SARS-CoV-2 tests, the rates of first mental disorders and general medical conditions after a positive SARS-CoV-2 test changed over the study period as illustrated by the time-varying hazard ratios in Fig. 2 (Supplementary Table 10). The time-varying hazard ratios show that compared to negative SARS-CoV-2 tests, the rates of mental disorders were consistently lower throughout the study period while the rates of general medical conditions were initially lower but then elevated through most of 2021 for individuals with positive SARS-CoV-2 tests.”

“Hospitalization with COVID-19 and time-varying effects

Compared to non-hospitalized individuals, the rates of mental disorders and general medical conditions for patients admitted with COVID-19 varied over the study period as illustrated by the time-varying hazard ratios in Fig 2 (Supplementary Table 16). The effect of hospital admission with COVID-19 on sequelae were highest around the turn of the years 2020 and 2021.”

We also inserted the following in the Methods section:

“We investigated time-varying effects by dividing calendar time into 2-month intervals and using stratified analyses to allow different baseline hazards and different hazard ratios in each interval.”

We have elaborated on this in the following sentence in the discussion section:

“Our analyses showed that the hazard ratios of sequelae varied over time and that, in the current situation with widespread vaccination and dominant Omicron subtypes, COVID-19 poses a lower risk of post-acute sequelae than in the beginning of the pandemic.”

Comparison with other infectious diseases

Almost the entire time-contribution to the test-positive group is vaccinated against Covid at least once. This surely complicates the comparison with other infectious diseases. How are

these groups defined with regards to overlap, i.e. if a person has a positive test and a prescription to an anti-infective.

Authors' reply:

We acknowledge that the number of events (which determine how the resulting HR is weighted) for both positive and negative exposures largely occurs in a population that has been offered at least one vaccination. The time-varying effects shown in Fig. 2 illustrate how these effects evolve over the study period, during which vaccination programs and circulating virus types change the underlying conditions and influence outcome rates. The intensity of other infectious diseases also varies throughout the study period, as do influenza vaccination programs, which are offered free of charge in Denmark to all susceptible individuals, including those aged 65 years and older.

We thank the reviewer for highlighting that the group composition when comparing to other infectious diseases was not clear. For the comparison with other infectious diseases, we have added the following to the "In the secondary analyses:" paragraph in the Methods section:

"All individuals with a positive SARS-CoV-2 test result were assigned to the positive group, regardless of whether they had prescriptions for anti-infective agents."

"All individuals admitted to the hospital with COVID-19 were included in the COVID-19 admission group, regardless of whether they were also admitted for other infectious diseases."

Evaluation of outcome

Primary care, including visits to general practitioners, is not part of the outcome. This is mentioned in limitations, but I would like a more thorough discussion of its implication.

Authors' reply: We agree that this issue could be discussed more thoroughly and have therefore expanded the discussion of the implications of not being able to include primary care visits as an outcome. The revised text, added to limitations in the Discussion section, now reads as follows:

"Second, we could only include outcome diagnoses made in inpatient or outpatient settings, including emergency room visits, since information on sequelae treated in primary care or symptoms not requiring professional care was not available in the Danish registers. Therefore, we might only capture the more severe end of the spectrum of sequelae that required inpatient or outpatient treatment, while differential effects of predominantly mild sequelae of COVID-19 or other infectious diseases were not detected by our analyses."

More specific comments:

Line 90: Suggest replacing with something like: "Last, to, we explored the role of CRP..."

Authors' reply: We have accommodated this in the revised version

Line 122: Also suggested in the previous round, I suggest to start these sentences with the comparator to ease comprehension: "Compared to individuals not tested,...."

Authors' reply: We have accommodated this in the revised version when appropriate, although it also tends to lengthen the text.

Line 127-128: This sounds very strange. How can this pattern be explained?

Authors' reply: We agree that this may sound counter-intuitive. The pattern presented in the manuscript could potentially be explained by the fact that compared to younger age groups, 80+ year-olds are more affected by severe conditions other than SARS-CoV-2 infection.

Line 143: Why the sudden shift to another comparator (negative test)? In the previous parts the non-tested were the reference.

Authors' reply: The shift to the comparator "negative tests" is not in line 143, it is in the top of the previous section. We have changed the sentence to make the change of comparator clearer, and it now reads:

"However, when compared with negative SARS-CoV-2 tests, positive SARS-CoV-2 tests were not overall associated with clinically relevant differences (HR 1.01, 95% CI 1.00-1.02, p-value 0.024)."

Line 212-214: Compared to what?

Authors' reply: The comparator is the general population, i.e., everyone with a positive or negative test and no admission to the hospital with COVID-19. The comparison is now clarified in the top of the following section

"Hospitalization with COVID-19 and rates of long-term mental disorders or general medical conditions

Compared with non-hospitalized individuals, comprising both those with positive and negative SARS-CoV-2 tests, hospitalized COVID-19 patients had elevated rates of mental disorders (HR 1.88, 95% CI 1.75-2.03, p-value <0.001) and general medical conditions (HR 2.53, 95% CI 2.40-2.67, p-value <0.001) (Table 1 and Supplementary Table 12)."

Also, we have revised and improved the sentence referred to by the reviewer such that it now reads:

"Compared with the general population, the rates of mental disorders and general medical conditions were increased for all virus-types among individuals admitted to the hospital with COVID-19 (Fig. 2, Sensitivity analysis 8-9)."

Line 249-251: Seems contradictory to line 188-192. Please explain better or correct.

Authors' reply: It is not contradictory, and we have revised the first sentence referred to by the reviewer in the revised version:

"Rates of mental disorders and general medical conditions were comparably increased for individuals hospitalized for COVID-19 and individuals hospitalized for other infections."

Line 293-294: This first explanation was not immediate - please be more explicit about the logic.

Authors' reply: We have accommodated this in the revised version:

"First, in the clinical treatment of patients with COVID-19, examination for general medical conditions may be prioritized before mental disorders were considered."

Line 326-327: Suggest changing to “...could explain why positive tests was associated with lower rates of mental disorders compared with negative tests.”

Authors’ reply: We have improved the sentence in question in the revised version:

"Sixth, testing behavior, such as incentives to get a SARS-CoV-2 test, were not captured in the registers and therefore not included in the analyses, although we demonstrated that testing behavior, as captured by the number of SARS-CoV-2 tests, did impact the associations with mental disorders."

Line 330-332: Confusing in light of line 188-192 (and 249-251). Was there higher levels for general medical conditions or not? The credibility of significant the increase in general medical condition is weak, given the many tests presented, still consistency across different parts of the manuscript is key. If this is based on a misunderstanding, clarity of presentation is key

Authors’ reply: We thank the reviewer for highlighting this and have made it more clear in the revised version:

"COVID-19 infection requiring hospitalization was associated with increased rates of sequelae compared to individuals not hospitalized. However, the rates of sequelae after COVID-19 requiring hospitalization were comparable to those observed after hospitalization for non-COVID-19 infections. Moreover, compared with negative SARS-CoV-2 tests, individuals with a positive SARS-CoV-2 test did not display rates of sequelae that were elevated to a clinically relevant extent."

We have also aligned it with the description in the abstract:

“However, when comparing patients hospitalized with COVID-19 to patients hospitalized with non-COVID-19 pulmonary infections or other infections, the rates of mental disorders or general medical conditions were increased to the same extent.”

Reviewer #3 (Remarks to the Author):

Authors’ reply: We thank the reviewer for co-reviewing our work and have responded above.

Reviewer #4 (Remarks to the Author):

Comments to revised version of paper entitled ‘COVID-19 post-infection sequelae are comparable to sequelae observed after other infections of similar severity – a nationwide Danish study with 40-month follow-up’

Overall impression

The authors have made significant improvements to the paper, enhancing its readability and overall flow.

It is clearly an improvement of the paper that the title has changed to highlight what the authors believe is the main conclusion of the paper, namely that symptoms after Covid-19 are not different from those of other similar infections. This has addressed one of the main issues

with the paper as previously indicated, that it addressed many different issues and research questions.

Authors' reply: We thank the reviewer for acknowledging that the revised manuscript is substantially improved as well as providing excellent questions and remarks for this revision.

However, the paper still presents a very comprehensive range of findings, sometimes lacking focus. While the introduction has been greatly improved and states some of the main points, it remains difficult to fully grasp the paper's primary objective. This is also reflected in the abstract, where only one conclusion is listed (see specific comments below).

In our primary review we noted that this study did not appear to be hypothesis driven, but rather explorative. The authors replied that the hypotheses were to investigate if there are long-term sequelae after SARS-CoV-2 and whether these are specific or similar to the long-term sequelae after other infections. However, this is not a hypothesis per se. We believe it would help the reader to fully understand the paper if the Introduction section included a section of what exactly is unclear (in brief) related to Long-covid (types of outcome/diagnose groups, time period after infection, variant, measures of Covid-19 severity etc.) and then list the primary aim (which is done) and then specify secondary aims clearly. As listed now in the Introduction, many exposures, effect modifiers and outcomes are listed in a way that is difficult to grasp.

Authors' reply: As suggested by the reviewer, we have in addition to the already present description of state-of-the-art, and description of primary aim, further specified what is currently unclear as well as secondary aims. We agree with the reviewer that this is a further improvement of the paper:

“Moreover, it is unclear how clinically relevant sequelae of COVID-19 affects individuals not admitted to the hospital with COVID-19 and whether the sequelae following COVID-19 are comparable to those following other infectious diseases.”

“Our secondary objective was to evaluate how these associations compared with those observed after other infectious diseases.”

The Results section is better structured in the revised text. The analyses are undoubtedly of high quality but some findings are described too vaguely, making it challenging to fully understand the key takeaways.

Authors' reply: We are happy that the reviewer finds the revised result section better structured, and that the analyses are of high quality. We have further clarified descriptions that might have been interpreted as vague using helpful comments and suggestions made by the reviewers.

The discussion has also improved, with many main findings now addressed. However, some results and potential biases remain undiscussed. In our primary review we noted that there were major methodological issues related to the paper which the authors in their response asked us to identify. These were listed in the subsequent text of the original review. We do not question the qualifications of the authors nor the involved statisticians and believe that

analyses have been calculated correctly. However, there are, inherently, a number of epidemiological issues regarding mainly bias and interpretation of results as indicated in the primary review that need discussion, and we believe that this has not been fully done according to the specific comment.

Authors' reply: As suggested by the reviewer, we have now, in the discussion section, further elaborated on the potential biases of our study and the potential impact on the interpretation of results. We believe that it thoroughly addresses the concerns raised by the reviewers. We also refer the reviewer to our answers to a previous reviewer's inquiry regarding the statistical methods and how they handle the time-varying structures inherent in epidemiological studies through time-varying exposure and time-varying effects.

Besides this, the generalizability of the findings to other populations should be considered in the discussion. A broader discussion of how this study contributes to the existing body of literature on long COVID is missing, though it is briefly mentioned in the conclusion.

Authors' reply: Following the reviewer's suggestion, we have expanded the discussion on the generalizability of the findings to other populations:

"Last, testing behavior, socioeconomic and demographic factors likely differ between countries, but the biological effects of SARS-CoV-2 infection on the subsequent risk of sequelae are expected, to a large extent, to be comparable between countries."

Furthermore, we have extended the discussion on how this study contributes to the existing body of literature on long COVID.

Thus, given the very comprehensive material, we believe, besides addressing the specific points as indicated, the readability of the text would improve by focus more on specific questions and mainly the question of comparability of outcomes of Covid-19 compared with other infections as indicated in the new title.

Authors' reply: As suggested by the reviewer, we have further focused the manuscript on mainly the question of comparability, in particular the comparison of outcomes related to COVID-19 with those of other infections.

Finally, in the revised version there appear some linguistic and typing errors (e.g. P. 13 L 284 'Sequalae' and P. 3 L. 62 'Studies comparing positive with negative SARS-CoV-2 63 PCR tests among non-hospitalized individuals found higher 6-month post-COVID risks of certain 64 hospital diagnoses, such as venous thromboembolism, [but?] the risk was not higher of serious complications like ischemic stroke, encephalitis, or psychoses'). We suggest that the text should be scrutinized for these.

Authors' reply: We thank the reviewer for pointing these typos out for us, which we have now corrected. Additionally, we have scrutinized the text for linguistic and typing errors and corrected these throughout.

Specific comments

Abstract

1. What is the overall conclusion of the study? Four sentences (lines 35 – 42) list the main findings. But what are the conclusions of those?

Authors' reply: We thank the reviewer for highlighting this and have added the following at the end of the abstract on page 2:

“In conclusion, COVID-19 post-infection sequelae are comparable to sequelae observed after other infections of similar severity.”

2. L. 35: ‘Compared with negative tests, positive SARS-CoV-2 PCR tests were not associated with clinically relevant increased rates of mental disorders or general medical conditions. Rates of general medical conditions after a positive compared with a negative SARS-CoV-2 PCR test were only elevated for virus-types preceding Omicron and for individuals with less than 3 vaccinations.’ Aren’t those two sentences not contradictory to each other?

Authors' reply: We agree with the reviewer that this may cause confusion, and in line with one of the other reviewer's comment, we have changed the two previous lines in the abstract on page 2 to the following:

“Positive SARS-CoV-2 PCR tests alone were not associated with clinically relevant increased rates of mental disorders or general medical conditions when compared with negative SARS-CoV-2 PCR tests, nor when compared to individuals with anti-infective prescriptions.”

3. “Compared with the general population, the rates were most elevated among hospitalized COVID-19 patients, and particularly with ICU treatment”. Rates of what? Needs clarification.

Authors' reply: As suggested by the reviewer we have clarified this in the sentence on page 2, which now reads as the following:

“Compared with the general population, the rates of mental disorders or general medical conditions were elevated among hospitalized COVID-19 patients, and particularly when ICU treatment was required.”

Introduction

1. The introduction has improved significantly; apart from what is written above we have only one minor comment. Line 52: “...which is of utmost importance for healthcare planning and the society.” In what way is it important?

Authors' reply: We have revised the sentence in question to the following:

“Strong epidemiological evidence regarding COVID-19 sequelae and whether they differ from sequelae after other infections is of utmost importance for healthcare planning, including resource allocations, but also for the society through increased understanding of the specificity of COVID-19 sequelae compared to other infections.”

2. Line 50. What other infections? It would be informative to give a few examples.

Authors' reply: We have now listed some examples:

“However, many other infections, such as influenza and pneumonia, can also lead to post-infection sequelae, and the post-COVID complications have yet to be compared comprehensively with sequelae after other infectious diseases.”

Results

1. Are individuals with a prior hospital contact for mental disorders and/or general medical conditions excluded from your study population? It is unclear in the text. If they are excluded, the introduction should state this more clearly to clarify that the results are based on a healthy study population. Consider moving Supplementary Figure 1 to the main paper to give an overview of the study population.

Authors' reply: Yes, we exclude individuals with a prior hospital contact for the outcome diagnosis since we investigate *first-time* diagnoses. We have moved Supplementary Figure 1 to Extended Data Fig. 1, emphasized "*first-time* mental disorders and general medical condition" in the abstract and discussion, and also inserted the following in the limitations paragraph in the Discussion section:

"... we excluded anyone previously diagnosed with any of the outcome disorders at any point since the start of the registers (1968 for mental disorders and 1977 for general medical conditions) to ensure analysis of first-time outcome diagnoses."

2. Line 118: "...were decreased during all periods of time since testing". This sentence is unclear. Please elaborate.

Authors' reply: we agree that the sentence could be improved and have divided it in two:

"The mental disorder rates did not significantly increase with the number of SARS-CoV-2 reinfections (Table 1, Supplementary Table 8). Moreover, there was no significant association between time since testing and mental disorders (Table 1, Supplementary Table 9)."

3. Lines 157-160. This section is unclear. Are you comparing test-positive individuals who had been prescribed an anti-infective drug with test-negative individuals who also had been prescribed an anti-infective drug?

Authors' reply: We compare SARS-CoV-2 positive tests with prescriptions for anti-infective agents to investigate whether COVID-19 is more prone to sequelae than other infections. The details are in the online methods that we refer to in the Methods section: "Analysis on: Positive SARS-CoV-2 test compared with out-of-hospital infections."

We excluded individuals with a prescription for any anti-infective agent between January 2019, and February 2020 to rule out recurring infections. We divided individuals into three exposure groups based on test result and redemption of anti-infective agent status (SARS-CoV-2 negative, prescription for anti-infective agents, SARS-CoV-2 positive). Since the three groups overlapped, we defined them in a time-varying hierarchical manner:

1. **No infection:** Individuals with only negative SARS-CoV-2 tests and without prescriptions for anti-infective agents.
2. **Any prescription for anti-infective agents:** Individuals with only negative SARS-CoV-2 tests and with a prescription for anti-infective agents.
3. **SARS-CoV-2 positive:** Individuals with a positive SARS-CoV-2 test, regardless of whether they have redeemed a prescription for anti-infective agents).

That is, individuals with both a positive SARS-CoV-2 test and a prescription for any anti-infective agent were allocated in the SARS-CoV-2 positive exposure group. “

We have added the following for clarification in the Methods section:

“All individuals with a positive SARS-CoV-2 test result were assigned to the positive group, regardless of whether they had prescriptions for anti-infective agents.”

This comparison ensures a cautious point of reference, as the SARS-CoV-2 positive group is generally more at risk. Given that the rates of sequelae were higher among individuals with anti-infective prescriptions than among individuals with positive SARS-CoV-2 tests, we conclude that the rates of sequelae are not higher after COVID-19 than after other infections.

We have revised the section for clarity:

“Individuals with positive SARS-CoV-2 tests did not have higher rates of mental disorders (HR 0.63, 95% CI 0.62-0.65, p-value <0.001) and general medical conditions (HR 0.72, 95% CI 0.70-0.73, p-value <0.001) compared with individuals with anti-infective prescriptions (and only negative SARS-CoV-2 tests).”

4. Lines 167+170. Compared to COVID-19 infected, non-hospitalized individuals?

Authors’ reply: The non-hospitalized group consists of everyone not admitted to the hospital with COVID-19, i.e., individuals with positive or negative tests. We have altered the sentence to the below:

“Compared with non-hospitalized individuals, comprising both those with positive and negative SARS-CoV-2 tests, hospitalized COVID-19 patients had elevated rates of mental disorders (HR 1.88, 95% CI 1.75-2.03, p-value <0.001) and general medical conditions (HR 2.53, 95% CI 2.40-2.67, p-value <0.001) (Table 1 and Supplementary Table 12).”

5. Lines 199-205. Suggest excluding these results (including Table 2) from the main paper, as they seem redundant. Consider moving them to supplementary materials.

Authors’ reply: These results were moved from the Supplementary to the main paper to accommodate one of the other reviewer’s requests during the first revision. Consequently, we prefer not to exclude these results from the main paper.

6. Lines 222-231. These results are very unclear. It is difficult to determine the exact comparison being made. Which groups are included in the analysis, and what CRP levels are being compared?

Authors’ reply: We have clarified it in the revised version by starting the sentences with ‘compared with’ as requested by another reviewer or ‘among’ to underline the groups included in the analysis. See the paragraph titled “CRP levels and the rates of long-term sequelae” in the Results section.

Discussion

1. It is highlighted that this study was able to include all PCR tests for SARS-CoV-2 conducted in Denmark, which was important for capturing mild/asymptomatic COVID-19 cases,

unlike previous studies. However, since data from the primary care sector was not included in your analysis, there is a significant risk that many individuals in the study population sought medical care for mental disorders and/or general medical conditions in the primary sector rather than in hospitals, which is likely the case for most mild/asymptomatic cases. This may have affected the estimation of rates and represents a possible sampling bias that should be discussed further.

Authors' reply: We agree with the reviewer and have expanded the discussion on this topic in the discussion.

“Therefore, we might only capture the more severe end of the spectrum of sequelae that required inpatient or outpatient treatment, while differential effects of predominantly mild sequelae of COVID-19 or other infectious diseases were not detected by our analyses.”

However, it does not represent a possible sampling (selection) bias, as it is unrelated to how individuals were included in the study population.

2. Lines 281-284. What could be the explanation for CRP levels not being associated with a higher risk of post-infection sequelae in hospitalized patients as opposed to non-hospitalized patients?

Authors' reply: This is a great question that has also puzzled us since it is contradictory to what we expected. A likely explanation is that hospitalized patients being severely ill already have high CRP levels, which limits our ability to detect a relationship with peak CRP levels among hospitalized patients.

3. Lines 286-292. This section should be included in the methods section and not in the discussion.

Authors' reply: We agree that it provides methodological details. However, we wish to keep this information in the Discussion section, as one of the reviewers requested this in the previous revision, because it is essential for interpreting results, especially considering that the Methods section follows the Discussion in this journal.

4. The results regarding the impact of prescribed anti-infective drugs on the HR of post-infection sequelae are not discussed. This should be mentioned, as there are many biases that could have impacted the results.

Authors' reply: We thank the reviewer for pointing this out as this was indeed missing from the discussion. Comparisons between individuals with positive SARS-CoV-2 tests and those with anti-infective prescriptions are somewhat uneven, since testing positive generally indicates less severe acute illness than receiving prescribed treatment. We have added the following to the third paragraph of the Discussion regarding comparisons with other infectious diseases:

“Individuals with prescriptions for anti-infective agents had higher rates of sequelae than those tested positive for SARS-CoV-2, reflecting that anti-infective use may reflect more severe acute illness.”

5. Under strengths and limitations, it should be noted initially that the study did not have access to data from the primary care sector. Although it is stated that the study had complete data, this is not entirely accurate due to the absence of primary care data.

Authors' reply: We see how this could be misleading and have therefore now noted initially that the outcome is not from primary care data:

"..., outcomes (diagnoses in an inpatient or outpatient setting, including emergency room visits, although not from primary care)..."

6. Is it possible that a proportion of the test-negative individuals or those who were never tested had tested positive using a home antigen test, thereby increasing the risk of misclassification bias?

Authors' reply: In the third paragraph of the Methods section, titled "Exposure to SARS-CoV-2", we have described it as follows:

"PCR tests were the best method for detecting SARS-CoV-2,⁴⁰ and we did not include results from other types of tests such as self-testing, as a positive self-test required subsequent confirmation with PCR testing."

Methods

1. The use of CRP measurements from the Register of Laboratory Results for Research should be described in more detail. How were the CRP levels linked to the SARS-CoV-2 test? Did the CRP measurement have to be conducted within a certain time frame relative to the test date to be included in the study? If so, how did you account for the potential time difference between a SARS-CoV-2 test and a measured PCR value?

Authors' reply: The details are in the Supplementary Methods Section but we thank the reviewer for pointing out that this information is necessary to have in the Methods section to understand the analyses. CRP measurements did in fact have to be conducted within a time frame relative to the test date. We have elaborated on this in the paragraph "In the secondary analyses:" of the Methods section:

"... peak CRP levels (see Supplementary Table 30 and Supplementary Methods Section for more details on exposure definitions). CRP was collected from the Register of Laboratory Results for Research covering the large clinical biochemical and immunological laboratories.⁶⁰ When considering SARS-CoV-2 PCR-tests, CRP measurements taken within two weeks (14 days) prior to the test date were included in the analysis, and in the case of admissions with COVID-19, only CRP measurements taken within two days before or during the admission were considered."

Figures and tables

1. Figure 3 is very hard to comprehend, as well as the section describing some of the main results (lines 152-161)

Authors' reply: We thank the reviewer for pointing this out. As suggested by the reviewer we have improved the clarity in these sentences as follows:

"Redeeming a prescription for any anti-infective agent was associated with higher rates of mental disorders (HR 1.41, 95% CI 1.38-1.44, p-value <0.001) and general medical conditions (HR 1.65, 95% CI 1.62-1.67, p-value <0.001) compared to no prescription among individuals with negative SARS-CoV-2 tests (Fig. 4 and Supplementary Table

11). Individuals with positive SARS-CoV-2 tests did not have higher rates of mental disorders (HR 0.63, 95% CI 0.62-0.65, p-value <0.001) and general medical conditions (HR 0.72, 95% CI 0.70-0.73, p-value <0.001) compared with individuals with anti-infective prescriptions (and only negative SARS-CoV-2 tests). These findings were consistent across different types of anti-infectives and specific general medical conditions.”

Reviewer #5 (Remarks to the Author):

Authors' reply: We thank the reviewer for co-reviewing our work and have responded above.

Point by point response to the reviewers
Manuscript #: NCOMMS-24-86598B
Title: Post-infection sequelae of COVID-19 and other infectious diseases – a nationwide Danish study with 40-month follow-up

REVIEWER COMMENTS

Reviewer #2 (Remarks to the Author):

The authors have made several substantive improvements to the paper. Overall, the paper has interesting findings that can contribute meaningfully to the important research on long-term post-acute COVID-19 sequelae.

Still, there are some minor issues that could improve the robustness of the paper if addressed. I leave it to the authors and editor to decide how far to take these revisions.

Authors' response: We thank the reviewer for their careful reading of the revised manuscript and for their constructive and thoughtful comments. We are pleased that the reviewer finds the paper's findings interesting and meaningful for research on long-term post-acute COVID-19 sequelae. Below, we address each of the remaining points in turn.

Selection into testing - changes in group composition. Time-specific HRs are useful, but they do not resolve the concern that there may be selection into testing and that this selection varies over the pandemic period.

Moreover, the claim that selection into testing is “of little importance” for the positive-vs-negative comparison is not justified without strong assumptions (too strong, most likely). Specifically, there are unobservable factors potentially affecting who chooses to test, and who is likely to test positive/negative, and these may vary across the pandemic period. This time-varying selection into testing cannot be fully ruled out and should be explicitly acknowledged as a limitation.

Authors' response: As suggested by the reviewer, we have revised the manuscript to explicitly acknowledge this issue as a potential limitation in the Strengths and Limitations paragraph in the Discussion section:

“Seventh, unobservable factors may influence both testing behavior and test results, and these influences may vary over the course of the pandemic.”

Streamlining the manuscript

Line 172 / paragraph “SARS-CoV-2 positive test results compared with prescriptions for anti-infective agents regarding rates of long-term sequelae.” I recommend moving this paragraph to the Supplementary Material. Even after multiple readings, its purpose and interpretation remain unclear, and it tends to confuse rather than clarify. This edit should not conflict with preregistration. The main text can note that additional preregistered analyses were conducted, while the detailed results and rationale are presented in the Supplementary Material.

Authors' response: We appreciate the reviewer's suggestion to move the paragraph "SARS-CoV-2 positive test results compared with prescriptions for anti-infective agents regarding rates of long-term sequelae" to the Supplementary Material. While we acknowledge that this change could simplify the manuscript, we believe that the paragraph captures an important aspect of the analysis that would be less visible if moved out of the main text. The important aspect is to quantify whether sequelae are specific to COVID-19 or occur to the same extent after other infectious diseases. We therefore propose to retain it in the manuscript as currently presented; however, if the editor prefers, we are open to relocating it to the Supplementary Material.

More generally, this relates to a point raised previously (and also by the other referee): the paper would benefit from a tighter focus. At present, the Supplementary Material risks overwhelming the reader.

Authors' response: We agree that the Supplementary Material is extensive and potentially overwhelming for some readers, which is also why it is supplementary material and not included in the main article, but of value to the interested readers who prefer all the details, which they can then access in the supplementary material.

Line 327/328 "The elevated rates of mental disorders observed among individuals with positive compared to negative tests could have several explanations" should be the other way around, the elevated rates were among the negatively tested.

Authors' response: We thank the reviewer for pointing out this error, and we have now corrected it in the main text:

"The decreased rates of mental disorders observed among individuals with positive compared to negative tests could have several explanations"

Line 46/47 "In conclusion, COVID-19 post-infection sequelae are comparable to sequelae observed after other infections of similar severity." Consider changing to "severe COVID-19 post-infection..." since this analysis is only concerned with in-patient and out-patient sequelae the stated conclusion seems too strong.

In general the fact that this analysis only concerns severe cases of post-infection afflictions should be incorporated more in the analysis.

Authors' response: We agree that the original wording was too strong given the scope of the analysis. We have revised the conclusion to specify "severe COVID-19 post-infection sequelae," thereby aligning the statement more closely with the fact that the analysis concerns inpatient and outpatient sequelae.

Reviewer #3 (Remarks to the Author):

Reviewer #4 (Remarks to the Author):

The authors have submitted a second revised version of the paper, including a rebuttal letter addressing all raised reviewer items point by point.

In general, we find that the authors have sufficiently addressed the raised issues in this second review. Now the aims, methods, results and conclusions of the study stand out quite more clear. I believe the lack of focus for many of the findings have now been explained so the paper now stands more focused.

We have no major comments to the present revision of the paper.

Authors' response: We thank the reviewer for highlighting that the prior revision sufficiently addressed the concerns at that the paper is now clearer. We thank the reviewers for their constructive and helpful comments.

Reviewer #5 (Remarks to the Author):

Referee Report on NCOMMS-24-86598-T

General comment

This nationwide follow-up of long-term post-acute COVID-19 sequelae aims to compare rates of mental, neurological and general medical conditions in people tested positive for COVID-19 with people diagnosed with other infections. The study is based on very interesting and rich data, and has great potential. However, it lacks some clarity and focus in presentation, and there are also methodological issues that should be looked into.

Main comments

- 1) **Motivation and clarity of presentation:** The introduction would benefit from a clear and logical buildup to the study's aims. Now it reads more like a discussion, with scattered arguments and insufficient focus. The last paragraph in particular requires a stronger outline of the study's objectives and the subsequent structure.
- 2) **CRP results:** The CRP results/discussion appears somewhat disconnected from the main focus of the manuscript. This topic may detract from the overall narrative. Consider its necessity, and if retained it needs better placement.
- 3) **Comparison to other infections:** These are very interesting results which should be given more place/attention. We suggest a stronger focus on this comparison in the analysis and discussion. We also suggest cutting some of the secondary analysis as it makes the scope of the paper too wide.
- 4) **Selection into testing:** The manuscript does not sufficiently explain or explore the selection mechanism into PCR-testing (tested/did not test), tested positive, or tested negative. This is important - who was tested, tested positive, and not tested, when? There is a need for descriptive statistics that explore/explain the difference between these groups to evaluate the validity of the comparison.
- 5) **Group composition over time:** The groups - non-tested, positive and negative - change their composition over time (cf. line 306-308)? How to think about comparisons between these groups then? Could the changing composition of the groups introduce biases in the analyses? And, when comparing non-tested to positive, for example, could the same individual in principle be in both groups? Please explain and describe group composition, and under what assumptions the methods used are robust changes in group composition. Also, consider a graphical presentation of the number of individuals in each group over time (x-axis).

Smaller comments:

- 1) **Descriptive statistics - Table 1:** It would be helpful to include a table of descriptive statistics for the analytical sample, broken down by the different groups in the study.
- 2) **Figure 1:** Could you clarify if/where the moderating effects of adjustment variables can be seen? Did results vary by gender, SES, or other observable characteristics?
- 3) **Figure 2:** There are quite a few tests here—are the confidence intervals adjusted for multiple testing? If not, it would be good to explicitly state this.
- 4) **Figure 3:** This figure is somewhat difficult to interpret in its current form. To improve clarity, consider either explaining it more thoroughly or moving it to the

supplementary material. If CPR is not central to the analysis, you might also consider omitting it altogether.

- 5) **Figure 4:** Would it be possible to streamline this figure? Perhaps the most important (panel a) could remain in the main text, with the others moved to the supplementary material.
- 6) **Figure 5:** This figure presents a lot of information—would it be possible to distill the key message into a simpler figure while moving the full version to the supplementary material?
- 7) **Contribution:** The paper already makes a strong contribution, but there may be an opportunity to further highlight its uniqueness—for example, by emphasizing the length of follow-up or comparisons to other infectious diseases.

Other comments - arranged in the order of the manuscript:

- **Line 32:** The text mentions four disease types but reports results for only two (lines 33-34). Could you clarify the reasoning behind this choice? Are the others not considered as relevant in this context?
- **Line 45:** I wasn't able to locate the source of the 10-20% figure in the referenced commentary. It might strengthen the argument to provide a more solid empirical backing for this number.
- **Line 47:** The introduction of CRP feels somewhat abrupt. Providing a bit more background would help contextualize its relevance. More broadly, the discussion of CPR seems somewhat detached from the main focus—consider whether it is essential to the paper.
- **Line 50:** Compared to what? Clarifying the reference point would improve readability.
- **Lines 50-71:** This section reads more like a discussion rather than a framing of the study. It could benefit from a sharper focus, with some of the more detailed content moved to the discussion section.
- **Lines 73-81:** The section would benefit from a clearer focus and a more structured presentation of the study's overall aims. As written, the argument could flow more logically to help guide the reader.
- **Line 75:** Compared to what? Each other or no tests? Clarification would help the reader follow the argument.
- **General:** The introduction would benefit from a clearer structure, with a stronger motivation and a more logical buildup to the research questions. Right now, it leans too much toward discussion rather than setting the stage for the study. And the final paragraph could more explicitly outline what follows.
- **Lines 84-85:** Since the Methods section is placed at the end, it would be helpful to briefly introduce what is meant by "mental disorders or general medical conditions."
- **Lines 94-103:** These are very interesting results.
- **Line 100:** The reference to an "age effect" might not be necessary. Also, "effect" may not be the most suitable term here—consider rewording if you decide to keep this point.
- **Lines 110-126:** The shift in comparison groups (from not-tested to negative tests) in the HRs is confusing for the reader. It could help to make this transition clearer. One possible approach is to explicitly state the comparison at the beginning of the sentence. For example, on line 110: "*Compared to individuals with a negative SARS-CoV-2 test, the number of medical conditions...*"

- **Line 127:** A new subsection heading here might help distinguish general medical conditions from the following content.
- **Lines 158-178:** Very interesting results.
- **Line 182:** Consider rewording for clarity: *“Compared to individuals with negative SARS-CoV-2 tests, positive tests were associated with lower rates of mental disorder for all SARS-CoV-2 variants...”*
- **Line 191:** Same suggestion as for line 182.
- **Line 218:** It looks like a word might be missing here—should it be “*assumed*”?
- **General note on discussion:** What are the implications of negative tests being associated with higher rates of mental illness than positive tests? Could this suggest selection effects in testing behavior? It would be helpful to explore this further.
- **Line 229:** If 92% of individuals were tested, does that mean the “non-tested” comparison group comprises only 8% of the sample? Would it be possible to conduct statistical tests on selection into different groups (testing positive, testing negative, and not testing)? Are there notable differences based on observables and disease history? Additionally, since there were more negative tests early in the pandemic (as seen in Fig. 5), does this affect the composition of the groups?
- **Table 1:** There may be a typographical error in the number of admissions in the readmissions analysis—should it be 1,064 instead of 10,686?

Round 2: Referee Report on NCOMMS-24-86598-T

The authors have done several substantial improvements to the paper since the last version. For example the introduction reads better, with a clearer focus. Also, the CRP-results are better placed. All in all, the paper presents much better. Still, there are issues that are not fully resolved.

General comment

The topic of post-Covid remains a contentious issue, both within and outside of academia. In this context, the paper has the potential to make a significant contribution, but it may also attract intense scrutiny. One such point of contention could arise from the finding that positive Covid test results are associated with lower levels of mental health issues compared to negative tests. This conclusion is difficult to accept and may lead skeptical readers to question the whole endeavour, and whether there is an underlying bias in the selected analytical methods. Furthermore, given the numerous comparisons made throughout the study, this particular finding could be perceived as a statistical artifact rather than a reliable result. While the authors briefly address the limitations related to mental illness and potential selection bias in testing, a mere sentence may not sufficiently alleviate the concerns of more critical readers.

Important note: My reading of the paper is that there are no important differences in COVID-19 post-infection sequelae, both when comparing testing positive/negative, and hospitalization compared to hospitalization for other infections. If my reading is correct, the worry is that this important result could get lost by results that really are not valid due to sample composition issues or multiple testing (example: lower rates of mental disorder among those testing positive). I recommend a revision that presents results in ways that reduces the risk of the contribution being undermined by such worries/criticisms.

Therefore, I strongly encourage the authors to ensure that the main findings are robust, and the issues outlined below represent my strongest concerns.

Focus of presentation

While there has been noticeable improvement in the clarity of the writing, the extensive amount of results and supplementary material still makes the paper challenging to evaluate and comprehend. Streamlining both the main manuscript and the supplementary content would greatly enhance its accessibility and facilitate a clearer understanding for readers.

Selection into testing

This issue was also brought up in the previous report.

Thank you for referring to the Supplementary Table 2 showing descriptives for the different groups. (Also, Extended Data Figure 1 and Sensitivity analysis 3 was useful for understanding this topic.)

However, the response offered was insufficient. There is a need for a more explicit discussion on how the selection into testing may introduce bias into the results, and to what extent this is an issue. For example, see Extended Data Figure 1: the large part of positive tests are at a very late stage of the pandemic when the Omicron virus was active and societal restrictions were lifted.

This can be illustrated from “Sensitivity analysis 5: Lockdown” (page 63): For mental disorders, approximately 90% of the positive tested group are from the “Post lockdown” period. Hence, if I understand correctly, the overall estimate is almost entirely driven by this period. Are there confounders (and observable imbalance) related to who were not testing positive until the pandemic was over? I find it hard to believe there was not. Maybe more healthy and educated individuals were able to shield themselves better during the pandemic, or more healthy individuals were less afraid to expose themselves to infection risks? Were not almost all individuals vaccinated at this point? I challenge the authors to take this issue more seriously.

Change in group composition

This issue was also brought up in the previous report, and is closely related to the previous comment.

The concern raised in the previous report was that group composition changes substantially over time, which complicates interpretation of hazard ratios.

The authors respond that:

Importantly, however, the same individuals cannot be in the different groups at the same time. In the Cox proportional hazards model, hazard rates are compared at specific time points, thus not comparing the same individual to itself.

And, in the manuscript:

We identified the test results in a hierarchical time-varying manner, where initially everyone was in the no test group and when tested moved to the negative or positive group. Subsequent negative tests were ignored in the positive group. With this definition, the same individuals could not be in different groups at the same time.

Thank you for this clarification. But, the change in group composition over time could undermine the proportional hazards assumption. Specifically, the Cox model assumes that the hazard ratio between groups is constant over time — an assumption that may not hold if individuals with differing risk profiles enter or exit groups at different times. I believe this can to some extent be informed by “Sensitivity analysis 5: Lockdown” and “Sensitivity analysis 3 (secondary outcomes): Calendar period”, but a more explicit/formal treatment/discussion of the implications of changing sampling composition and the potential violation of the proportional hazards assumption is needed.

Comparison with other infectious diseases

Almost the entire time-contribution to the test-positive group is vaccinated against Covid at least once. This surely complicates the comparison with other infectious diseases. How are these groups defined with regards to overlap, i.e. if a person has a positive test and a prescription to an anti-infective.

Evaluation of outcome

Primary care, including visits to general practitioners, is not part of the outcome. This is mentioned in limitations, but I would like a more thorough discussion of its implication.

More specific comments:

Line 90: Suggest replacing with something like: “Last, to . . . , we explored the role of CRP...”

Line 122: Also suggested in the previous round, I suggest to start these sentences with the comparator to ease comprehension: “Compared to individuals not tested,....”

Line 127-128: This sounds very strange. How can this pattern be explained?

Line 143: Why the sudden shift to another comparator (negative test)? In the previous parts the non-tested were the reference.

Line 212-214: Compared to what?

Line 249-251: Seems contradictory to line 188-192. Please explain better or correct.

Line 293-294: This first explanation was not immediate - please be more explicit about the logic.

Line 326-327: Suggest changing to “...could explain why positive tests was associated with lower rates of mental disorders compared with negative tests.”

Line 330-332: Confusing in light of line 188-192 (and 249-251). Was there higher levels for general medical conditions or not? The credibility of significant the increase in general medical condition is weak, given the many tests presented, still consistency across different parts of the manuscript is key. If this is based on a misunderstanding, clarity of presentation is key.

Report on NCOMMS-24-86598-T

General comment

The authors have made several substantive improvements to the paper. Overall, the paper has interesting findings that can contribute meaningfully to the important research on long-term post-acute COVID-19 sequelae.

Still, there are some minor issues that could improve the robustness of the paper if addressed. I leave it to the authors and editor to decide how far to take these revisions.

Selection into testing - changes in group composition. Time-specific HRs are useful, but they do not resolve the concern that there may be selection into testing and that this selection varies over the pandemic period.

Moreover, the claim that selection into testing is “of little importance” for the positive-vs-negative comparison is not justified without strong assumptions (too strong, most likely). Specifically, there are unobservable factors potentially affecting who chooses to test, and who is likely to test positive/negative, and these may vary across the pandemic period.

This time-varying selection into testing cannot be fully ruled out and should be explicitly acknowledged as a limitation.

Streamlining the manuscript

Line 172 / paragraph “SARS-CoV-2 positive test results compared with prescriptions for anti-infective agents regarding rates of long-term sequelae.” I recommend moving this paragraph to the Supplementary Material. Even after multiple readings, its purpose and interpretation remain unclear, and it tends to confuse rather than clarify. This edit should not conflict with preregistration. The main text can note that additional preregistered analyses were conducted, while the detailed results and rationale are presented in the Supplementary Material.

More generally, this relates to a point raised previously (and also by the other referee): the paper would benefit from a tighter focus. At present, the Supplementary Material risks overwhelming the reader.

Line 327/328 “The elevated rates of mental disorders observed among individuals with positive compared to negative tests could have several explanations” should be the other way around, the elevated rates were among the negatively tested.

Line 46/47 “In conclusion, COVID-19 post-infection sequelae are comparable to sequelae observed after other infections of similar severity.” Consider changing to “severe COVID-19 post-infection...” since this analysis is only concerned with in-patient and out-patient sequelae the stated conclusion seems too strong.

In general the fact that this analysis only concerns severe cases of post-infection afflictions should be incorporated more in the analysis.